

# Reviews and syntheses: Review of proxies for low-oxygen paleoceanographic reconstructions

Babette A. A. Hoogakker[1], Catherine Davis[2], Yi Wang[3], Stephanie Kusch[4], Katrina Nilsson-Kerr[5], Dalton S. Hardisty[6], Allison Jacobel[7], Dharma Reyes Macaya[1,8,9], Nicolaas Glock[10], Sha Ni[10], Julio Sepúlveda[11], Abby Ren[12], Alexandra Auderset[13], Anya Hess[14], Katrin Meissner[15], Jorge Cardich[16], Robert Anderson[17], Christine Barras[18], Chandranath Basak[19], Harold J. Bradbury[20], Inda Brinkmann[21], Alexis Castillo[9], Madelyn Cook[22,23], Kassandra Costa[24], Constance Choquel[21], Paula Diz[25], Jonas Donnenfield[26], Felix J. Elling[27], Zeynep Erdem[28], Helena L. Filipsson[21], Sebastian Garrido[1,9], Julia Gottschalk[29], Anjaly Govindankutty Menon[10], Jeroen Groeneveld[30], Christian Hallmann[31,32], Ingrid Hendy[22], Rick Hennekam[33], Wanyi Lu[34], Jean Lynch-Stieglitz [35], Lelia Matos[36,37], Alfredo Martínez-García[38], Giulia Molina[36,37], Práxedes Muñoz[39], Simone Moretti[38], Jennifer Morford[40], Sophie Nuber[12], Svetlana Radionovskaya[41], Morgan Reed Raven[42], Christopher J. Somes[43], Anja S. Studer[44], Kazuyo Tachikawa[45], Raúl Tapia [30], Martin Tetard[46], Tyler Vollmer[35], Shuzhuang Wu[47], Yan Zhang[48], Xin-Yuan Zheng[49], Yuxin Zhou[42]

[1]The Lyell Centre, Heriot-Watt University, Edinburgh, EH14 4AP, UK
[2]North Carolina State University, Raleigh, NC 27607, USA
[3]Tulane University, New Orleans, LA70118, USA
[4]Institute of Marine Sciences, University of Quebec Rimouski, Rimouski, QC G5L 3A1, Canada
[5]Department of Earth Sciences, University of Bergen and Bjerknes Centre for Climate Research, Bergen, 5007, Norway
[6]Department of Earth and Environmental Sciences, Michigan State University, East Lansing, MI 48824, USA
[7]Department of Earth and Climate Sciences, Middlebury College, Middlebury, VT 05753, USA
[8]Center for Marine Environmental Sciences, University of Bremen, Bremen, 28359, Germany
[9]ANID, Millennium Science Initiative Program Nucleo Milenio UPWELL, La Serena, 1305, Chile
[10]Institute for Geology, Hamburg University, Hamburg, D-20146, Germany
[11]Department of Geological Sciences and Institute of Arctic and Alpine Research, University of Colorado Boulder, CO 80309, USA
[12]Department of Geosciences, National Taiwan University, Taipei 106, Taiwan
[13]School of Ocean and Earth Science, University of Southampton, Southampton, SO14 3ZH, UK
[14]Department of Earth and Planetary Sciences, Rutgers, the State University of New Jersey, NJ 08854, USA
[15]Climate Change Research Centre and ARC Centre of Excellence for Climate Extremes, University of New South Wales, Sydney, NSW 2052, Australia
[16]CIDIS-Facultad de Ciencias e Ingeniería, Universidad Peruana Cayetano Heredia, Lima, Lima 15102, Peru
[17]Lamont-Doherty Earth Observatory of Columbia University, NY 10964, USA
[18]Laboratoire de Planétologie et Géosciences, Université d'Angers, 49000 Angers, France
[19]Department of Earth Sciences, University of Delaware, Newark, DE 19716, USA
[20]Department of Earth, Ocean and Atmospheric Sciences, University of British Columbia, Vancouver, V6T 1Z4, Canada
[21]Department of Geology, Lund University, Lund, 223 63, Sweden
[22]Department of Earth and Environmental Sciences, University of Michigan, Ann Arbor, MI 48109, USA
[23]Department of Geosciences, University of Arizona, Tucson, AZ 85721, USA
[24]Department of Geology and Geophysics, Woods Hole Oceanographic Institution, MA 02543 USA
[25]Centro de Investigación Mariña, XM1, Universidade de Vigo, Vigo, 36310 Spain
[26]College of Earth, Ocean, and Atmospheric Sciences, Oregon State University, Corvallis, OR 97331, USA



[27]Leibniz-Laboratory for Radiometric Dating and Isotope Research, Christian-Albrecht University of Kiel, Kiel, 24118, Germany

[28]Department of Marine Microbiology & Biogeochemistry, NIOZ Royal Netherlands Institute for Sea Research, 't Horntje (Texel), 1797 SZ, The Netherlands

[29]Institute of Geosciences, Kiel University, Kiel, 24118, Germany

[30]Institute of Oceanography, National Taiwan University, Taipei 106, Taiwan

[31]GFZ German Research Centre for Geosciences, Potsdam, 14473, Germany

[32]Institute of Geosciences, University of Potsdam, Potsdam, 14476, Germany

[33]NIOZ Royal Netherlands Institute for Sea Research, Department of Ocean Systems, Den Burg (Texel), 1790 AB, The Netherlands.

[34]Tongji University, Shanghai 200092, China

[35]School of Earth and Atmospheric Sciences, Georgia Institute of Technology, Atlanta, GA 30332, USA

[36]Centre of Marine Sciences (CCMAR), University of Algarve, Faro, 8005-139, Portugal

[37]Marine Geology and Georesources Division, Portuguese Institute for the Sea and Atmosphere (IPMA), Lisbon, 1495-165, Portugal

[38]Max Planck Institute for Chemistry, Mainz, 55128, Germany

[39]Departamento de Biología Marina, Universidad Católica del Norte, Coquimbo, Chile

[40]Chemistry Department, Franklin & Marshall College, Lancaster, PA 17604, USA

[41]Department of Earth Sciences, University of Cambridge, Cambridge, CB23EQ, UK

[42]Earth Science Department, University of California, Santa Barbara, CA 93106, USA

[43]GEOMAR Helmholtz Centre for Ocean Research Kiel, Kiel, 24148, Germany

[44]Department of Environmental Sciences, University of Basel, Basel, 4056, Switzerland

[45]Aix Marseille Univ, CNRS, IRD, INRAE, Coll France, CEREGE, Aix-en-Provence, 13331, France

[46]GNS Science, Lower Hutt, 5040, New Zealand

[47]Institute of Earth Sciences, University of Lausanne, Lausanne, CH-1015, Switzerland

[48]Ocean Sciences Department, University of California, Santa Cruz, CA 95064, USA

[49]Department of Earth and Environmental Sciences, University of Minnesota Minneapolis, MN 55455, USA

*Correspondence to*: Babette A.A. Hoogakker (b.hoogakker@hw.ac.uk) and Catherine Davis (cdavis24@ncsu.edu)

**Abstract.** A growing body of observations reveals rapid changes in both the total inventory and distribution of marine oxygen over the later half of the 21st century, leading to increased interest in extending oxygenation records into the past. Use of paleo-oxygen proxies have the potential to extend the spatial and temporal range of current records, bound pre-anthropogenic baselines, provide datasets necessary to test climate models under different boundary conditions, and ultimately understand how ocean oxygenation responds beyond decadal scale changes. This review seeks to summarize the current state-of-knowledge about proxies for reconstructing Cenozoic marine oxygen: sedimentary features, sedimentary redox-sensitive trace elements and isotopes, biomarkers, nitrogen isotopes, foraminiferal trace elements, foraminifera assemblages, foraminifera morphometrics, and benthic foraminifera carbon isotope gradients. Taking stock of each proxy reveals some common limitations in that the majority of proxies function best at low-oxygen concentrations and many reflect multiple environmental drivers. We also highlight recent breakthroughs in geochemistry and proxy approaches for constraining pelagic (in addition to benthic) oxygenation that are rapidly advancing the field. In light of both the emergence of new proxies and the persistent multiple driver problem, the need for multi-proxy approaches and FAIR data storage and sharing is emphasized. Continued



refinement of proxy approaches and both proxy-proxy and proxy-model comparisons are likely to support the growing needs of both oceanographer and paleoceanographers interested in paleo-oxygenation records.

## 1 Introduction

Dissolved oxygen in the oceans is necessary to sustain aerobic life, control biogeochemical processes, and is closely linked to carbon remineralization, export, and storage. Oxygen in the ocean has declined since at least the mid-20th century. This
decrease has been observed in estuaries and coastal regions (Diaz & Rosenberg, 2008; Rablais 2009, Rablais et al., 2010; Conley et al., 2011), continental shelves, and the open ocean (Schmitko et al., 2017; Chan et al., 2008; Bograd et al., 2008; Breitburg et al., 2018; Keeling et al., 2009; Levin, 2017; Stramma et al., 2008; Stramma et al., 2010). Direct measurements of oxygen have only been routine for decades at most, and even then, are spatially limited. Inaccessible subsurface regions and open ocean features, such as oxygen minimum zones (OMZs), are especially difficult to monitor. Thus, proxies are required
to extend modern records and investigate long-term drivers of deoxygenation.

Drivers of ocean deoxygenation include ocean warming, causing decreasing oxygen solubility in seawater and increasing remineralization; increased productivity leading to higher subsurface oxygen utilization during respiration; and decreased ventilation, due to changes in circulation or stratification (Keeling et al., 2009; Breitburg et al., 2017). These drivers can influence ocean deoxygenation on different timescales and to different degrees. Warming is a key driver of modern
deoxygenation in the open ocean as well as in coastal systems (Schmitko et al., 2017; Levin, 2018; Rabalais et al., 2010). In coastal systems, anthropogenic nutrient increases (eutrophication) from activities such as sewage efflux and fertilizer input, is frequently the primary cause of deoxygenation on short time scales (Rabalais et al., 2010; Breitburg et al., 2018). Productivity changes can be equally important in driving decadal (Deutsch et al., 2011, 2014) to centennial and longer scale changes in open ocean settings (e.g., Dickens & Owen, 1994; Hendy et al., 2004). Ventilation changes may act across different scales of
space and time. For example, deoxygenation induced by stratification can be variable on timescales of days to years and beyond, especially in coastal regions and restricted basins (reviewed in Rabalais et al., 2010). However, seawater oxygen content is also responsive to ventilation changes on centennial, millennial and longer time scales, associated with changes in deep water source, upwelling, overturning circulation, ocean gateway dynamics, and the geometry of whole ocean basins (Hoogakker et al., 2015; Fyke et al., 2015; Cardich et al., 2019; Auderset et al., 2022; Hess et al., 2023).

Climate models indicate that a decrease in dissolved oxygen concentrations will continue for hundreds to thousands of years into the future (Bahl et al., 2019; Kwiatkowski et al., 2020; Oschlies 2021; Gulev et al., 2021). The combined effect of future warming and seawater oxygen depletion could have adverse impacts on the marine environment, potentially culminating in a mass extinction rivalling those in Earth's past (Penn and Deutsch, 2022). However, recent oceanic oxygen loss inferred from observations, especially in the deep ocean, is currently underestimated in state-of-the-art climate models, which also do not
reproduce the observed patterns of deoxygenation in the tropical thermocline (Oschlies, 2018). This mismatch is likely due to i) unresolved circulation, mixing, and transport processes; ii) misrepresentation of respiratory oxygen demand, iii) missing




biogeochemical feedback mechanisms, and iv) insufficient simulation length to reach equilibrium in the deep ocean (Oschlies, 2018).

One of the main challenges in modelling concentrations of marine oxygen is the representation of ocean physics, which in some regions crucially depends on small-scale processes. A realistic representation of OMZs (Busecke et al., 2019; Montes et al., 2014), and deep-water formation (Heuzé, 2021) requires high(er)-resolution ocean models. Eddy-resolving ocean models are best suited to represent these regions and water masses, but these high-resolution models are computationally very expensive. They cannot be integrated long enough for the simulated ocean and its tracers to reach equilibrium. Seawater oxygen content in the deep ocean is a tracer that needs especially long equilibrium times, requiring simulations of thousands of model years. Another challenge is the representation of biogeochemical processes in models (Fennel et al., 2022). Half of the CMIP6 coupled climate and Earth System models do not include any representation of marine biogeochemistry (IPCC, 2021: Annex II). If they do, the mathematical representations of marine ecosystems and biogeochemistry are quite simple. These ecosystem models are also under-constrained and tuned to reproduce present day observations. Any existing physical model biases will, thus, impact the biogeochemical parameters and lead to biogeochemical (including oxygenation) biases (Duteil et al, 2012). In addition, these models are missing important processes. For example, most climate models include NPZD (nutrient, phytoplankton, zooplankton, detritus) models with only one single phytoplankton functional type representing the biodiversity of the whole planet. Only a few models consider several nutrients and/or several phytoplankton functional types that can then compete against each other for nutrients and light. These more complex models are able to simulate ecosystem shifts due to changes in climate and nutrient availability and resolve species competition at regional scale. Shifts in predominant plankton types may be important as they can result in changes in the local abundance, size, sinking speed and quality of sinking organic particles, and therefore remineralisation depth and oxygen levels. But more complex models have even more tuneable parameters and are therefore even more under-constrained than the simple NPZD models (Andersen, 2005). Finally, feedbacks involving sediment/ocean fluxes that can impact biogeochemistry, biology, and therefore oxygen concentrations, such as the release of phosphorus and iron from anoxic sediments (Niemeyer et al., 2017), are generally not included in current models.

To better constrain biological and physical processes in the ocean, and improve their representation in models, we need dedicated observational programs. We also need proxy-based oxygen reconstructions from the geologic past when the climate system was different to present day to test numerical models and to improve process understanding.

## 2 Proxies

Proxies provide indirect representations of environmental variables that cannot be measured directly. Examples include seawater temperature, pH, and dissolved oxygen. A proxy is a measurable physical or chemical variable that is conserved in a natural climate archive and allows us to infer information about the variable of interest in a qualitative or quantitative manner. To build a useful proxy, it is important to understand how the proxy relates to the variable of interest and what other environmental parameters might influence the proxy pre- and post-deposition in sediments. This involves understanding the



biology (especially if the proxy is captured in fossil and organic material), chemistry, and physics of both proxy and
sedimentary systems.

Paleo-oxygen proxies are generally developed and calibrated through a combination of theoretical, empirical, and experimental
approaches. A theoretical approach to proxy development is exemplified in geochemical subfields (i.e., inorganic and organic
geochemistry of sediments and biogenic calcites). For example, a theoretical understanding of redox potential can lead to
robust predictions about concentrations of elements and ions across oxygen gradients, and thus the directionality of their
incorporation into sediments or biogenic minerals. Theoretical approaches generally require empirical validation as many
complexities remain difficult to model. For example, redox-associated chemistry and incorporation of products into biogenic
minerals is biologically mediated, and influenced by other environmental (e.g., temperature) variables, and taxa-specific
dynamics related to their life cycle, metabolism and ontogeny. As a result, theoretical development is usually limited to the
identification of proxies of interest and qualitative predictions.

The use of recently deposited sediments on the sea floor (frequently referred to as 'core-tops') recovered across natural oxygen
gradients is the most frequent empirical approach. Core-top calibrations can be critical for proxies that require timescales or
depositional environments difficult to replicate in a laboratory setting, such as foraminiferal assemblages, sedimentary features,
and sedimentary trace metals. This approach has the benefit of testing how a proxy manifests in the complex natural
environment. One key limitation is the need to deconvolve highly correlated environmental controls, such as productivity,
organic carbon content, and oxygen, which are classically difficult to disentangle as drivers of foraminiferal assemblages
(Gooday, 2003). This may also impact other proxies such as the isotopic composition of nitrogen ($\delta15N$) and organic matter.
The second key limitation of core-top calibrations is the no-analogue problem; extrapolation beyond modern examples may
be required to describe paleo-oxygen environments which are unlike current conditions, particularly during the more extreme
events of ocean deoxygenation found in the geologic record. Core-top calibrations are also limited where geographical or other
cryptic initial states are incompatible. For instance, differing concentrations of redox-sensitive elements in source waters, or
seed stocks of key species for biomarkers or fossil assemblages. Furthermore, core-tops are not always modern, in some cases
having been deposited thousands of years or more before they were collected (Mekik and Anderson, 2018). Sediment trap
studies and plankton tows are other examples of important, yet less-frequently used empirical approaches.

Experimental approaches are often considered the 'gold-standard' for quantitative calibration of single-driver proxies. As of
now, most paleo–oxygen proxies are qualitative or semi-quantitative. Experimental approaches have the benefit of allowing
for the isolation of single controlled variables and have been used to greatest effect so far in biogenic calcites. However, there
are a few drawbacks to this approach. The first is that proxies are removed from the complexities of the natural environment,
thus results must be validated with field observations where possible. In other cases, the timescales (e.g., sedimentary features,



sedimentary trace metals) or complex initial conditions (e.g., biomarkers assemblages) necessary to replicate natural observations are difficult or impossible to generate in a laboratory setting.

## 3 Terminology around seawater oxygen concentrations

As will be evident from the discussion of the different proxies, the nomenclature to define different oxygenation 'stages' has historically been inconsistent and confusing (Canfield and Thamdrup, 2009). This can be ascribed in part to the increasing interdisciplinarity of modern oxygen research. Classically, geochemists define an oxic zone, supporting aerobic metabolism, followed by an oxygen-depleted zone, sometimes referred to as suboxic, where metabolism is supported by nitrate-, manganese (Mn) and iron (Fe) -reduction, and an anoxic zone where metabolism is supported by sulphate reduction and methanogenesis

(Froelich, 1979; Berner, 1981). However, this scheme has been regarded as confusing and contradictory by Canfield and Thamdrup (2009) who proposed instead to use terminal electron acceptors and respiration processes to define chemical/metabolic zones (Figure 3.1). Ecologists and biologists have frequently focused on oxygen levels associated with negative outcomes for aerobic organism (fish, crustaceans, etc.), and have defined a sublethal threshold and lethal oxygen concentrations, which greatly vary among taxa and may be influenced by other factors such as temperature (Vaquer-Suyer and

Duarte, 2008). This sublethal threshold is referred to as hypoxia. It leads to mortality events, loss in biodiversity, habitat reduction, predation potential and disruption of life cycling (Service, 2004; Rabalais et al., 2002). The dearth of observational oxygen data at the full range of spatial and temporal scales applicable to either geochemical or ecological systems further complicates definition in terminology. To avoid confusion between the different terms used, an illustrative Figure 3.1 is provided to give a sense of the zonation, chemical speciation and metabolic processes, alongside the 'oxygen working' range

of the different proxies.



**Figure 1: Overview of oxygen "stage" nomenclature used in this review. A)** shows the ranges most often associated with the descriptive terms OMZ, Oxygen Deficient Zone (ODZ), anoxia, dysoxia/hypoxia, and anoxia in seawater. In **B)** oxygen concentrations are shown on a log linear scale along with a simplified schematic of several proxy-relevant components of other redox-sensitive reactions. Chemical concentrations other than oxygen are non-dimensional, but all relate to scales in both A) and C). The redox ladder is modified from Canfield and Thamdrup (2009). **C)** shows the ranges of oxygen and/or redox chemistries over which different proxy types can be used to reconstruct paleo-environments, based on proxies applied to sediment samples. Proxy types are ordered as they are discussed in the manuscript, with section numbers associated with each. Proxy types shown in olive can be used to reconstruct oxygen from benthic settings, those in green can be used for pelagic settings.



## 4 Justification for the review Section

Interest in seawater oxygen proxies is increasing, partly due to current trends of ocean deoxygenation and uncertainties about the future at different timescales. A methodological overview of proxies was included in Moffitt et al. (2015). Since this review was published, methodological developments, updates, and insights have emerged that were not captured previously, or were applied in older sediments. Some of these new methods are being designed to reconstruct the upper open ocean histories.

This review is limited to proxies that can be applied through the Cenozoic (i.e., the last 66 million years), although we briefly touch upon some well-studied earlier examples, such as Cretaceous oceanic anoxic events (OAEs). The focus on the Cenozoic, when our oceans were overall well oxygenated, allows an investigation of scenarios and timescales most immediately relevant to inform the future.

Extending modern records into the past may provide baselines for pre-anthropogenic marine oxygen content and necessary data to test climate models under different boundary conditions from today and improve process understanding. While the past is only a partial analogue to the future, it can provide a portfolio of oxygen scenarios to bound future projections. This is especially the case for past climate episodes that were characterized by greenhouse gas concentrations similar to those projected into the future. In step with the growing interest in modern and future ocean oxygen, there has been rapid proxy development over the past decades. Implementation of new technologies as well as a burgeoning interest in paleo-oxygenation has led to an influx of new proxies and refocusing and advancement of established proxies.

## 5 Format of the review

This review is meant to serve as an overview of the current state of proxy development at a pivotal moment for the field. We summarize the major classes of proxies for the benefit of both new and experienced paleoceanographers, and those working in adjacent fields. It is our hope that an introduction to and update on the suite of available proxies will increase their utility for those interested in marine oxygen research. Moreover, we hope that a clear discussion of current limitations and future directions can pave the way for improving the tools at our disposal for generating new paleo-oxygenation records.

Our review of the various proxy methods is split into a traditional overview of sediments as proxy carriers (section 6.1), followed by a discussion about sedimentary redox trace elements and isotopes (section 6.2), organic proxies (section 6.3), nitrogen isotopes (section 6.4), foraminiferal trace elements (section 6.5), foraminifera assemblages (section 6.6), foraminifera morphometrics (section 6.7), and finally benthic foraminifera carbon isotope gradients (section 6.8). This is then followed by a discussion about the benefits of multi-proxy applications, and data storage and management.





## 6 Proxy methods

### 6.1 Sediments as Proxy Carriers

Reconstructions of past marine environments rely on sediment samples, from deep sea cores, or sedimentary outcrops of uplifted marine sediments. Sedimentary observations form the backbone of metadata essential to support the growing arsenal of proxies employed to define Earth's biogeochemical evolution. In particular, quantitative mineralogy and lithologic descriptions should accompany sample archives to support existing and future geochemical proxy measurements and interpretations. This is critical as a given proxy may only be applicable to specific rock types or biogenic material or may

require alternative interpretative frameworks depending on mineralogy or lithology. Programs like IODP (International Ocean Discovery Program) have prioritized lithologic metadata alongside formalized and accessible sample archives. However, samples collected by individual labs may lack these data and/or accessible archives in the absence of collaboration. Recently developed databases, such as the Sedimentary Geochemistry and Paleoenvironment Project (SGP), are working to circumvent some of these issues by requiring lithologic context and detailed sediment descriptions to accompany geochemical data

submissions. Importantly, sedimentary features are crucial to guide sample selection for quantitative analyses, especially intervals with specific redox interests. For example, descriptors such as changes in organic carbon content, laminations, and the deposition of pyrites can be useful first indicators of sedimentary redox/oxygen changes.

### 6.1.1 Historical based sedimentary redox / bottom water oxygen reconstructions

The presence/absence of laminations (Fig. 2) have historically played an important role in reconstructing low-oxygen systems,

and they remain a popular tool today. The presence of laminations is a key indicator of conditions that are inconsistent with the survival of benthic fauna beyond seasonal timescales. These observations, along with biofacies oxygen indices considering bioturbation, fauna, diversity, body size, and trophic levels, have been used to characterize paleoredox conditions, including specific oxygen levels (Rhoads and Morse, 1971; Behl and Kennett, 1996; Sperling et al., 2022). More recent work demonstrates that both carnivory and vision are also linked to environmental oxygen levels (Sperling et al., 2013; McCormick

et al., 2019). These indices have been used to reconstruct oxygen trends during the evolution of early animal life (Sperling et al., 2015; Canfield and Farquhar, 2009; Boyle et al., 2014; Tarhan et al., 2015; van de Velde et al., 2018), oxygen impacts on mass extinctions (Reddin et al., 2020; Sampaio et al., 2021), as well as local oxygen levels independent of broader evolutionary context.

It is worth noting that laminations can be caused by factors unrelated to redox changes, and thus interpretations of laminations should only be made after scrutinizing sedimentology and should be complemented with geochemical techniques. For instance, laminated sediments have been found in the Southern Ocean (e.g., Kemp et al. 1996, 2000, 2006, Grigorov et al., 2002), Eastern Equatorial Pacific (Kemp & Baldauf 1993, Kemp et al., 1996), and the North Atlantic (Boden & Backman 1996) associated with diatom mats or giant diatoms. Laminae in these environments are the consequence of diatom mats (e.g.,



Thalassiothrix spp.) suppressing benthic activity (King et al., 1995). Additionally, laminations can also form due to grain size changes associated with particle sorting in sediment gravity flows, sediment-bed interaction, and seasonal to interannual changes in the grain size of settling particles (e.g., coarse terrigenous input in a wet season alternating with finer sediments) (Kemp 1996, O'Brien 1996).

The presence and relative abundance of pyrite, and observations of pyrite's crystal structure have been applied as indicators of water column euxinia (Fig. 2). Specifically, the size distribution of framboidal pyrite has been proposed to reflect formation in euxinic versus more oxidizing water columns. Smaller size framboids found in the euxinic Black Sea, for example, are interpreted to reflect a fast growth rate within the euxinic water column, as opposed to formation under longer timescales within sulphidic sediments (Wilkin et al., 1996; 1997a; 1997b; Wignall and Newton, 1998). The size fraction of pyrite

framboids has been applied and calibrated extensively within OAEs, but also other intervals, with widespread geochemical evidence of marine anoxia (Wilkin et al., 1996; Wignall et al., 2005; Kuroda et al., 2005; Jenkyns et al., 2010). Because syngenetic water column-formed pyrite can incorporate or absorb trace elements (Huerta-Diaz and Morse, 1992), the trace element content of pyrite is an important paleoredox archive (Large et al., 2014).

### 6.1.2 Non-destructive methods for sediment analyses

Observations of sedimentary facies are used as a first-order evaluation of the depositional environment. Traditional methods (e.g., non-destructive core description and physical property measurements) are a fast, low-cost initial qualitative way to interpret redox/oxygen conditions. Over the past few decades, there have been important technological advances to describe sedimentary features in quantitative and non-destructive ways. Analytical instruments and imaging technology (e.g., microtomography, X-ray fluorescence (XRF) scanner, multi-sensor core logger, scanning electron microscopy) can further

improve the spatial and temporal resolution of the sedimentological and physical property measurements (see below), allowing non-destructive and 3D detection of various sedimentary features on the sub-millimetre to micron scale.

**X-ray computed tomography (CT)** is a high-resolution (~0.1-1 mm) imaging technique that allows visualization of 3D structure of objects, determined by X-ray attenuation associated with variations of density and element compositions in

sedimentary records (Fig. 2 see reviews by Mees, 2003 and St-Onge, 2007). CT imaging can be used on both whole round and section halves of sediment cores with minimal pretreatment. As a non-destructive method, it has been used to determine physical properties of sediments (e.g., density, porosity, and grain size (Fortin, 2013; Tanaka, 2011; Orsi, 1994; Amos, 1996; Mena, 2015)) and to identify benthic communities (e.g., bioturbation analysis and trace fossil detection/ichnological analysis in the sediments (Dorador, 2020; Rodriguez-Tovar, 2022)). Microstructure information obtained using CT has greatly

improved the accuracy of sedimentological description, whereas physical property data are critical for understanding oxygen penetration in the sediment profile and subsequent diagenetic processes controlled by pore water redox concentrations.



**Multi-Sensor Core Logger (MSCL)** or **Multi-Sensor Track (MST)** is widely used for continuous measurements of physical properties on centimetre scales in either whole round or section halves of sediment cores. These core loggers are usually equipped with detectors for measuring magnetic susceptibility, gamma ray density, natural gamma radiation, p-wave velocity, and resistivity, which provides density, porosity, and Fe-bearing mineral information for first order evaluation of ambient redox state in pore waters.

**XRF scanners** measure the relative abundance of elements (from Al to U, following the periodic table) on section half sediment cores with sub-millimetre to centimetre (i.e., high-resolution) scales in non-destructive manner (Fig. 2., Croudace 2006, Crudace 2015, 2019). XRF data are considered semi-quantitative as elemental variability in the sediment cores is measured as counts and not concentrations. XRF data quality is affected by X-ray tube ageing, water content, smoothness of the sample surface, and grain size (Böning et al., 2007; Tjallingii et al., 2007; Weltje and Tjallingii, 2008). Thus, appropriate sample preparation (e.g., core scraping and use of a thin polyester XRF film to smooth the surface) is required for high-quality data acquisition (Löwemark, 2019). Additionally, sediment composition (e.g., organic carbon and calcium carbonate content) may affect XRF counts because lighter elements (e.g., carbon, nitrogen, and oxygen) are outside of the XRF detection range. For instance, higher sedimentary organic carbon can dilute the number of counts for all elements (Löwemark, 2010). Normalization of the absolute counts with respect to an element that is less affected by biological and diagenetic processes (e.g., normalizing to Al) is used to assess the relative variability of elemental compositions (Löwemark, 2010). Despite the limitations, scanning XRF is able to provide high-resolution data with fast and non-destructive measurements, allowing a first-order assessment of redox-sensitive element abundance (e.g., molybdenum, uranium, and manganese) prior to more labour-intensive analyses (e.g., solution-based bulk elemental concentration analyses, as discussed in Section 6.2).

### 6.1.3 Fossils contained in sediments

Sediments provide the backbone for any marine paleo-environmental reconstruction, especially its preserved or fossilized biogenic materials. This can include morphologically identifiable skeletal material such as foraminiferal tests or diatom frustules, or 'molecular fossils'.

Foraminifera (Kingdom Chromista; Infrakingdom: Rhizaria; Order Foraminiferida) are amoeboid protists characterized by a cytoplasmic body covered by a shell or 'test' of one or more interconnected chambers. The test wall can be made of agglutinated particles, organic material, or biomineralized crystals of calcite, aragonite, or rarely silica (Loeblich and Tappan, 1988). Calcareous tests, in particular, are frequently preserved in marine sediments after death or reproduction (Debenay, 2012). As a result, a rich fossil record of foraminifera extends from the Cambrian into the present (Loeblich and Tappan, 1988; Sen Gupta, 2003; Debenay, 2012). Foraminifera have colonized a diversity of environments. The majority are benthic, where they occupy virtually any water depth and substrate, and on and into the sediment up to tens of cm deep (e.g., Vickerman, 1992; Gooday, 2003, Sen Gupta, 1999). Others are planktic, with habitats ranging from the ocean's surface into the



mesopelagic (Schiebel & Hemleben, 2017). As a result, where preserved, foraminifera can offer a near continuous record of ecological succession, with individual shells capturing environmental conditions over their week to years-long lifespans. Thus, foraminifera form the basis of several of the proxy methods for seawater oxygen reconstructions (sections 6.4 - 6.8).

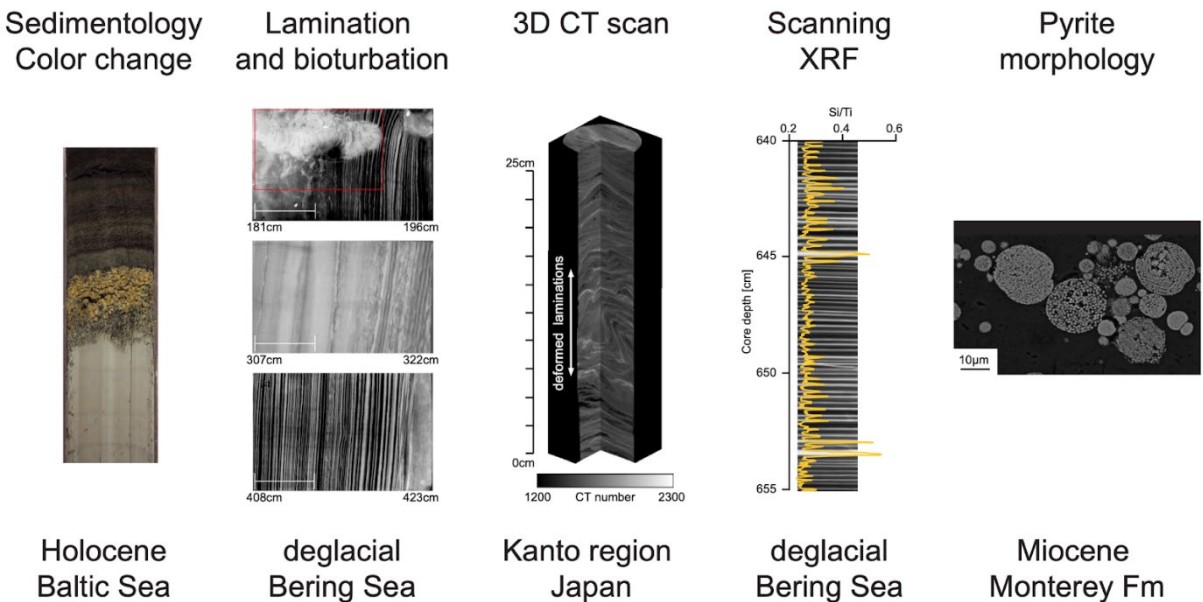

**Figure 2: Examples of sedimentary features discussed in the main text. Left to right: clay to laminated gyttja sediments at the transition from Baltic Ice Lake to Littorina Sea in the early Baltic Sea Holocene; laminations from the Bering Sea ODZ during the last deglacial; 3D CT scan from sediments deformed during the 2011 Tohoku Earthquake from offshore Japan; SEM image of framboidal pyrite from the Miocene Monterey Formation of California, USA (adapted after Berndmeyer et al., 2012; Kuehn et al., 2014; Nakashima and Komatsubara, 2018).**

## 6.2 Sedimentary redox trace elements and isotopes

### 6.2.1 Introduction

The potential for the concentration of trace metals in sediments to act as proxies for past Earth surface conditions has been recognized since early observation of metal enrichments in organic-rich sediments (Goldschmidt, 1954). Trace metals represent one of the most commonly used proxies for the reconstruction of paleo-redox conditions in sediments (Algeo, 2004; Algeo & Maynard, 2004; Algeo & Rowe, 2012; Bennett & Canfield, 2020; Algeo & Li, 2020; Brumsack, 2006; Calvert & Pedersen, 1993; Little et al., 2015; Morford & Emerson, 1999; Nameroff et al., 2004; Scott & Lyons, 2012; Sweere et al., 2016; Tribovillard et al., 2006; Calvert & Pedersen, 2007; Zhou & McManus, 2023).

Sedimentary trace metal enrichments are associated with precipitation and/or adsorption of metals from the ambient bottom and/or pore waters along a redox gradient (redox potential Eh) primarily controlled by decomposition of organic carbon using various oxidants (Calvert 2007, Froelich et al., 1979). These redox reactions proceed in a well-defined sequence (Fig. 3 and





3.1), during which trace metals may be scavenged from ambient waters and subsequently enriched in sediments (i.e., authigenic

enrichment as distinct from detrital input) as a result of changes in valence state (e.g., Mn, U, Re, and Mo) and/or speciation

(e.g., Cd, Ni, and Zn with solid phase precipitation but no valence change) (Algeo & Li, 2020). Because sedimentary Eh varies

in response to both bottom water oxygen availability and the rain rate of organic carbon, reconstructions using redox-sensitive

elements to reconstruct bottom water oxygen must explicitly account for changes in the rain rate of organic carbon (see Section

6.2.4).

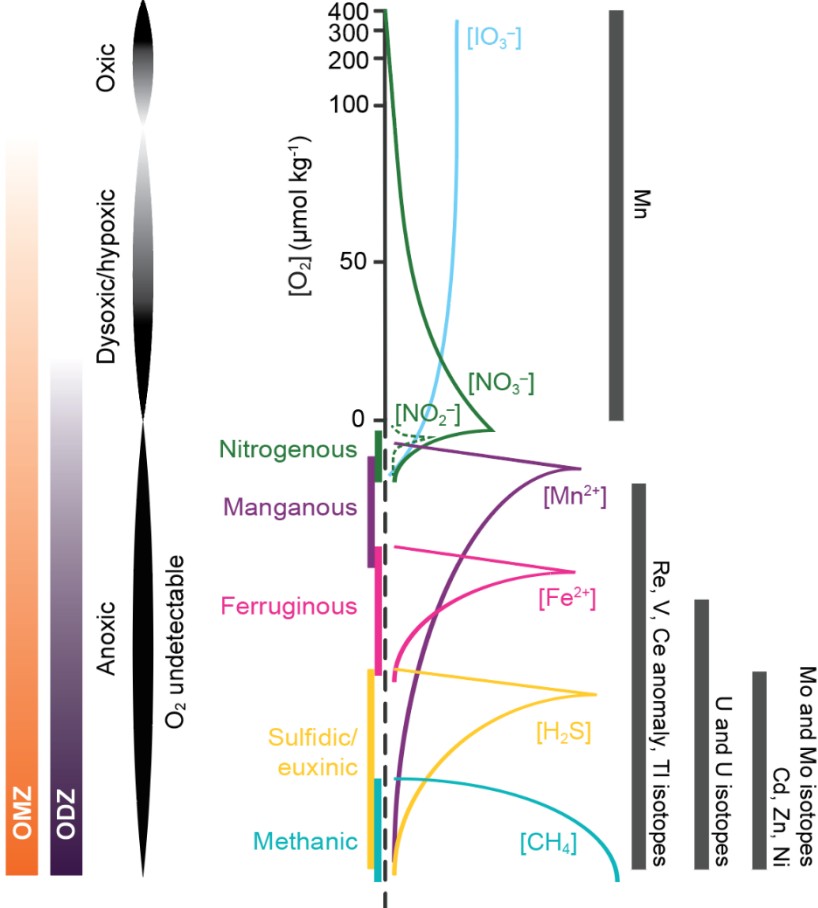

**Figure 3: Redox ladder (modified from Fig. 1 in the introduction) and redox-sensitive trace metals and metal isotopes discussed in this section. Dissolved oxygen ranges for OMZs, ODZs, anoxia, dysoxia/hypoxia (low-oxygen), are labelled in the figure. The redox ladder is modified from Canfield and Thamdrup (2009). The oxygen/redox potential range for use of each redox-sensitive metal and metal isotope redox proxy is shown as coloured bars on the right.**

Trace element analysis has the advantage of facilitating "multi-proxy" data acquisition. Sedimentary trace metal concentration

measurements are free from vital effects compared to trace metal incorporation in biogenic carbonate (e.g., foraminifera shells,

see Section 6.5), and they are particularly valuable when carbonate preservation is poor and sediments have remained relatively

undisturbed post-deposition. Recently, a better understanding of redox-sensitive metal preservation in surface sediments and



applications of statistical techniques have made it possible to quantify dissolved oxygen concentrations in some coastal systems
of the Eastern Pacific (Valdés et al., 2021; Sánchez et al., 2022; Costa et al., 2023), opening the door for additional regional
redox-sensitive trace metal calibrations and creating new possibilities for quantitative oxygen reconstructions.

Isotopic fractionation of redox-sensitive metals (e.g., Mo, U, Cr, and Fe) may also occur during the exchange between seawater
and other ocean sinks/sources (e.g., scavenging from reducing water columns), and technical advances in mass spectrometry
have allowed measurements of "non-traditional" stable metal isotope systems and enabled their use in reconstructions of past
ocean oxygenation changes (e.g., Andersen et al., 2017; Kendall et al., 2017; Severmann et al., 2008; Frei et al., 2011).
Compared to authigenic enrichments, redox-sensitive metal isotope proxies may allow for more (semi-)quantitative redox
reconstructions via isotope mass balance, and potentially provide a more globally integrated perspective on ocean oxygen
variability.

Trace element enrichments and their isotopes have provided key insights into ocean processes on various timescales (from
Precambrian to present) and research is ongoing to refine the interpretations of these proxies to shed new light on our
understanding of global ocean oxygen responses to variations in Earth's climate and other environmental variables.

### 6.2.2 Materials needed and analytical methods

Quantitative elemental concentrations are measured on dried sediments, which are either fully digested or partially dissolved
(i.e., leached) to target authigenic phases. One example is authigenic pyrite, which forms in the water column, and can
incorporate or absorb trace elements. As such, the trace elemental contents of pyrite can also be an important paleoredox
archive (Large et al., 2014).

Samples are generally treated using bulk digestion methods such as acid digestion and alkaline fusion, depending on the
sediment composition and elements of interest. To avoid contamination, all sample preparation should be performed in metal-
clean laboratories with acid-cleaned vessels and trace-metal grade chemicals. Acid digestion is commonly used to analyse
major, minor, and trace elements, and involves using one or more strong acids (e.g., $HNO_3$, $HCl$, $HF$, and $HClO_4$). The acid
digestion process is usually performed on a hotplate or in a microwave system that provides a high temperature (up to 200°C)
and/or pressurized environment to facilitate sample digestion. Incomplete digestion may occur when sediments contain a
considerable amount of refractory minerals (e.g., zircons that are difficult to digest) and should be carefully monitored (Huang
et al., 2007; Bertand et al., 2012b). Alkaline fusion is more efficient in digesting refractory minerals, by fusing sediment
samples in a crucible with a mixture of alkaline substances (e.g., $LiBO_2$) at very high temperatures (>1000 °C) (Hu and Qi,
2014). The fusion process produces a glass bead that can be dissolved in $HNO_3$ for solution-based analysis or directly used for
in -situ analysis with laser ablation (Eggins, 2003). Care must be taken to avoid cross-contamination in labs where B and Li



are measured as carbonate system proxies. Additionally, the use of high temperatures may lead to loss of volatile elements (e.g., As, Re and Pb), and further studies are required to understand potential impacts of high temperature on other trace metals.

Major and minor element compositions can be measured using quantitative X-ray fluorescence (XRF), inductively coupled plasma optical emission/atomic emission spectroscopy (ICP-OES or ICP-AES), atomic absorption spectrophotometry (AAS),

and microwave plasma atomic emission spectroscopy (MP-AES); whereas minor and trace elements may be measured using inductively coupled plasma mass spectrometry (ICP-MS) that has a lower detection limit. Accuracy and precision of the measurements are often assessed by analysing standard reference materials simultaneously (e.g., PACS 1/2 [Marine sediment from the Harbour of Esquimalt] and MESS-3 [Marine sediment from the Beaufort Sea]). To isolate authigenic enrichments of redox-sensitive trace metals in marine sediments, and to obtain enrichment factors, elemental concentrations can be normalized

with respect to detrital (i.e., lithogenic) elements. These are most commonly major elements that appear to be conservative such as Al, Ti, and Li in phyllosilicate-rich sediments derived from glacial erosion (Loring, 1990).

Quantification of enrichment ($E_{EF}$) may be determined following Tribovillard et al. (2006) (Eq. 6.2.1) and Böning et al. (2009) (Eq. 6.2.2):


Equation 6.2.1   $E_{EF} = (EI/NE)_{sample} / (EI/NE)_{background}$

where EI corresponds to the metal of interest and NE corresponds to the element for normalization.

Equation 6.2.2   Metal excess (normalized by Al) = $EI_{sample} - (EI/Al)_{background} * Al_{sample}$

The upper continental crust has been widely used as a lithogenic background reference (Rudnick and Gao, 2003). Because lithogenic background ratios may vary by region, by source materials (e.g., aeolian, river sediment input, and coast line or glacial erosion) and by timescales, care should be taken to determine and cite an appropriate value (see Section 6.2.5.1).


Metal isotope measurements often target authigenic phases to avoid contamination from detrital components. As such, diluted acids (e.g., diluted HCl and $HNO_3$) or weaker acids (e.g., acetic acid) are used in the partial digestion or leaching process. Leaching methods vary between and within labs even for the same isotope measurements (e.g., U isotopes (Tissot et al., 2018)). Initial sample reconnaissance experiments should be used to determine the optimal leaching procedure. For high-precision

stable metal isotope analysis, it is generally necessary to purify the element of interest from sample matrices to avoid possible spectral or non-spectral interferences on the instrument (e.g., through column chemistry).




Metal stable isotopes are now analysed routinely using thermal ionization mass spectrometry (TIMS) or multi-collector ICP-MS (MC-ICP-MS) instruments. The ionization source of TIMS instruments is less energetic compared to a plasma source,

thereby making TIMS analysis less suitable for elements with high first ionization energies. Precise and accurate stable isotope ratio measurements on either type of instruments depend on robust correction of instrumental mass bias produced during analysis. Instrumental mass bias is time-variant during TIMS analysis because of differential evaporation of light versus heavy isotopes from the sample on a heated filament. Robust correction of such mass bias typically requires a double spike method (e.g., Sibert et al., 2001, Ripperger et al., 2007, Tian et al., 2019). In contrast, mass bias produced in MC-ICP-MS is relatively

stable, even though its magnitude is typically larger relative to mass bias on TIMS for the same element. This feature makes it possible to use a sample–standard bracketing (SSB) method as an alternative way for mass bias correction in MC-ICP-MS analysis to tackle isotope systems that only have two stable isotopes (e.g., V and Tl). SSB is also significantly easier and cheaper to implement compared to the double spike (Wu et al., 2016, Nielsen et al., 2016, Nielsen et al., 2004). These advantages have made MC-ICP-MS a dominant tool for stable metal isotope analyses. Additionally, metal stable isotopes can

be measured by in situ techniques, including secondary ion MS (SIMS) and laser ablation MC-ICP-MS, which has shown unique potential in unravelling micron-scale information from samples with complex textures or zonation that are otherwise inaccessible by bulk analysis. Currently, in situ stable isotope analysis is more frequently used in studies of high-temperature and cosmogenic processes, as well as environmental conditions of early Earth. This leaves ample opportunities to adapt existing in situ methodologies and develop new ones for more recent paleoceanographic research.


**Table 6.2.1: Quantitative analytical methods for trace metals and metal isotopes.**

| Analysis | Digestion method | Instrument | Quality control |
|---|---|---|---|
| Bulk major elements | Full digestion (alkaline fusion or acid digestion) | ICP-OES (rapid and cost efficient), solution-based or laser ablation (for $LiBO_2$ fused beads) ICP-MS (low detection limit), XRF, AAS | Instrumental drift correction (e.g., internal standards) and standard reference materials |
| Bulk minor-trace elements | Full digestion (alkaline fusion or acid digestion) | Usually ICP-MS, solution-based or laser ablation of $LiBO_2$ fused beads | Instrumental drift correction (e.g., internal standards) and standard reference materials |
| Stable metal isotopes | Leaching authigenic phases | TIMS, (laser ablation) MC-ICP-MS, and in situ SIMS | Double spike and sample-standard bracketing |



### 6.2.3 Redox-sensitive metal and metal isotope proxies

### 6.2.3.1 Redox-sensitive metal proxies with valence state changes by redox potential

**6.2.3.1.1 Manganese**

Manganese (Mn) has three oxidation states (II, III, and IV). The reduced forms of Mn are soluble in low-oxygen waters (< 10 $\mu mol$ $O_2$) (Madison et al., 2013; Oldham et al., 2017), which include Mn(II) and soluble Mn(III) complexed by inorganic or organic ligands (Mn(III)-L) (Oldham et al., 2015). The oxidized form of Mn(IV) forms solid Mn(IV) oxides. Consequently, the residence time of dissolved Mn in the oxygenated deep ocean is on the order of 10-40 years (Bender et. al., 1977;

Klinkhammer & Bender, 1980; Hayes et al., 2018). As reduced Mn(II) can be oxidized to Mn(III)/Mn(IV) oxyhydroxides with even micromolar levels of oxygen concentrations (Tebo et al., 2004; Morgan, 2005; Clement et al., 2009), sedimentary Mn enrichment can be used as an oxic indicator in pore waters (Burdige & Gieskes, 1983; Froelich et al., 1979; Calvert & Pedersen, 1996). However, free Mn(II) can also precipitate as $MnCO_3$ and/or co-precipitate with authigenic calcite in reducing pore waters with high alkalinity for example when methanogenesis occurs (Calvert and Pedersen, 1996; Mucci, 2004), which may

lead to a false positive for oxic conditions. Thus, Mn should be evaluated simultaneously with other redox-sensitive metals.

### 6.2.3.1.2 Iron

In oxic environments, Fe exists as Fe (oxyhydr)oxides, including ferrihydrite, lepidocrocite (γ-FeOOH), goethite (α-FeOOH) (Poulton and Raiswell, 2005; van der Zee et al., 2003), hematite (α-$Fe_2O_3$), maghemite (γ-$Fe_2O_3$), and magnetite ($Fe_3O_4$) (Berner, 1981; Schwertmann, 2008). As the ambient water becomes depleted in oxygen, Fe (oxyhydr)oxides can be reduced to Fe(II).

With sulphide production during sulphate reduction, reduced Fe(II) can be converted to Fe sulphides that include mackinawite (FeS), greigite ($Fe_3S_4$), and pyrite ($FeS_2$) (Poulton et al., 2004; Raiswell and Canfield, 2012; Roberts, 2015). In strongly reducing waters (e.g., methanic conditions), siderite ($FeCO_3$) can also form (Berner, 1981; Roberts, 2015). Combined with Fe sulphides, carbonate-bearing Fe and Fe (oxyhydr)oxides are considered as the highly reactive Fe pool, because these forms of Fe readily react with free sulphide (e.g., $HS^-$) in early diagenetic stages (März et al., 2008a; Poulton and Raiswell, 2005;

Raiswell and Canfield, 2012). By leaching out different Fe phases, Fe speciation uses highly reactive Fe (FeHR) / total Fe (FeT), and pyrite Fe (Fepy) / FeHR to distinguish oxic, ferruginous, and sulphidic conditions (e.g., Clarkson et al., 2014; Lyons and Severmann, 2006; Poulton and Canfield, 2011; Poulton and Raiswell, 2005). Modern sediment calibrations indicate a threshold of FeHR/FeT >0.38 for anoxic water columns (Poulton and Canfield, 2011; Raiswell and Canfield, 1998). Under anoxic regimes (FeHR/FeT>0.38), Fepy/FeHR has been used to differentiate sulphidic (Fepy/FeHR>0.7~0.8) from ferruginous

(Fepy/FeHR<0.7) waters (Clarkson et al., 2014; Li et al., 2015; März et al., 2008b; Poulton and Canfield, 2011; Poulton et al., 2015). When FeHR/FeT<0.38, high Fepy/FeHR values (>0.8) have also been used to indicate oxic water columns with pore water sulphide accumulation in organic carbon-rich sediments (Canfield et al. 1992; Hardisty et al. 2018; Raiswell and Canfield, 1998; Raiswell et al. 2018).





### 6.2.3.1.3 Uranium, rhenium, and vanadium

Uranium (U), rhenium (Re), and vanadium (V) behave conservatively in seawater – the residence time in the ocean is ~750,000 years for Re (Akintomide et al., 2021), ~300,000-600,000 years for U (Dunk et al., 2002; Ku et al., 1977; McManus et al., 2005; Morford and Emerson, 1999, Lau, et al., 2019), and ~50,000-100,000 years for V (Shiller and Boyle, 1987; Tribovillard et al., 2006, Nielsen 2021). As a result, sedimentary concentration changes of U, Re, and V on time scales shorter than tens of thousands of years are likely not a response to the changes in the dissolved concentration in the overlying water column.

Instead, the downward flux of metal reduction, in accordance with the redox potential of the pore water, is likely the driver of the sedimentary variations (Böning et al., 2004; Colodner et al., 1995; Sundby et al., 2004).

Rhenium exists as $ReO_4^-$ in oxic waters, but can be reduced to Re(IV) oxides (e.g., $ReO_2$) in reducing environments. Redox potential of the Re(VII)/Re(IV) couple is higher than that of U(VI)/U(IV), situated between $MnO_2$/Mn(II) (manganous) and

$Fe^{3+}$/$Fe^{2+}$ (ferruginous) and is similar to the redox potential of $NO_3^-$/$NO_2^-$ (Bratsch 1989, Algeo & Li, 2020). Re preservation in sediment could also be associated with thiolation of $ReO_4^-$ to particle-reactive $ReO_nS_{4-n}^-$ which enhances its particle reactivity towards iron sulphides (Akintomide et al., 2021) and/or co-precipitation with the Fe-Mo-S phase in sulphidic (no oxygen detected with sulphide present) waters (Helz and Dolor, 2012; Helz, 2022). Free sulphide levels in the most sulphidic water columns are still insufficient to support thiolated $ReO_4^-$ as major species due to higher required sulphide levels relative to

molybdate thiolation (Helz and Dolor, 2012; Vorlicek et al., 2015). However, this potential exists in euxinic pore waters (Akintomide et al., 2021).

Vanadium mainly occurs as V(V) in oxic waters in the form of vanadate (e.g., $HVO_4^{2-}$ and $H_2VO_4^-$). However, unlike U and Re, vanadate can be scavenged by adsorption onto Fe-Mn (oxyhydr)oxides and clay minerals (e.g., Wehrli and Stumm 1989;

Morford and Emerson, 1999). The redox potential of the V(V)/V(IV) is similar to that of the Re(VII)/Re(IV) couple (Algeo and Li, 2020). Thus, as oxygen draws down, vanadate can be reduced to the V(IV) species (vanadyl, $VO^{2+}$ and $VO(OH)^{3+}$) by organic compounds, which can co-precipitate/complex with mineral particles and organic matter (Emerson & Huested, 1991; Algeo and Maynard, 2004). Under more reducing conditions (e.g., sulphidic), vanadyl might be further reduced to the V(III) species by free sulphide in the ambient waters, which precipitate as solid oxides ($V_2O_3$) or hydroxides (VOOH) (Wanty and

Goldhaber, 1992). Despite the different authigenic enrichment mechanisms, V reduction and sequestration into sediments still begin under low-oxygen conditions, making it a tracer of such conditions.

In oxic water columns, U exists as the soluble U(VI) and binds to carbonate ions forming $Ca_2UO_2(CO_3)_3$ (Endrizzi & Rao, 2014, Langmuir, 1978). Redox potential of the U(VI)/U(IV) couple is below that of the $Fe^{3+}$/$Fe^{2+}$ couple but above $SO_4^{2-}$/$H_2S$

(Fig. 1, Morford and Emerson, 1999; Zheng et al., 2002). In reducing environments, U(VI) turns into U(IV) in the form of the solid uraninite ($UO_2$) or adsorbs onto sediment solids, which may involve biologically mediated processes (Crusius et al.,



1996; Klinkhammer and Palmer, 1991; McManus et al., 2005; Zheng et al., 2002; Lovley et al., 1991; McManus et al., 2006; Stirling et al., 2015; Rolison et al., 2017).

**6.2.3.1.4 Molybdenum**

In oxic waters, Mo primarily exists as soluble molybdate ($Mo(VI)O_4^{2-}$) and behaves conservatively (~440,000 years residence time) (Emerson and Huested, 1991; Bostick et al., 2003; Miller et al., 2011). However, ~30–50% of molybdate (Kendall et al., 2017) may be sequestered through adsorption onto Mn and Fe (oxyhydr)oxides (Zheng et al., 2000; Tribovillard 2006) in oxic waters. Unlike the low-oxygen indicators (e.g., U, Re, and V), reductive Mo removal requires sulphidic conditions that

lead to progressive thiolation of molybdate (thiomolybdate series $Mo(VI) O_xS_{4-x}^{2-}$, x=0~4) (Erickson and Helz, 2000; Helz et al., 1996; Hlohowskyj et al., 2021; Tribovillard et al., 2006; Vorlicek et al., 2004;). Thiomolybdates are particle reactive and are readily scavenged into sediments and onto iron sulphides (Helz et al., 1996; Helz et al., 2004; Freund et al., 2016). Mo removal from sulphidic pore waters had been associated with a 'geochemical switch' linked to a $[H_2S]$ threshold of > 11 μmol (when $MoS_4^{2-}$ starts to dominate in the waters, Helz et al., 1996). Yet, recent studies have suggested that under weakly sulphidic

conditions ($[H_2S]$ < 11 μmol), intermediate thiomolybdate species could be the dominant Mo species in the water column that contribute to Mo sequestration (e.g., Vorlicek et al., 2004; Scholz et al., 2018; Tessin et al., 2019). Multiple pathways have been proposed for Mo removal from sulphidic waters, including: (1) the organic matter (OM) pathway that leads to Mo (IV or VI)-OM complexes (Chen et al., 1998; Wichard et al., 2009; Dahl et al., 2010b; Dahl et al., 2017); (2) the Fe-sulphide pathway that has thiolated Mo adsorption to iron sulphide phases with subsequent Mo(VI) reduction to Mo(IV) (Miller et al., 2020)

and/or that incorporates Mo(IV) into Mo-Fe-S structures such as $FeMoS_2(S_2)$ (Helz et al., 1996; Bostick et al., 2003; Vorlicek et al., 2004; Helz et al., 2011;Vorlicek et al., 2018; Helz and Vorlicek, 2019); and (3) the biological pathway that implies biological uptake (e.g., by sulphate reduction bacteria) and Mo reduction by enzymes (e.g., Orberger et al., 2007; Biswas et al., 2009; Dahl et al., 2017). Authigenic Mo enrichment has thus been interpreted as an indicator of sulphidic environments provided that coeval enrichments of other redox-sensitive trace metals (e.g., U or Re that are not scavenged by Mn oxides in

oxic environments) are observed (Calvert and Pedersen, 2007; McKay et al., 2007; Scott et al., 2008; Tribovillard et al., 2006).

**6.2.3.2 race metal proxies with speciation changes in sulphidic waters (cadmium, zinc, and nickel)**

Unlike the previously discussed trace metals, Cd, Zn, and Ni behave like micronutrients, with a depletion in the surface ocean due to phytoplankton uptake and increasing concentrations at depth due to decomposition of organic carbon (Bruland, 1983; Flegal, Sanudo-Wilhelmy, & Sceflo, 1995; Nozaki, 1997; Bruland & Lohan, 2003). Authigenic metal enrichments in the

sediments primarily occur in sulphidic waters because they can either form insoluble sulphides (e.g., Cd and Zn; Rosenthal et al., 1995; Tribovillard et al., 2006; Little et al., 2015) and/or can be incorporated into the pyrite structure (e.g., Ni, Huerta-Diaz and Morse, 1992; Large et al., 2014), making them a possible tracer of sulphidic conditions. However, we note that Ni has also





been linked to sinking fluxes of organic material in upwelling regions because of its close link with productivity and less diagenetic alteration associated with sedimentary sulphur and manganese cycling (Böning et al., 2015).

### 6.2.3.3 Redox-sensitive element isotope systems

Limited research on redox-sensitive metal isotope systematics in modern ocean systems hampers interpretations of many "non-traditional" isotope signals in sediments. Nonetheless, the redox cycling of Mo and U isotopes ($^{98}$Mo/$^{95}$Mo and $^{238}$U/$^{235}$U, respectively, expressed as $\delta^{98}$Mo and $\delta^{238}$U) in modern systems are relatively well understood (Andersen et al., 2017; Brüske et al., 2020; Bura-Nakić et al., 2018; Kendall et al., 2017; Noordmann et al., 2015). Sedimentary (co-)variations in $\delta^{98}$Mo and $\delta^{238}$U have been widely applied to trace past changes in ocean oxygenation, especially on orbital to million year timescales (e.g., Andersen et al., 2020; Chen et al., 2015; Chiu et al., 2022; Clarkson et al., 2021; Dahl et al., 2010; Dickson, 2017; Gordon et al., 2009; Hardisty et al., 2021; Kendall et al., 2015; Sweere et al., 2021; Wang et al., 2016; Zhang et al., 2018). Less is known about Re isotopes and their usefulness for constraining past changes in ocean oxygenation although early studies are working to constrain the Re isotope mass balance (Dellinger et al., 2021).

### 6.2.3.3.1 Uranium isotopes

As discussed in Section *6.2.3.1.3*, soluble U(VI) is reduced to insoluble U(IV) in low-oxygen environments. The sedimentary U sequestration process also introduces significant isotopic fractionation (e.g., Andersen et al., 2017; Brown et al., 2018; Lau et al., 2019; Stirling et al., 2015; Zhang et al., 2020), as the nuclear volume effect causes a preferential removal of the heavy $^{238}$U relative to the lighter $^{235}$U isotope. Because of the long residence times of U (~3-6 x 10$^5$ years; Dunk et al., 2002), the isotopic composition of U in seawater is globally homogenous (-0.39‰ in the modern ocean; Andersen et al., 2017). Uranium uptake in reducing sediments is the primary U sink in the global ocean, and, hence, seawater $\delta^{238}$U changes can be associated with the extent of sea floor anoxia (e.g., Andersen et al., 2017; Lau et al., 2019; 2020).

Oxic sediment deposits that record the seawater U isotope value (e.g., shallow marine carbonates and Mn oxide crusts) have been used to infer the areal extent of anoxic sinks in the global ocean using isotope mass-balance models (e.g., Jost et al., 2017; Clarkson et al., 2018; 2021a; Romaniello et al., 2013; Wang et al., 2016; Zhang et al., 2018; 2020). However, post-depositional diagenesis of carbonate could result in much larger offsets from the seawater U isotope value (e.g., Chen et al., 2022). In contrast, sediments deposited within anoxic conditions, such as organic-rich black shales, will typically record enriched $\delta^{238}$U values during more intense anoxia, although the true expression of isotope enrichment is complicated by processes that may vary with depositional environments (e.g., diffusion of U between the sediment-water interface and the zone of U reduction within the sediment; see Andersen et al., 2014). However, long-lasting anoxia/euxinia within restricted basins (e.g., limited water renewal) can deplete the dissolved U pool, leading to limited fractionation. This results in the bulk sediment $\delta^{238}$U values approaching the seawater value, albeit with a heavier signature (~+0.6‰) due to diffusion limitations (Andersen et al., 2014; Brüske et al., 2020; Chiu et al., 2022; Lau et al., 2020; Rolison et al., 2017; Romaniello et al., 2013). Within anoxic facies,





carbonate-associated-uranium isotopes have also been used to infer local deoxygenation in sediments, with the large advantage that this proxy is not significantly impacted by post-depositional oxidation (Clarkson et al., 2021b).

**6.2.3.3.2 Molybdenum isotopes**

In sulphidic environments, Mo in oxic waters (mostly as $MoO_4^{2-}$) is converted into particle-reactive thiomolybdate species in the presence of free sulphide (see Section *6.2.3.1.4*). During this transformation the Mo isotopes are fractionated, with the
more sulphidised thiomolybdate species becoming isotopically lighter relative to seawater (He et al., 2022; Kerl et al., 2017; Tossell, 2005). The Mo isotopic composition of mildly euxinic sediments is, thus, expected to be lighter than the seawater value (~2.34‰ in the modern ocean; Nakagawa et al., 2012; Nägler et al., 2013). Low $\delta^{98}Mo$ values of sediments deposited in such environments can be modelled assuming higher scavenging rates for the more sulphidised Mo species, allowing semi-quantitative reconstructions of $H_2S$ concentrations (Dahl et al., 2010; Matthews et al., 2017; Sweere et al., 2021). However, in
strongly sulphidic conditions (e.g., the Black Sea), the conversion of $MoO_4^{2-}$ to $MoS_4^{2-}$ (the most sulphidised species) is near-complete (Erickson and Helz, 2000), such that little-to-no fractionation is expressed (Tossell, 2005). The sedimentary $\delta^{98}Mo$ values therefore approach seawater compositions (Neubert et al., 2008; Brüske et al., 2020).

In conclusion, sedimentary Mo isotopes can be used to trace the local euxinic water-column conditions when the global
seawater Mo isotopic composition is known/close to modern (e.g., Holocene/Pleistocene to Paleocene sediments, Andersen et al., 2018; Azrieli-Tal et al., 2014, Hardisty et al., 2021; Matthews et al., 2017, Sweere et al., 2021, Riedinger et al., 2021). Additionally, environments with quantitative drawdown of dissolved Mo may also be studied to infer global seawater $\delta^{98}Mo$ values to estimate global-scale extends of oxic-anoxic-euxinic Mo sinks on geological timescales (e.g., Dahl et al., 2021; Dickson and Cohen, 2012; Dickson et al., 2016).

**6.2.4 Organic carbon and trace element burial**

Because redox potentials vary as a function of both reductant and oxidant availability, enrichments of trace elements in sediments could result from bottom water oxygen availability and/or the rain rate of organic carbon. For example, glacial authigenic U enrichments in the Southern Ocean have been found to occur primarily as a function of changes in export production (e.g., Chase et al., 2001; Kumar et al., 1995). As some redox-sensitive metals (e.g., Mo, V, Ni, and Cd) can be
concentrated in phytoplankton due to biological uptake, they are also efficiently transported to the sediments via bio-detritus and particulate organic carbon (e.g., in upwelling areas) (Böning et al., 2004; Muñoz et al., 2012; Valdés et al., 2014; Castillo et al., 2019; Nameroff et al., 2004; Muñoz et al., 2023). Thus, to isolate bottom water oxygen concentrations, reconstructions using redox-sensitive trace elements must be accompanied by independent constraints on the supply of particulate organic carbon (e.g., Anderson et al., 2019; Bradtmiller et al., 2010; Jaccard et al., 2009; Jacobel et al., 2020; Pavia et al., 2021). It is
particularly important that the proxy for organic carbon flux is independent from the influence of oxygen, rendering classic proxies like total organic carbon (TOC) ineffective since its sedimentary abundance is itself a function of oxic respiration (e.g.,



Burdige, 2006; Tyson, 2020). For a description of marine organic compounds that are susceptible to oxygen see Section 6.3 on organic biomarkers as proxies for bottom water oxygen.

In the last few decades, a significant body of work has focused on the use of excess barium ($Ba_{xs}$) as a semi-quantitative proxy for organic carbon rain (e.g., Dymond et al., 1992; Paytan et al., 1996, Ganeshram et al., 2003, Martinez-Ruiz et al., 2019; Carter et al., 2020). $Ba_{xs}$ is the concentration of barite, in excess of the lithogenic concentrations, precipitated from supersaturated microenvironments formed within organic aggregates settling through the water column (Carter et al., 2020). Although regional relationships between $Ba_{xs}$ and organic carbon are quite strong (Costa et al., 2023), the correlation is poor

on a global scale (Costa et al., 2023 and references therein) likely due, at least in part, to basinal variations in the surface concentration of dissolved Ba and relative differences in undersaturation within the water column. However, care should be taken to avoid the use of $Ba_{xs}$ in intervals where sulphate reduction may have caused the diagenetic remineralization of barite, especially in continental margin sediments with sulphidic pore waters (Carter et al., 2020, Griffith & Paytan, 2012; Hendy, 2010; McManus et al., 1998; Torres et al., 1996; Von Breymann et al., 1992).


Because trace element enrichment proceeds according to sequential redox thresholds, attempts have been made to define bottom water oxygen 'thresholds' below which redox-sensitive trace elements would be expected to become enriched in the sediments (e.g., Algeo & Li, 2020; Bennett & Canfield, 2020). Unfortunately, this approach is inappropriate for reconstructing bottom water oxygen concentrations because variations in organic carbon can be the primary determinant of trace element

enrichment.

### 6.2.5 Other factors controlling trace metal preservation/metal isotope fractionation

In addition to redox potential and organic carbon rain rate, sedimentary enrichments of trace metals can be altered by a number of processes, which may affect metal delivery to the sediments and result in metal remobilization/recycling in the sediment pore waters. These caveats may lead to decoupled sedimentary responses from the water column oxygen variability.

### 6.2.5.1 Detrital influences on authigenic enrichments of trace metals

Shifts in sedimentary elemental compositions may be associated with changing proportions of sediment sources (e.g., lithogenous, biogenous, and hydrogenous) with inherently different elemental matrices. For example, nearshore sediments are highly influenced by terrestrial inputs (e.g., fluvial and aeolian sediments) and organic fluxes from primary productivity, whereas deep-sea sediments generally only receive the finest fraction of lithogenic particles (e.g., clays from dust). Estimates

of authigenic concentrations are based on an assessment of the lithogenic contribution (detrital) using Al as the most common approximation. It is based on the conservative behaviour of Al during weathering and soil formation, and the assumption that Al concentrations are very similar in most common sedimentary rocks (Calvert and Pedersen, 2007), which may not hold true on a global scale. Titanium has also been used for normalization, but the concentration of this element is more variable than



Al in different rock types (Calvert and Pedersen, 2007). Additionally, the estimate of detrital contributions assumes that the
detrital element (e.g., Al) analysed is only in the aluminosilicate fraction. Therefore, estimations attributed to other phases
(e.g., hydrogenous) could be underestimated (Van der Weijden, 2002). Caution should be employed when correlating
normalized data because it could modify the original correlations between elements (Cole et al., 2017).

### 6.2.5.2 Post-depositional diagenetic effects on sedimentary trace metals

As with many other proxies, the utility of redox-sensitive metals as a paleo-oxygen indicator relies on the post-depositional
persistence, or preservation, of the initial redox signal. Modifications to trace element distributions during diagenesis include:
wholescale overprinting of sedimentary redox signals (e.g., Jacobel et al., 2020; Zheng et al., 2002), partial removal (e.g.,
Bonatti et al., 1971; Chase et al., 2001; Hayes et al., 2014; Jacobel et al., 2017; Jung et al., 1997; Morford et al., 2009), and
oxidation and down-core precipitation along concentration gradients (Colley et al., 1989; Colley and Thomson, 1985; Jacobel
et al., 2017; Olson et al., 2017, McKay et al., 2014, Anschutz et al., 2002, Deflandre et al., 2002).


One of the most significant pore water alterations that can modify the original sedimentary metal enrichment signal is post-
depositional organic matter remineralization, which progressively consumes pore water oxygen and changes local redox
potential (Morford and Emerson, 1999; Nameroff et al., 2002) (Fig. 1). As oxygen is depleted, Fe/Mn (oxyhydr)oxides are
reduced to release Mn(II) and Fe(II) that diffuse into the pore waters. Aqueous Mn(II) may diffuse upwards until it reaches
oxygenated pore waters and can re-precipitate as Mn oxides (Lynn and Bonatti, 1965; Burdige & Gieskes 1983). As a result,
a post-depositional Mn spike may occur right above the depth where the pore water oxygen concentration goes to zero (Burdige
& Gieskes, 1983; Froelich et al., 1979). Preservation of this peak is affected by subsequent variability in the oxygen penetration
depth (e.g., Finney et al., 1998; Mangini et al., 2001, Anschutz et al., 2002, Deflandre et al., 2002). Shoaling of the oxygen
penetration depth would push the Mn peak into reducing pore waters, in which it will dissolve and diffuse upwards, leaving
no trace of the former peak (Froelich et al., 1979). In contrast, if the oxygen penetration depth increases in the sediment, the
Mn peak will be preserved because it will remain within the oxygen-rich zone (e.g., Froelich et al., 1979; Mangini et al., 1990,
Deflandre et al., 2002). Therefore, it has been proposed that peaks in authigenic Mn concentrations in sediments are best
interpreted for pore water oxygen concentrations that are increasing over time rather than indicators for (static) high pore water
oxygen concentrations (e.g., Volz et al., 2020; Pavia et al., 2021).


Reductive dissolution of Fe/Mn (oxyhydr)oxides may lead to additional metal release as they are carrier phases for many trace
metals (Algeo & Tribovillard, 2009; Scholz et al., 2011, 2017). For instance, remobilized V due to Mn oxide reductive
dissolution (Seralathan and Hartmann, 1986; Legeleux et al., 1994; Hastings et al., 1996) may either diffuse upward into
bottom waters (Heggie et al., 1986; Shaw et al., 1990) or diffuse downwards and re-precipitate at a deeper sediment depth
(Colley et al., 1984; Jarvis and Higgs, 1987). Post-depositional buildup of reducing conditions (e.g., sulphate reduction) would





also facilitate additional trace metal sequestration (e.g., Mo) by sedimentary organic carbon and/or sulphides (e.g., pyrite or other metal sulphides) (Al-Farawati & van den Berg, 1999; Erickson & Helz, 2000; Helz et al., 1996; Helz & Vorlicek, 2019).

Remobilization of authigenic U has also been studied extensively in regions characterized by large oscillations in pore water redox potential (e.g., Morford et al., 2009). When pore water oxygen increases, U remobilization may occur and allow U diffusion to the overlying bottom water and/or re-precipitation at a deeper depth (Fig. 4).

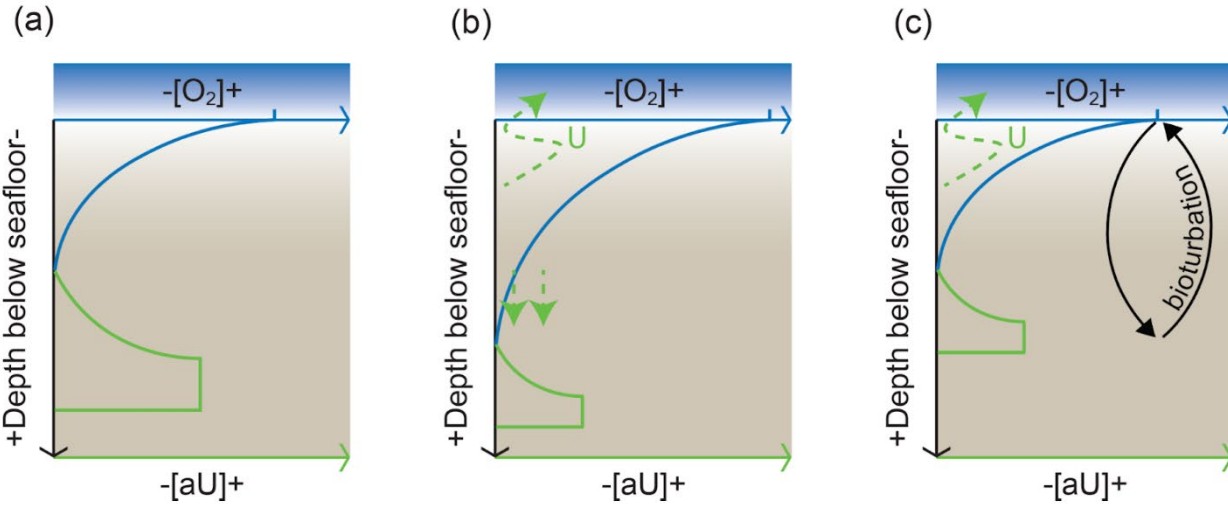

**Figure 4: Schematic of authigenic U (aU) post-depositional diagenesis, based on Jacobel et al. (2020). A) In a baseline scenario, the**
**relatively low-oxygen concentration at a certain depth below seafloor (blue line) leads to aU precipitation (green line). B) As the bottom water oxygen concentration and the depth of oxygen penetration both increase, a portion of the previously precipitated aU becomes remobilized and diffuses upwards (green arrow). The rest of the remobilized U re-precipitates downcore. C) Bioturbation (black arrows) mixes aU-containing sediment upward and exposes it to a better-oxygenated environment, where aU is remobilized and released into the bottom water (Morford et al., 2009).**


These observations have several important implications. An absence of redox-sensitive metal enrichment cannot be taken as evidence that such conditions were absent, as short-lived events may not be recorded. This is especially true in low accumulation rate environments with less than 2 (Jung et al., 1997; Mangini et al., 2001) or 3 cm kyr$^{-1}$ (Jacobel et al., 2020),
where pore waters may retain active redox fronts long after the time of initial deposition, especially if sedimentary organic
carbon is low. Caution is also needed in interpreting the shape of sedimentary enrichment features as primary signals and both the sharpness of peaks and their temporal structure (Crusius and Thomson, 2000; Jacobel et al., 2020, 2017; Thomson et al., 1996) may be modified post-depositional.





### 6.2.5.3 Sedimentation rate changes

The impact of sedimentation rate on authigenic enrichment should also be considered when evaluating metal accumulation.
Changes in sedimentation rate would be expected to impact accumulation since sedimentation rate directly influences the rate of organic carbon respiration and depth of bioturbation. Sedimentation rate has been used as a proxy for the flux of organic carbon to the sediment-water interface, with higher rates associated with more reducing conditions and a shoaling of the oxygen penetration depth (Boudreau, 1994; Tromp et al., 1995). A shallower oxygen penetration depth would reduce pore water exposure to oxygen and allow a better preservation of trace metals (Fig. 5). A special case is the occurrences of instantaneous
depositional events (e.g., turbidite layers), which could introduce pulses of sediment delivery that significantly reduce oxygen exposure of the underlying sediments. Rapid sediment accumulation would then facilitate buildup of reducing pore waters that lead to post-depositional diagenesis (e.g., Fe/Mn reduction) (Anschutz et al., 2003, McKay et al., 2014, Wang et al., 2019). However, an increase in the rate of nonreactive sediment can also dilute the relative concentrations of organic matter and trace metals while reducing the downward diffusion of dissolved gasses (oxygen) or aqueous species (trace metals, sulphate).
Modelled authigenic Mo and U as a function of sedimentation rate show dramatic decreases in their authigenic concentration with increasing sedimentation rate (Liu and Algeo, 2020; Hardisty et al., 2018; Morford et al., 2007).

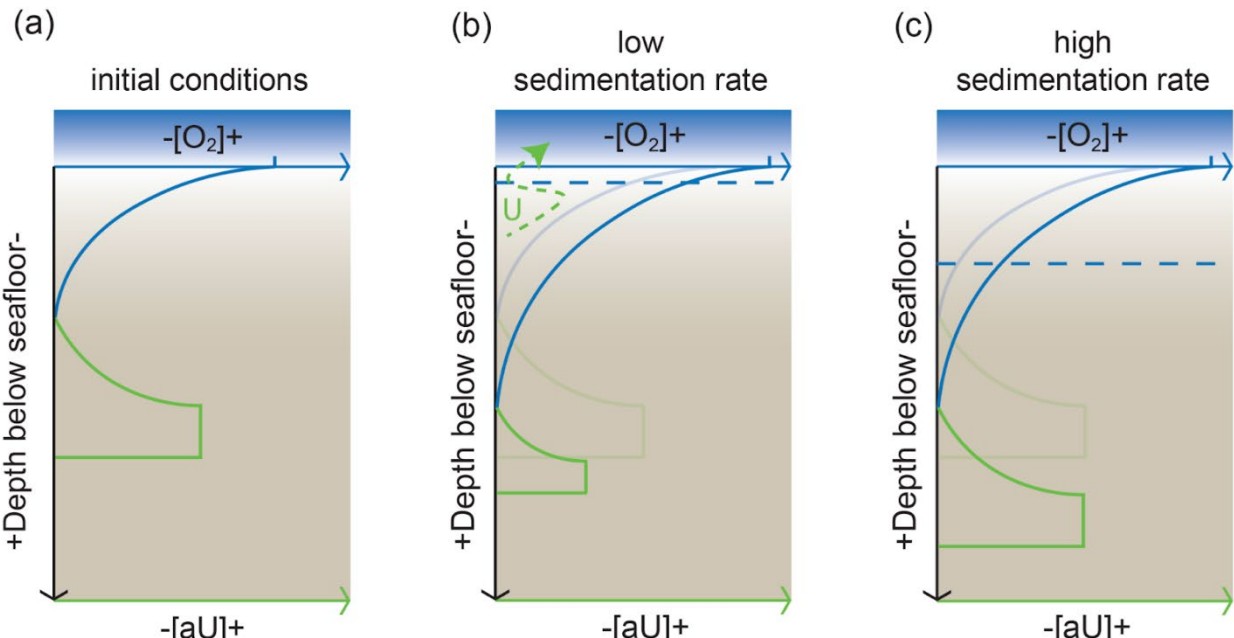

**Figure 5: Schematic of aU preservation in sediments with varying sedimentation rates, inspired by Costa et al (2018). A) In a baseline scenario, the relatively low-oxygen concentration at a certain depth below seafloor (blue line) leads to aU precipitation (green line).**
**B) When sedimentation rates are low (dashed blue line is the original seafloor position), as the bottom water oxygen concentration increases, aU remobilization takes place and preservation is poor. C) When sedimentation rates are high, aU is insulated from the oxygen penetration down the sediments and the aU signal is better preserved.**



**6.2.5.4 Particulate shuttle and basin effect on trace metal delivery to the sediments**

In many cases, trace metal enrichments may be controlled by a 'particulate shuttle' (e.g., particulate Fe/Mn (oxyhydr)oxides
and phytoplankton remains, Zheng et al., 2002; Algeo and Lyons, 2006, 2009; Tribovillard et al., 2012; Sweere et al., 2016;
Scholz et al., 2017; Ho et al., 2018; Severmann et al., 2008; Muñoz et al., 2023) and the resupply of certain trace metals ('basin
reservoir effect'; e.g., Algeo and Lyons et al., 2006). Due to affinity of Mo on Fe/Mn (oxyhydr)oxides, molybdate adsorbs
onto particulate Fe/Mn oxyhydroxides in the oxic waters while being transferred through the water column. These particles
are then reduced when oxygen is depleted in the ambient waters releasing molybdate that either diffuses back into the water
column or is scavenged in the sulphidic sediments (Morford and Emerson, 1999; Morford et al., 2005). The latter process
accelerates the transfer of authigenic Mo to the sediment relative to other redox-sensitive trace metals (e.g., U) that are not
affected by the particulate shuttle, leading to elevated authigenic Mo/U ratios in the sediments (e.g., Cariaco Basin, Algeo and
Tribovillard, 2009). Operation of particulate Fe-Mn (oxyhydr)oxide shuttles occurring close to a (variable) redoxcline in the
water column could thus be interpreted from Mo-U covariation in the sediments.


On the other hand, dissolved trace element supply is limited in hydrographically restricted basins compared to the open ocean.
Subsequent scavenging of certain trace metals may deplete the dissolved metal reservoir in the water column. Consequently,
trace metal enrichment can vary considerably in restricted ocean areas compared to an open ocean setting (Algeo and Lyons,
2006; Algeo and Rowe, 2012; Sweere et al., 2016; Algeo and Li, 2020; Bennett and Canfield, 2020). For instance, dissolved
Mo/U in the Black Sea is lower than that in the open ocean due to continuous scavenging of Mo in the water column, leading
to lower authigenic (Mo/U) ratio (Algeo and Tribovillard, 2009). Redox-sensitive enrichments may thus vary significantly in
different depositional systems depending on metal delivery from the water column. The relationship between Mo and U
enrichment factors ($Mo_{EF}$ and $U_{EF}$) to establish the shuttle effect and to infer the oxygenation conditions of the depositional
environment can be found from the model proposed by Algeo & Tribovillard (2009) and Tribovillard et al., (2012).
It is necessary to take site-specific impacts and changes in environmental variables through time, such as organic carbon rain
rate, into account when linking trace metal enrichments to redox conditions for each individual depositional system (Algeo
and Li, 2020).

**6.2.5.5 Interpretive approaches for reducing uncertainty**

Several studies have applied the use of a suite of elements, including ratios and corrections for detrital phases, to compensate
for some of the issues mentioned above, exploiting the variable response of different elements to the array of controlling
parameters (Algeo & Lyons, 2006; Crusius et al., 1996; Jones & Manning, 1994). A recent review finds some redox
dependency for all studied trace element ratios, but also stresses several complications, including the need for these proxies to
be carefully calibrated for each individual setting (Algeo & Liu, 2020).



With an understanding of individual preservation mechanisms, the use of multiple redox proxies leads to a more nuanced interpretation of past conditions. Chromium, Re and Mo have been used to discern the global extent of anoxic and sulphidic conditions, respectively (Reinhard et al., 2013). Molybdenum and U have similarly been used to distinguish between sulphidic or non-euxinic conditions and changes over time while also providing evidence for water mass restriction (Zhang et al., 2022; Algeo and Tribovillard, 2009). Although it is possible that Re and Mo co-precipitate, varying Re/Mo ratios might provide

evidence for dissolution of other carriers thereby increasing the delivery of Re or Mo to pore waters (Helz, 2022). For instance, in the Humboldt upwelling ecosystem (Northern Chile and central Peru) Mo/U and Re/Mo ratios have been used to differentiate suboxic (low-oxygen and non-euxinic) from anoxic conditions in the depositional environment (Salvatteci et al., 2014, 2016; Valdés et al., 2014, 2021; Castillo et al., 2017). Higher Re/Mo and lower Mo/U than the seawater value (Crusius et al., 1996) would reflect suboxic (low-oxygen) conditions in the absence of $H_2S$ (Tribovillard et al., 2006, Algeo and Tribovillard, 2009).

On the contrary, in the presence of reducing and occasionally sulphidic bottom waters, Mo accumulation increases relative to U, implying that the Mo/U ratio in sediments could be equal to or higher than that of the water column (molar ratio of $7.53 \pm 0.25$, Millero, 1996). However, the fidelity of these interpretations hinges on a clear understanding of differing authigenic preservation mechanisms, redox or non-redox related (e.g., organic carbon flux, particulate shuttle and basin reservoir effect), and the potential for diagenetic loss that would compromise the record.


Other empirical geochemistry proxies have also been proposed to evaluate depositional settings (Algeo & Liu, 2020 and references therein) (e.g., restricted basins vs. continental margins with intensive upwelling). For instance, a decrease in sedimentary Mo/TOC ratios has been associated with water mass restriction in anoxic marine environments (e.g., in a silled basin) based on the observation that water column Mo can be depleted in stagnant basins (Algeo and Lyons, 2006). Based on

the close association of Cd and productivity (Section 6.2.3.2, Horner et al., 2021), elevated Cd/Mo ratios may be used to indicate upwelling zones on the continental margin. Low sedimentary Cd/Mo (close to seawater value) caused by metal depletion in sulphidic water columns, along with high Co/Mn values attributed to a dominated river supply over a deep metal source (e.g., from upwelling) have been used to indicate restricted basin environments (Sweere et al., 2016). We caution that these empirical relationships require a better mechanistic understanding for trace metal cycling and can only be used in the

marine environments where they have been calibrated.

### 6.2.6 Future perspectives

### 6.2.6.1 Towards more quantitative oxygen proxies on a local scale

A few recent studies have investigated the potential for using redox-sensitive trace metals in quantitative oxygen reconstructions, especially on a local scale where other contributing factors to metal enrichments are less variable (e.g.,

sedimentation rates and lithogenic background). For instance, Costa et al. (2023) develop U/Ba as a local- to regional-specific bottom water oxygen proxy, which explicitly takes organic carbon rain rate into account via normalization with respect to Ba.



Local U/Ba calibrations for the Arabian Sea, Eastern Equatorial Pacific (EEP) and Western Equatorial Pacific suggest that U/Ba may be used to capture bottom water oxygen concentrations in regions with >50 μmol kg$^{-1}$ oxygen and high oxygen variability (several tens of μmol kg$^{-1}$). This shows the potential for redox-sensitive trace element concentrations to be quantitatively related to bottom water oxygen when the flux of organic carbon is accounted for.

Due to anthropogenic activities, coastal marine ecosystems are susceptible to pollution by potential contaminant metals and metalloids from industrial, domestic and harbor waste. Some pollutant metals for aquatic life and human health are also redox sensitive, such as Mo, Zn, B, Mn, Ba, Co, Ni, Sr, Cr, Cd, Zr, V, Cu, Ce. In some regions, the concentrations of these elements have recently increased, exceeding the preindustrial values, and the values do not align with increases in lithogenic background inputs (Muñoz et al., 2022; Valdés et al., 2023). As river sediments may concentrate and deliver anthropogenically-sourced metals during transport to the ocean (Pizarro et al., 2010; Yevenes et al., 2018), using standard lithogenic background corrections may overestimate enrichment factors as the authigenic fraction will be augmented with the anthropogenic input. In regions with high anthropogenic input, it is thus recommended to use the local lithogenic background in the region of study (e.g., crust, river, dust, wetland sediment) (Muñoz et al., 2022, 2023; Valdés & Tapia, 2019; Valdés et al., 2023).

### 6.2.6.2 A better understanding of trace metal delivery to the sediments in the GEOTRACES era

Although redox-sensitive trace metals have been widely used as oxygen proxies for the past few decades, there is still a lack of comprehensive research on the behaviour of multiple trace metals in the ocean. Extensive water column analyses on redox-sensitive trace metals and metal isotopes are essential for revealing their global distribution, source and sink fluxes, and preservation mechanisms in the sediments. The GEOTRACES program has provided a unique opportunity for mapping dissolved and particulate trace metal (e.g., Mn) and metal isotope distribution in the modern ocean (Schlitzer et al., 2018), and would allow a direct comparison with core-top trace metal and metal isotope measurements. The new GEOTRACES data would also advance our understanding on multiple trace metal mass balance and potential isotopic fractionation that may occur via incorporation or adsorption. These are critical for improving metal isotope mass balance modelling that has been used in quantitative global oxygen reconstructions (e.g., Lau et al., 2019). However, it is noted that the GEOTRACES program only includes some redox-sensitive metal/metal isotopes (e.g., Fe, Mn, U, and Mo), and these data are not collected on all cruises (Schlitzer et al., 2018). Future coordinated efforts to expand routine analysis to more redox-sensitive trace metals and metal isotopes (e.g., with robust method development and participation of more research groups), as well as *in situ* surface sediment collection (e.g., multi-coring), would significantly advance our proxy development and improve the knowledge of proxy controls and potential caveats.

### 6.2.6.3 Expanding metal isotope applications in the Cenozoic through proxy development

Apart from the U and Mo isotope systems discussed above, many other "non-traditional" isotope systems are being actively explored as important redox tracers.



Due to the very long residence time of some trace metals compared to the seawater (Section 6.2.3), other metal isotope proxies
have been investigated to study ocean oxygen variability on shorter (e.g., orbital) timescales. For instance, chromium (Cr,
residence time of ~10 kyrs, Reinhard et al. 2014) isotopes of sediments deposited under sulphidic water columns (e.g., Cariaco
Basin off the Venezuela coast) may record the seawater value due to quantitative Cr removal from water columns (Gueguen
et al., 2016; Reinhard et al., 2014). However, modern marine sediment Cr isotope data are limited and quantitative Cr removal
from the water columns is only observed on a few occasions (Huang et al., 2021), which also hinders our understanding of Cr
cycling in the water column and sediment. Thus, it is unclear under what conditions sediments faithfully record the seawater
Cr value (see Bauer 2020 for a review). Another promising global oxygen content tracer is thallium (Tl) isotopes, which have
been shown to be primarily controlled by the global Mn oxide burial on timescales of $<10^6$ years (Nielsen et al. 2011, 2019,
Owens et al. 2017). The seawater Tl isotopic composition is homogenous due to its relatively long residence time (~18 kyrs)
(Nielsen et al., 2019, Owens et al., 2017), and significant Tl isotopic fractionation upon oxidative sorption onto Mn oxides
(Nielsen et al., 2013) indicates that even a relatively small change in Mn oxide burial fluxes caused by the global oxygen
content change may result in significant changes in the seawater Tl isotopic value. Quantitative Tl removal has been observed
in reducing pore waters (with Mn reduction, Ahrens et al., 2021) and a recent core-top calibration suggests that authigenic Tl
isotopic compositions can faithfully record the seawater value if pore water is reducing at/near the sediment-water interface
leading to complete Tl sequestration from ambient pore waters (Wang et al., 2022). As criteria for determining the fidelity of
sedimentary Tl isotope records are developed, paleo-reconstructions of seawater Tl isotopic compositions in the future will
ultimately help reveal variations in Tl global ocean content on millennial to orbital timescales, with important implications for
marine carbon storage that may have driven the glacial-interglacial transitions.

In addition to assessment of the global ocean oxygen level, there is demand for local oxygen reconstructions. This task can be
suitably undertaken by proxies with residence times similar to, or shorter than, the average ocean mixing time of ~1,000 yrs.
A promising proxy is cerium (Ce, residence time on the order of ~50–100 years, (Alibo and Nozaki, 1998)) and Ce isotope
ratios. Because Ce has relatively high redox potential, a Ce-based proxy is expected to be one of few redox proxies that can
be useful under dysoxic (low-oxygen) conditions. In the modern ocean, Ce is oxidized to $Ce^{4+}$ and scavenged by Fe/Mn
(oxyhydr)oxides rapidly out of oxic seawater, leading to anomalously low Ce concentration relative to neighbouring rare earth
elements (REE) lanthanum (La) and praseodymium (Pr) (Rudnick and Gao, 2003). However, use of Ce anomalies in sediments
is limited by the fact that the Ce anomaly is dependent on concentrations of its neighbouring REEs. Thus, alteration in
concentrations of the neighbouring REEs may complicate interpretation of Ce anomaly records from marine sediments. Stable
Ce isotopes could provide valuable information supplementing Ce anomalies, while avoiding the dependence of Ce anomaly
on other REEs. Experiments have shown that oxidative adsorption of dissolved Ce onto Mn oxides can produce ~0.5‰
fractionation in $^{142}Ce/^{140}Ce$ with adsorbed Ce being isotopically light, whereas Ce adsorption onto Fe oxides or Ce oxidation
by oxygen produces a smaller Ce isotope fractionation of ~0.2‰ or less (Nakada et al., 2013). This contrasting behaviour in





stable Ce isotope fractionation implies a close link between Ce isotope variations and Mn cycling. However, a modern calibration of the Ce isotope system in marine environments is lacking, and seawater $\delta^{142}$Ce values are not currently known.

Given the many unknowns in the 'non-traditional' stable metal isotope systems, future research that incorporates both seawater and sediment core-top measurements across oxygen gradients is required to establish a robust relationship between metal isotope responses and ambient oxygen. In addition, controlled laboratory experiments to study metal isotope fractionation during scavenging processes (e.g., sorption onto Mn oxides) under various redox conditions are desirable to advance our understanding of fundamental controls on stable metal isotope fractionation.

**6.3 Biomarkers**

**6.3.1 Organic Matter and Lipid Biomarkers**

Organic matter encompasses a wide spectrum of carbon-based compounds of primarily biological origin, which are principally based on carbon-carbon and/or carbon-hydrogen bonds (Killops and Killops, 2005). From a quantitative perspective, most organic matter reaching marine sediments derives from phytoplankton sources from the surface ocean, with additional
contributions from bacteria and archaea involved in autotrophic chemosynthesis and heterotrophic processes, particularly in ODZs (Wakeham, 2020). Terrestrially derived organic matter can also be important along continental margins (e.g., Bianchi et al., 2018). Killops and Killops (2005), Eglinton and Repeta (2014), and Peters et al. (2005) provide detailed reviews on the production, composition, degradation, and preservation of organic matter in marine and terrestrial environments.

From a compositional perspective, organic matter largely consists of a few compound classes, including proteins (amino acids),
carbohydrates, nucleotides, nucleic acids, and lipids (Killops and Killops, 2005, Peters et al., 2005). Although the former often predominate quantitatively in fresh organic matter (Wakeham et al., 1997), lipids offer by far the largest range of applications in paleoceanography due to their preservation potential in sedimentary systems (Briggs and Summons, 2014; Luo et al., 2019). Lipids include a wide-range of compounds that are all characterized by their relatively small molecular size and their mostly hydrophobic nature, which makes them insoluble in water and soluble in organic solvents, such as alkanoic acids,
mono/di/triglycerides, waxes, phospho- and glycolipids, lipopolysaccharides, isoprenoids, hopanoids, steroids, terpenes, and also pigments, as well as their intact or fragmented fossil remains (Peters et al., 2005). In living cells, lipids play a central role as structural components of membranes, for energy storage, and as signalling molecules (Hazel & Williams, 1990; van Meer et al., 2008; Harayama et al., 2018). Their intact structure includes a recalcitrant hydrocarbon skeleton that can contain functional moieties such as unsaturations (double bonds) and functional groups (e.g., ester and ether bonds, ketyl, hydroxyl,
and carboxyl or amine groups).

The versatility of lipids as the basis of paleoceanographic proxies can be explained by (a) their overall preservation potential in sedimentary systems over geological time scales, (b) their chemotaxonomic and metabolic association with biological sources, (c) their role in controlling cellular physiological processes that lead to lipid remodelling (e.g., degree of unsaturation



or cyclization) in response to environmental stressors (e.g., temperature, oxygen, salinity), and (d) the preservation of stable isotope signatures (primarily carbon and hydrogen, but also nitrogen and sulphur) in their backbone skeletons (e.g., Eglinton & Eglinton, 2008; Eglinton and Repeta, 2014; Peters et al., 2005). Below we provide a brief overview of these processes and how they relate to the reconstruction of redox processes in paleoceanographic studies. Importantly, lipid biomarker applications in paleoceanography follow two approaches: 1) inferring specific source organisms or metabolisms (chemotaxonomy) prevalent in OMZ settings using intentionally biosynthesized compounds and their degradation products,

and 2) inferring redox conditions using lipid degradation products that only form under oxygen-deficient conditions and may either have ubiquitous sources or no known biological sources (orphan biomarkers). In case of chemotaxonomic approaches, it should be kept in mind that, while other sources of biomarkers may exist or may be discovered in the future (outlined below), independent sedimentological evidence can provide source constraints in a given setting.

### 6.3.2 Lipid Preservation and Redox Potential

The accumulation and preservation of organic material and lipids in marine sediments hinges on a series of physical and biogeochemical factors. These factors include (a) the amount of primary productivity in surface waters, (b) the processes controlling sinking fluxes and attenuation rates of particulate organic matter in its journey through the water column to the seafloor, (c) the nature and composition of the organics reaching the sediment/water interface, (d) the redox potential of the depositional environment, (e) the rates of sediment accumulation and burial, (f) the presence of protective minerals

(specifically clays), and (g) the availability of reduced sulphur species (e.g., Hedges and Keil, 1995; Blair and Aller, 2012). From their biosynthesis in cells to their preservation in sediments, lipids are subjected to a continuum of post-depositional transformations that modify their physico-chemical properties. Initially, these transformations are driven by diagenesis, predominantly microbial enzymatic degradation influenced by the redox potential, that lead to the hydrolysis of polar head groups, and/or the loss of functional groups, and/or the aromatization of ring structures, and the saturation of double bonds

(Killops and Killops, 2005; Peters et al., 2005). As sedimentary systems become impacted by tectonic processes and enhanced temperature and pressure gradients, catagenesis and metagenesis lead to changes in the three-dimensional configuration of the molecules (stereochemistry) and finally to their thermal cracking (Peters et al., 2005). Whereas the absolute abundance of organic material and lipids decreases along this continuum, the relative abundance of lipids within the total organic material pool increases as a consequence of their higher degradation resistance and preservation potential compared to other compound

classes such as carbohydrates and nucleic acids (Briggs and Summons, 2014; Luo et al., 2019). Despite the loss of structural information that lipids endure during degradation, their backbone skeletons preserve diagnostic paleoceanographic information that can be preserved for up to ~1.64 billion years, depending on factors such as oxygen exposure time and thermal maturity (Luo et al., 2019). Thus, since some lipids are more labile (i.e., more prone to degradation) than others (i.e., more recalcitrant), their utility as paleoceanographic proxies is determined by their preservation potential in sedimentary systems over geological

time scales (Fig. 6).



The sensitivity of organic matter preservation to bottom water oxygen has been long debated (Pedersen et al., 1992; Paropkari et al., 1992, 1993). Processes such as oxygen exposure time, the adsorption to mineral phases, and the rate of sediment accumulation have been shown to have the greatest impact (Hedges and Keil, 1995; Hartnett et al., 1998; Burdige, 2007;

Zonneveld et al., 2010; Arndt et al., 2013; Hemingway et al., 2019). Organic matter and lipid preservation are enhanced by reducing conditions at the water-sediment interface and within the sediment through (a) reduced exposure time to oxygen-utilizing enzymes, (b) decreased bioturbation, and (c) interactions with reduced sulphur species that lead to lipid sulphurization (e.g., Kohnen et al., 1991). Thus, variable organic matter and lipid preservation, as well as the extent of lipid sulphurization, provide a means of estimating past changes in bottom water oxygen. Empirical studies of organic matter preservation across a

range of bottom water oxygen levels in the Arabian Sea find enhanced preservation of as much as an order of magnitude when bottom water oxygen levels fall below a threshold ranging between 20 and 50 µmol kg$^{-1}$ (Cowie et al., 2014; Keil & Cowie, 1999; Koho et al., 2013; Rodrigo-Gámiz et al., 2016). Similarly, enhanced accumulation rates and/or preservation of TOC and specific biomarkers under low-oxygen conditions has been found in sediments of the Arabian Sea (Sinninghe Damsté et al., 2002c; Woulds et al., 2009) and off the east coast of the U.S. (Prahl et al., 2001). These studies have shown that the preservation

response of biomarkers is nonlinear and that there is a range of sensitivities among different lipid classes to bottom water oxygen.

Anderson et al. (2019) reported that, compared to bulk productivity proxies like opal and Ba$_{xs}$, the accumulation rates of algal lipid biomarkers in sediments deposited during the last glacial period in the Central Equatorial Pacific Ocean were five times greater than during the early Holocene due to low bottom water oxygen. This interpretation is consistent with independent bottom water

oxygen constraints at these sites during the last glacial period (20-50 µmol kg$^{-1}$) compared to modern bottom water oxygen concentrations (~170 µmol kg$^{-1}$) (Hoogakker et al., 2018; Umling & Thunell, 2018; Jacobel et al., 2020). Jacobel et al. (2020) demonstrated that when biomarkers are measured in parallel with inorganic proxies for productivity such as opal and Ba$_{xs}$, it is possible to discriminate between production and preservation as factors causing changes in concentration or accumulation rate of TOC or of individual compounds, such as oxygen diffusion into the sediments following an increase in bottom water oxygen. The

impacts of post depositional organic matter or biomarker oxidation, a process sometimes referred to as "burndown" (e.g., Colley et al., 1989; Colley and Thomson, 1985; De Lange, 2008, 1986; Prahl et al., 1989), can be reduced by working at locations with high sediment accumulation rates.

The accumulation and preservation of organic matter and biomarkers can also be enhanced through sulphurization, in which

organic matter and organic compounds react with sulphide (H$_2$S, HS$^-$) and/or polysulphides (S$_x^{2-}$), removing functional groups and generating cross-linked polymers that can be relatively resistant to breakdown by microbial exoenzymes (Sinninghe Damsté et al., 1988; Boussafir and Lallier-Vergès 1996; Van Kaam-Peters et al., 1998). Through these reactions, lipid biomarkers can be bound to high-molecular-weight organic matter (kerogen) via monosulphide (C-S-C) or disulphide (C-S-S-C) bonds (Vairavamurthy et al. 1992; Amrani and Aizenschtat 2004; Kutuzov 2019). These bonds can be broken during

catagenesis (Keleman et al., 2012) or by chemical desulphurization in the lab (Orr and White 1990; Prahl et al., 1996; Adam





et al., 2000). S-bound and especially disulphide-bound lipids appear to form during very early sedimentation and diagenesis, sometimes prior to the appearance of detectable dissolved sulphide in pore water (Francois, 1987). Early sulphurization can trap biomarker signals before diagenetic reworking and can make these S-bound lipids a relatively high-fidelity archive of biomarker information. In both modern and ancient sediments, S-bound lipid distributions are often distinct from free

(extractable) lipids, reflecting important aspects of environmental oxygenation such as pigments, steroid distributions, and C- or S-isotope compositions (Kohnen et al., 1991b; Wakeham et al., 1995; Kok et al., 2000; Rosenberg et al., 2018; Ma et al., 2021). Sulphide in the environment may also contribute to the stabilization of free lipids by reducing double bonds (Hebting et al., 2006). Reconstructions of lipid distributions from sulphidic environments should consider the potential for sulphurization to transform, bias, and/or preserve biomarker information.

Understanding the location and timing of sulphurization also provides insights into the distribution and intensity of anoxia. For example, intervals of enhanced sulphurization and preservation of carbohydrate-derived organic matter in a TOC-rich Jurassic black shale were attributed to photic zone euxinia during deposition (Boussafir et al., 1995; van Kaam-Peters et al., 2003; van Dongen et al., 2006). Changes in sulphurization intensity have also been linked to shifts in the distribution of anoxia across OAE2 (Raven et al., 2018). Sulphurization intensity can be approximated by S:C molar ratios, where values greater

than about 0.02 exceed the initial sulphur content of most marine photosynthetic biomass (Francois, 1987). Higher S:C ratios require highly functionalized and therefore relatively young organic precursors prior to sulphurization (Brassell, 1985). This early sulphurization, prior to burial, may impact a relatively large pool of functionalized organic matter (Raven et al., 2019). Subsequently, sulphurization over thousand-year timescales can continue to impact free lipid biomarkers such as tricyclic triterpenoids and steroids (Shawar, 2021; Werne et al., 2004 Kok et al., 2000). Different sulphurization products and biogenic

organic S can be distinguished through spectroscopy, especially synchrotron x-ray absorption spectroscopy (Kohnen 1989; Amrani 2004; Vairavamurthy 1998; Raven 2021b), or by using stable isotopes. Organic sulphur in biomass from oxic systems generally has an isotopic composition similar to that of sulphate in the environment (Kaplan and Rittenburg, 1964; Trust and Fry, 1992), while organic S from sulphurization typically has variable but broadly more [34]S-depleted isotopic values (Anderson and Pratt 1995; Canfield 2001). In detail, individual organic sulphur compounds have a wide range of stable sulphur isotope

($\delta^{34}$S) values (up to tens of ‰) and may preserve additional layers of information about the lipid pool during early diagenesis (Amrani et al., 2005; Raven et al., 2015; Rosenberg et al., 2017; Shawar et al., 2020). Sulphurization indicators can thus complement multi-proxy reconstructions of redox conditions at the same time that they provide insights into taphonomic bias and organic matter burial.

### 6.3.3 Biomarkers as tracers of microbial processes associated with oxygen deficiency

Biomarkers provide chemotaxonomic and metabolic information of source organisms inhabiting water columns and/or sediments impacted by a wide range of redox conditions and electron acceptors. This is particularly important for biomarkers from organisms associated with the cycling of nitrogen (e.g., anammox, ammonia oxidation, nitrogen fixation), sulphur (sulphate reduction, sulphide oxidation), and carbon (methanogenesis and methanotrophy), as well as from those organisms



feeding on them. Thus, biomarkers provide qualitative redox information that ranges from fully oxygenated conditions to oxygen-deficient, anoxic non-sulphidic, and anoxic sulphidic/euxinic conditions (both within and below the photic zone). We refer to Tab. S1 for a tabularized summary of biomarkers described in the following sections (6.3.3.1 to 6.3.6.2).

### 6.3.3.1 Biomarkers for Nitrogen Cycling in ODZs

Oxygen-deficient environments are hotspots for microbial processes involved in the removal of fixed nitrogen, such as denitrification and other dissimilatory nitrogen transformations, anaerobic ammonium oxidation (anammox) coupled to nitrite reduction, and nitrite-dependent anaerobic methane oxidation (n-damo) (Lam and Kuypers, 2011; Thamdrup, 2012). Accordingly, the presence of lipids synthesized by bacteria with these anaerobic metabolisms can be used to infer hypoxic or anoxic conditions in the past. For detailed reviews on the use of biomarkers for nitrogen cycling, we recommend the recent work by Rush and Sinninghe Damsté (2017) and Kusch and Rush (2022).

### 6.3.3.1.1 Anammox

Anammox, the anaerobic oxidation of $NH_4^+$ to $N_2$ using $NO_2^-$, is the only nitrogen loss process for which several specific biomarkers have been identified. Ladderanes are highly unusual lipids with moieties consisting of cyclobutane rings (Sinninghe Damsté et al., 2002a). They make up the cell membrane of the anammoxosome, the specialized organelle in which the anammox process takes place, and substantially reduce proton permeability (Moss et al., 2018). A suite of ladderane fatty acids and their short chain oxic degradation products are available to trace anammox. However, ladderanes do not preserve well (Rush et al., 2011) and, thus far, the oldest sedimentary record containing ladderanes extends only 140 kyr (Jaeschke et al., 2009). More recently, a unique bacteriohopanepolyol (BHP) biomarker for marine anammox '*Candidatus* Scalindua profunda' was identified (Schwartz-Narbonne et al., 2020): a bacteriohopanetetrol isomer with unknown stereochemistry, BHT-x (Rush et al., 2014; Schwartz-Narbonne et al., 2020), has strongly depleted $\delta^{13}C$ values in sediments consistent with fractionation associated with the reductive acetyl-CoA pathway used by anammox (Hemingway et al., 2018; Lengger et al., 2019), and has a distinct niche in the nitrite maximum of stratified water column settings (e.g., Kusch et al., 2022; Matys et al., 2017). In comparison to ladderanes, BHPs preserve well and BHT has been detected in samples as old as ca. 50–55 Myr (van Dongen et al., 2006; Talbot et al., 2016). In older rocks, BHPs may survive as hydrocarbons after the loss of hydroxyl functionalities due to reducing processes, or after decarboxylation reactions that also shorten the hopanoid side chain. The diagnostic value of these resulting hopenes and hopanes is reduced in comparison to the original lipid. So far, BHT-x has been successfully used to trace OMZ conditions in (non-dated) sediment records in the Benguela Upwelling system (van Kemenade et al., 2022), during the last Glacial in the Gulf of Alaska (Zindorf et al., 2020), and during Pliocene/Quaternary sapropel formation in the Mediterranean (Rush et al., 2019; Elling et al., 2021). For paleoceanographic purposes, sedimentary ladderanes and BHT-x should primarily capture the water column anammox signal (nitrite maximum), which can be orders of magnitudes higher than the benthic background signal (Rush et al., 2012).





### 6.3.3.2 Nitrite-dependent anaerobic methane oxidation (n-damo)

More recently, bacteria performing anaerobic methanotrophy have been detected. '*Ca.* Methylomirabilis oxyfera' is an exceptional methanotroph that produces its own oxygen via the production of NO by $NO_2^-$ reduction (Ettwig et al., 2010), also known as n-damo. Biomarkers of n-damo include bacteriohopanehexol, 3Me-bacteriohopanehexol, and 3Me-

bacteriohopanepentol (Kool et al., 2014) as well as the novel demethylated hopanoids 22,29,30-trisnorhopan-21-ol, 3Me-22,29,30-trisnorhopan-21-ol, and 3Me-22,29,30-trisnorhopan-21-one (Smit et al., 2019). Although not common, bacteriohopanehexol is also produced by thermophilic *Alicyclobacillus acidoterrestris* (Řezanka et al., 2011) and the bacterial symbiont of a marine *Petrosia* sponge (Shatz et al., 2000). Thus, the C-3 homologs of bacteriohopanehexol and bacteriohopanepentol as well as the trisnorhopanes may be better indicators for the presence of '*Ca.* Methylomirabilis oxyfera'

and n-damo. Although there seem to be several n-damo biomarkers, it should be noted that the role of n-damo bacteria (NC10 phylum) in marine ODZs is still not well constrained (e.g., Padilla et al., 2016). However, to date the presence of the above mentioned biomarkers in marine sediments can likely be interpreted to indicate anoxia.

### 6.3.3.2 Feedback mechanisms to nitrogen loss

Removal of nitrogen in OMZ settings causes imbalances in N:P ratios that can promote/intensify aerobic processes involved

in the nitrogen cycle. For instance, in the geologic record, enhanced diazotrophy, the primary source of bioavailable nitrogen in the ocean (Hutchins et al., 2008), has been invoked to have sustained biological productivity during times of intensified ocean deoxygenation and consequent fixed nitrogen loss (e.g., Kuypers et al., 2004). Accordingly, biomarker evidence for important feedback mechanisms to nitrogen loss (e.g., diazotrophy and nitrification), can provide context for paleoceanographic data and shed light on past OMZ-related biogeochemical cycles.


Biomarkers for reconstructing nitrogen fixation have been long sought after. To date, there are no biomarkers commonly produced by cyanobacteria, who are the dominant diazotrophs, or the subset of cyanobacteria that can fix nitrogen (e.g., Talbot et al., 2008; Saenz et al., 2012). Until the discovery of diazotroph biomarkers, molecular evidence for $N_2$-fixation and denitrification has been mostly based on the characteristic kinetic isotope fractionation effects (e.g., Sigman, 2009) preserved

in the $\delta^{15}N$ values of nitrogen-containing organics such as pigments and proteins (e.g., Sachs et al., 1999; Ohkouchi et al., 2006; Higgins et al., 2010, 2012; Junium et al., 2015). For example, the consistent observation of low $\delta^{15}N$ values of chlorophyll-derived tetrapyrroles in ancient sediments deposited under wide-spread anoxic and euxinic conditions (e.g., OAEs) and the abundance of chlorophyll-derived degradation products suggest a direct link between surface water $N_2$-fixation and water column N-loss processes (e.g., Junium and Arthur, 2007; Ohkouchi et al., 2006). Recent studies also suggest that $\delta^{15}N$

values of amino acids ($\delta^{15}N_{AA}$) may be useful tools to study water column nitrogen dynamics (McCarthy et al., 2013; Batista et al., 2014) since phenylalanine $\delta^{15}N$ values show a good relationship with established $\delta^{15}N$ proxies such as bulk sediment and foraminifera-bound N (Li et al., 2019) (see also Section 6.4; nitrogen isotopes). However, for a subset of diazotrophic



cyanobacteria, so-called heterocystous cyanobacteria, specific heterocyst glycolipids (HGs) with C5 and C6 sugar head groups have now been identified (Bauersachs et al., 2009a,b; Bauersachs et al., 2010). HGs comprise the innermost laminated layer

of the heterocysts, forming a protective envelope for the oxygen-sensitive nitrogenase enzyme (e.g., Gambacorta et al., 1998). C5 sugar HGs are proposed to be specific biomarkers for marine endosymbiotic heterocystous cyanobacteria while C6 sugar HGs occur in free-living heterocystous cyanobacteria (Schouten et al., 2013b; Bale et al., 2015). Enhanced deposition of C5 and C6 HGs in the Mediterranean Sea during Plio-Pleistocene sapropel events has been linked to anoxia, indicating that diazotrophy by heterocystous cyanobacteria was an important feedback to nitrogen loss (Bale et al., 2019, Elling et al., 2021).

Heterocystous cyanobacteria occur mostly in brackish water bodies (e.g., Baltic Sea), but are rare in the open ocean (except as symbionts of diatoms; Stal et al., 2009; Zehr et al., 2011). However, diazotrophy is also observed in open ocean ODZ systems that are associated with enhanced upwelling and primary production such as the Eastern Tropical Pacific (White et al., 2013; Loescher et al., 2014; Jayakumar et al., 2017). It is unknown whether HGs can also track heterocystous cyanobacteria in these environments in the past or present. Thus, tetrapyrrole $\delta^{15}N$ values may still provide the most unequivocal evidence for $N_2$-

fixation in the past. Since they preserve well over long timescales (the oldest tetrapyrroles date to 1.1 Ga; Gueneli et al., 2018), the nitrogen isotopic composition of these molecules or their smaller maleimide fragments (e.g., Grice et al., 1996) can be used to gauge $N_2$-fixation over much of Earth's history.

Nitrification, the aerobic transformation of ammonia ($NH_4^+$) to nitrite ($NO_3^-$), is performed either as a two-step process by

ammonia-oxidizing bacteria (AOB), ammonia-oxidizing archaea (AOA) and nitrite-oxidizing bacteria (NOB), or by complete ammonia-oxidizing bacteria (comammox). Although nitrification is considered an obligately aerobic process, AOA and NOB persist in suboxic and anoxic waters, and two novel *Nitrospina*-like lineages (NOB) have been found and implicated in nitrite oxidation in ODZs (Sun et al., 2019). Thus, for the majority of known AOA and NOB, active nitrification under low-oxygen conditions requires a source of cryptic oxygen, i.e., the presence of short-lived oxygen-bearing intermediates that typically

occur below detection limit but provide important substrates (Kappler & Bryce, 2017). For AOA, internal oxygen production has been observed as a response to anoxia (Kraft et al., 2022). AOA are the dominant sources of glycerol dialkyl glycerol tetraethers (GDGTs) to marine sediments and produce the specific biomarker crenarchaeol (Sinninghe Damsté et al., 2002b), methoxy archaeol (Elling et al., 2014, 2017), as well as specific quinones ($MK_{6:0}$ & $MK_{6:1}$; Elling et al., 2016). Crenarchaeol has been shown to track Thaumarchaeota in the suboxic zones of modern (e.g., Wakeham et al., 2007; Sollai et al., 2015;

Kusch et al., 2021) and paleo systems, particularly during times of ocean deoxygenation, such as during Mediterranean sapropel deposition (Menzel et al., 2006; Polik et al., 2018). As such, increased deposition of crenarchaeol relative to organic matter may be useful for tracing intensified suboxic-anoxic nitrogen cycling (Rush et al., 2017; Elling et al., 2020). The presence of crenarchaeol alone can, however, not be used to infer suboxic conditions. AOB, NOB, and comammox do not seem to synthesize chemotaxonomically specific lipids. Known lipids of AOB include generic BHPs and unsaturated fatty

acids (Sakata et al., 2008) and some hopanoids produced by NOB have not previously been found in other bacteria (Rush & Sinninghe Damsté, 2017; Elling et al., 2022).



### 6.3.4 Biomarkers for Sulphur Cycling in ODZs

In addition to the abiotic sulphurization mechanisms described above, ODZs are characterized by active sulphide oxidation
and sulphate reduction mediated by diverse bacteria (e.g., Callbeck et al., 2021; van Vliet, et al., 2021). These sulphur
metabolisms are not only present in sulphide-rich anoxic sediments and euxinic water columns, but open ocean ODZs and
particle microniches also harbour a cryptic sulphur cycle (Canfield et al., 2010; Raven et al., 2021a). Diverse biomarkers with
high preservation potential have been identified for various sulphide-oxidizing bacteria (SOB) and sulphate-reducing bacteria
(SRB) and allow detailed reconstructions of water column stratification in paleoceanographic studies.

### 6.3.4.1 Sulphide oxidation

Phototrophic green sulphur bacteria (GSB; Chlorobiaceae) and purple sulphur bacteria (PSB; Chromatiaceae) are the principal
microbes metabolizing reduced sulphur species, such as $H_2S$, whilst fixing carbon through anoxygenic photosynthesis. As such
they occupy anoxic environmental niches with access to light and $H_2S$, amongst other sulphur species (Summons and Powell,
1987). Apart from benthic microbial mats at shallow water depths, this involves euxinic photic zones of marine water columns.
Characteristically, green and purple sulphur bacteria biosynthesize diaromatic carotenoids that function as accessory pigments.
Isorenieratene and chlorobactene, as well as their fossilized equivalents isorenieratane and chlorobactane, are commonly used
as indicators for the presence and activity of green- (chlorobactane) and brown-pigmented (isorenieratane) species of the
Chlorobiaceae during sediment deposition, providing clues about the depth of the chemocline given that brown pigmented
Chlorobiaceae are adapted to lower irradiance than their green-pigmented relatives (Summons and Powell, 1987; Schaeffer et
al., 1997; French et al., 2015). In very shallow chemoclines, the relative abundance of anoxygenic phototrophy is typically
skewed towards a higher proportion of PSB, which characteristically biosynthesize the monoaromatic carotenoid okenone that
can survive in sediments as the fossil equivalent okenane (Brocks and Schaeffer, 2008). All of the saturated $C_{40}$ carotenoids
can survive for exceedingly long time spans and have been detected in sediments up to 1.64 Ga in age (see review and updated
analytical method in French et al., 2015). Once subjected to thermal breakdown during sedimentary burial, the methylation
pattern of the remaining arylisoprenoids (2,3,4- vs. 2,3,6- substitution pattern) can still yield clues to the biological precursor,
whereas the relative abundance of longer versus shorter aryl isoprenoid chains may allow distinguishing long-lived and
persistent euxinia from short-lived and episodic photic zone euxinia (Schwark and Frimmel, 2004). Using the reverse
tricarboxylic acid cycle during carbon fixation, biomass of green sulphur bacteria may also be recognized by their characteristic
enrichment in $^{13}C$ compared to that of oxygenic phototrophs, whilst their bacteriochlorophyll-c/d/e pigments can be recognized
both intact, as well as after breakdown to maleimides such as 3-isobutyl-4-methylmaleimide (e.g., Grice et al., 1996; Naeher
et al., 2013).



### 6.3.4.2 Sulphate reduction

Sulphate reduction is a heterotrophic anaerobic bacterial pathway leading to the formation of hydrogen sulphide, which can fuel the cryptic sulphur cycle in offshore ODZs by supplying reactive sulphur species as intermediates for other redox reactions (Callbeck et al., 2021). A group of compounds commonly associated with SRB are non-isoprenoid 1-*O*-monoalkyl or 2-*O*-monoalkyl glycerol ethers (MAGEs) and 1,2-*O*-dialkyl glycerol ethers (DAGEs). They have been identified in hyperthermophilic bacteria and commonly occur in settings influenced by hydro/geothermal activity (e.g., Bradley et al., 2009) or seep systems hosting consortia of anaerobic methane oxidizing archaea (ANME) and SRB (Niemann & Elvert, 2008). Hernandez-Sanchez et al. (2014) also identified 1-*O*-MAGEs in suspended particulate matter sampled from oxygenated surface waters and suggested a role of bacteria other than SRB in the production of these lipids. However, recent evidence of sulphate reduction in sinking marine particles (Raven et al., 2021a) can explain the observation of SRB or 1-*O*-MAGEs in oxygenated surface waters and sediments (e.g., Hernandez-Sanchez et al., 2014; Teske et al., 1996). SRB belonging to the *Desulfosarcina*/*Desulfococcus* group (syntrophic partners of the ANME-1 and -2 clades, except ANME-2d) and *Desulfobulbus* spp. (syntrophic partner of the ANME-3 clade) also produce characteristic alkanoic acid fingerprints with strong [13]C-depletion, including C16:1ω5 ($C_{16:1}\Delta^{12}$), cy-C17:0ω5,6 and C17:1ω6 (Niemann and Elvert, 2008). No biomarkers are known for sulphur-disproportionating bacteria, which perform reverse sulphate reduction.

### 6.3.5 Biomarkers for Carbon Cycling in ODZs

Oxygen-deficient conditions in the ocean are also intimately linked to the methane cycle via both the generation (methanogenesis) and utilization (methanotrophy) of methane. Methanogenesis and anaerobic methanotrophy are performed by anaerobic archaea using sulphate, nitrate, iron, and manganese as electron acceptors (for a recent overview see Guerrero-Cruz et al., 2021). Methane is also respired by aerobic methane-oxidizing bacteria (MOB). Although these MOB are aerobes, their lipids are useful proxies in paleoceanography since MOB are typically present at oxic-anoxic transitions where both methane and oxygen are available and can thrive under oxygen-deficient conditions (Guerrero-Cruz et al., 2021). In addition, the utilization of methane is recorded by strongly [13]C-depleted biomarker signatures irrespective of their chemotaxonomic specificity (e.g., Jahnke et al., 1999).

### 6.3.5.1 Methanogenesis

Methane in the ocean is primarily produced in anoxic marine sediments although aerobic sources also exist (Metcalf et al., 2012; Bižić et al., 2020). In sediments, methanogenesis is performed by strictly anaerobic primarily hydrogenotrophic and acetoclastic Euryarchaeota (Ferry and Lessner, 2008). Methanogenic Euryarchaeota primarily produce isoprenoid tetraethers without cyclic moieties (GDGT-0) and the isoprenoid diether archaeol (Koga et al., 1993, 1998). Ratios >2 of GDGT-0 over the thaumarchaeal isoprenoid tetraether crenarchaeol have been proposed as a proxy for methanogens in the paleo record





(Blaga et al., 2009), although these ratios have to be interpreted in the context of the biomarker assemblage due to the presence of GDGT-0 in many non-methanogenic archaea (Schouten et al., 2013a; Elling et al., 2017).

Likewise, *Methanothermococcus thermolithotrophicus* has been shown to synthesize hydroxylated GDGTs (OH-GDGT) with

0-2 pentacyclic moieties (Liu et al., 2012). Although the lipids mentioned above are not exclusive to methanogens, other sources such as methanotrophic Euryarchaeota or Thaumarchaeota typically have much more diverse GDGT fingerprints (Schouten et al., 2013a; Elling et al. 2017) and also synthesize additional lipids (see below). Since methanogenesis is performed in situ, for paleoceanographic purposes it is crucial to identify in situ overprints. One approach to distinguish paleo and in situ signals is the screening for biomarkers that imply metabolic activity, such as the functional quinone analogs

methanophenazines (Abken et al., 1998; Elling et al., 2016), which (to date) have only been shown to be produced by the order *Methanosarcinales*, or coenzyme F430, the cofactor of methyl coenzyme M reductase possessed by all methanogens (Kaneko et al., 2021). It should, however, be noted that anaerobic methanotrophic archaea are also suspected to produce these biomarkers (although direct evidence is still missing).

### 6.3.5.2 Methanotrophy

Anaerobic methanotrophy with sulphate (reverse methanogenesis) is performed by ANME archaea in syntrophic consortia with sulphate-reducing bacteria (Boetius et al., 2000). ANME biomarkers include isoprenoids such as tetramethylhexadecane (crocetane; ANME-2), pentamethylicosane (PMI) and unsaturated PMIs, archaeol, and *sn*2-hydroxyarchaeol or *sn*3-hydroxyarchaeol (Koga et al., 1993, 1998; Elvert et al., 1999; Thiel et al., 1999; Hinrichs et al., 2000; 2003). Furthermore, ANME ecotypes (classified as ANME-1, ANME-2, ANME-3) seem to be discernible by *sn*2-hydroxyarchaeol/archaeol ratios

and the $\delta^{13}$C signature of archaeol, the alkanoic acid signature of the SRB partners (Blumenberg et al., 2004; Niemann & Elvert, 2008), and specific intact polar lipid (IPL) compositions (e.g., Rossel et al., 2011), although the IPL characteristics might be lost in the paleo record. However, the remaining core lipid GDGT signature may aid the identification of ANME in the paleo record. GDGT-2/crenarchaeol ratios exceeding the threshold of 0.2 could indicate the presence of ANME (Weijers et al., 2011). The characteristic depletion in $^{13}$C of ANME biomarkers can be traced in the paleo record.


Methane produced under anoxic conditions can also be utilized by aerobic MOB, and their presence in paleoceanographic records indicates a methane-rich environment. Aerobic MOB synthesize a range of characteristic lipids, including a suite of amino-functionalized BHPs and their respective unsaturated and C-3 methylated homologs. Aminopentol is considered a characteristic biomarker for Type I gammaproteobacterial MOB and aminotetrol is commonly produced by Type II

alphaproteobacterial MOB (e.g., Rohmer et al., 1984; Jahnke et al., 1999; Talbot et al., 2001). They also produce structurally similar methylcarbamate (MC) BHPs (MC-pentol and MC-tetrol), which seem to be much more common in methane-influenced marine environments that often lack aminopentol (Rush et al., 2016). It should be noted that minor amounts of aminopentol and aminotetrol are also produced by SRB of the *Desulfovibrio* genus (Blumenberg et al., 2006; 2012), NOB (Elling et al., 2022), and several terrestrial thermophilic bacterial species (Kolouchová et al., 2021). Likewise, small amounts



of MC-triol have recently been identified in cultures of *Nitrobacter vulgaris* and marine *Nitrococcus mobilis* (Elling et al., 2022). C-3 methylation of hopanoids alone can no longer unequivocally be linked to AOM unless confirmed by depleted $\delta^{13}C$ values. The gene for C-3 methylation is not present in all methanotrophs but present in various non-methanotrophic bacteria (Welander and Summons, 2012), C-3 methylated BHPs accumulate in the euxinic Black Sea water column (Kusch et al., 2022), and other known sources include the phototrophic purple non-sulphur bacterium *Rhodopila globiformis* (Mayer et al.,

2021). The presence of MOB may, however, also be confirmed by other biomarkers such as alkanoic acids and quinones. Type I MOB produce methylene-ubiquinone $MQ_{8:7}$ (Nowicka & Kruk, 2010), which has been observed in the suboxic and anoxic zones of the Black Sea (Becker et al., 2018), and *Methylococcaceae* synthesize $C_{16:1}\Delta^{8c}$ (C16:1ω8c) and $C_{16:1}\Delta^{12t}$ (C16:1ω5t) alkanoic acids (e.g., Bodelier et al., 2009). Type II MOB of the *Methylocystaceae* and *Beijerinckiaceae* are characterized by $C_{18:1}\Delta^{10c}$ (C18:1ω8c) (e.g., Bodelier et al., 2009).

**6.3.6 Non-specific and orphan biomarkers that accumulate in oxygen-deficient depositional settings**

**6.3.6.1 Redox controlled processes**

Independent of their specific biological source, various lipids will undergo diagenetic molecular modifications that are principally controlled by environmental redox chemistry. One of the first established indicators for oxic versus anoxic conditions was the ratio of pristane (Pr) over phytane (Ph) — $C_{19}$ and $C_{20}$ isoprenoid hydrocarbons that derive from the phytol

sidechain of chlorophyll by oxidative decarboxylation or by reduction, respectively (Rontani et al., 2003). Strongly elevated values (e.g., >4) are observed under oxic conditions whereas values <<1 are found under anoxic conditions, yet the use of the Pr/Ph index is complicated by alternative (non-chlorophyll-derived) sources of both Pr and Ph (e.g., Goossens et al., 1984), as well as by questions of organic matter transport pathways and oxygen exposure (e.g., Ten Haven et al., 1987). A hopanoid-based indicator involving the relative abundance of long chain $C_{31}$-$C_{35}$ homohopanes over $C_{30}$ hopanes (known as the

homohopane index) follows a similar rationale and assumes that longer side chains are preferentially preserved in sediments under reducing conditions (Peters et al., 2005). Similarly, phototroph-derived chlorophylls are commonly used as indicators of primary productivity (Carpenter et al., 1986; Harris et al., 1996). However, certain degradation products such as pyropheophytin and steryl chlorin esters are only formed under anoxic conditions (Szymczak-Żyla et al., 2008), and the proportional abundance of these chlorophyll degradation products has been proposed as proxy for bottom water anoxia (e.g.,

Szymczak-Żyła et al., 2017).

**6.3.6.2 Orphan biomarkers**

A range of lipids have been shown to be associated with OMZ settings for which the source organisms are unknown. Although these lipids cannot be linked to specific taxa or metabolisms, they can still be useful indicators for paleoceanographic purposes. One common orphan biomarker in sediments from anoxic settings is the isoprenoid 19,23,27,31-octamethyldotriacontane

(lycopane) for which methanogenic archaeal (Brassell et al., 1981) and phototrophic (Wakeham et al., 1993) origins have been



suggested. For paleoceanographic purposes, the lycopane/$C_{31}$ $n$-alkane ratio has been suggested as a proxy for paleoxicity (Sinninghe Damsté et al., 2003), although it must be applied with caution in areas where plant wax input is large.

Derivatives of branched GDGTs such as overly branched GDGTs (OB-GDGTs), are produced in specific patterns in anoxic
marine zones (Liu et al., 2014; Xie et al., 2014) and have been interpreted as biomarkers for OMZ presence in the paleo record (Connock et al., 2022). Yet, the sources of these lipids, oxygen thresholds for their production, and potential production in sediments remain unstudied.

### 6.3.7 Analyses and resources required

For biomarker analyses, sediments are extracted with organic solvents to recover the total lipid extract (TLE, or bitumen),
which is further processed to separate compound classes that are subsequently analysed using mass spectrometry techniques. Various established extraction methods exist that differ in choice of extraction agent (pure organic solvents of different volatility and polarity, or mixtures that contain water and/or buffers) as well as extraction technique ('manual' techniques such as ultrasonication and Soxhlet, or automated pressurized systems such as ASE and microwave). Extraction efficiencies between these methods differ at the compound class or functional group level, thus, should be chosen depending on target
compounds. Extraction is typically followed by wet-chemical processing to obtain polarity fractions (e.g., saponification and column chromatography using self-packed columns or commercially available SPE columns). The removal of elemental sulphur using activated Cu may be necessary during sample processing to avoid interference during chromatography, whereas a desulphurization treatment with Raney-nickel can be applied to reduce organosulphur compounds, and hence release S-bound organics that would otherwise evade detection (Kohnen et al., 1991).


Non-polar compounds in the size range up to ca. 600 Da can be analysed using gas chromatography-mass spectrometry (GC-MS), whereas larger and/or more polar compounds require the use of liquid chromatography-mass spectrometry (LC-MS) techniques, or derivatization reactions and/or cleavage of polar functional groups. Different types of mass spectrometers are being used (see below) that differ in mass resolution and accuracy. To obtain compound-specific stable isotope values ($\delta^{15}$N,
$\delta^{13}$C, $\delta^{2}$H), organic geochemists use gas chromatography coupled to isotope ratio mass spectrometry (GC-IRMS) systems. In case of lipids that are not GC amenable, compounds are either cleaved off their polar moieties for GC-IRMS analysis (resulting in loss of specificity and information) or compounds are isolated using LC and subsequently analysed using spooling wire micro-combustion IRMS (Pearson et al., 2016) or nano-elemental analyser (EA)-IRMS analysis (e.g., Kusch et al., 2010; Ogawa et al., 2010). More recently, isotope abundances have also been determined for polar compounds using high-
temperature (HT-)GC-IRMS (e.g., Lengger et al., 2021).





### 6.3.8 Major analytical advancements

Major advancements in the field of organic geochemistry have been made since the introduction of high-resolution accurate mass-mass spectrometry (HRAM-MS) techniques, including quadrupole time-of-flight (Q-ToF), Orbitrap, and Fourier transform ion cyclotron resonance (FT-ICR). Paired with ultra-high performance LC systems, improved column materials and

chemistries (e.g., core shell, HILIC), Q-ToF and Orbitrap platforms have opened up a new window into lipidomics of environmental samples (e.g., Hopmans et al., 2021; Wörmer et al., 2013). Orbitrap technology specifically offers substantial further analytical potential that includes compound-specific isotope analysis and position-specific isotope analysis (Eiler et al., 2017). Recent analytical advancements have also been made using scanning techniques. Matrix-assisted laser desorption/ionization (MALDI-)FT-ICR-MS allows mapping of spatial biomarker abundances in situ, which facilitates

obtaining ultra high-resolution records from sediment cores (e.g., Alfken et al., 2021). For a comprehensive overview of high-resolution analytical organic geochemical methods, we refer to Steen et al. (2020).

### 6.3.9 Future prospects

Biomarkers are excellent tools in paleoceanography due to their preservation potential in sediment and their biological and environmental associations. Open ocean ODZs in general are high productivity systems that foster preservation of high

amounts of organic matter. Moreover, the high preservation potential of lipids leads not only to high abundances of biomarkers in the paleo record, but also the preservation of a diverse pool of compounds. This structural diversity allows use of comprehensive biomarker assemblage approaches to reconstruct the water column structure (e.g., Connock et al., 2022; Dummann et al., 2021). Biomarkers also preserve well under conditions that lead to the absence of calcareous microfossils or trace metals in sediments, such as carbonate undersaturation or oxygen exposure. Nonetheless, challenges as well as new

frontiers remain for the biomarker community. Most profoundly, biomarker proxies have yet to allow the quantitative reconstruction of oxygen concentrations in the past. Indirectly, the half-maximal inhibitory concentration ($IC_{50}$) of oxygen in specific organisms provides a means to infer maximum oxygen concentrations. For example, anammox bacteria have an upper oxygen limit of ~20 µmol kg$^{-1}$ (Kalvelage et al., 2011) and a recent study suggests that BHT-x ratios (the normalized abundance of BHT-x/[BHT+BHT-x]) of ≥ 0.2 indicate oxygen concentrations <50 µmol kg$^{-1}$ (van Kemenade et al., 2022). Recently, Kim

and Zhang (2023) demonstrated how the methane index can be used to calculate sedimentary methane fluxes. Analogous proxies to reconstruct oxygen are lacking, as a range of lipids are produced in the water column as well as in sediments (e.g., anammox, sulphate reduction), and different biomarkers may sometimes provide divergent paleoenvironmental information (e.g., in settings receiving input from land or affected by lateral input). In addition to making the available biomarker tools more quantitative, efforts of the biomarker community are also focused on exploiting the potential afforded by technological

advances to expand the utility of biomarker applications in paleoceanography. New isotope tools, such as compound-specific [15]N analysis of amino acids, as well as the discovery of new biomarkers through lipidomics approaches in combination with (meta)genomics offer large potential to trace oxygen limitation in more detail. These new tools may also aid the identification



of the sources of orphan biomarkers that accumulate under anoxic conditions. Identifying the source organisms will improve our understanding of the metabolism or ecological niche recorded by these lipids and, ultimately, the paleoenvironmental conclusions drawn from them.





**Figure 6 : A) Idealized water column and sediment redox zonation and associated metabolisms and their biomarker signatures discussed in the text. Dashed lines indicate species and metabolisms that may be present in specific settings B) Biomarker**





preservation timescale. Solid lines indicate timescales on which distinct compound classes are preserved. Note that observations are less frequent where the scale extends to the oldest known record.

## 6.4 Nitrogen isotopes

### 6.4.1 Introduction

Nitrogen (N) has two stable isotopes, $^{14}N$ and $^{15}N$. The lighter isotope, $^{14}N$, comprises $99.63\pm0.02\%$ of the N found on Earth. Natural processes in the ocean discriminate between the two isotopes leading to subtle changes in the $^{15}N/^{14}N$ ratio of different nitrogen compounds. This is reported in notation, with atmospheric $N_2$ as the reference material:

$$\delta15N\ (‰\ vs.\ Air) = ((^{15}N/14N)Sample/(15N/14N)Air - 1)\cdot 1000$$

The most widely used technique for N isotopic analysis relies on online combustion of samples to $N_2$ using an elemental analyzer (EA) coupled with isotope ratio mass spectrometry (IRMS). This method requires a typical sample size of 1–2 mmol N per analysis, with analytical precision near 0.1% (1 standard deviation) and has mostly been applied to bulk sedimentary N (Robinson et al., 2012; Tesdal et al., 2013). Ongoing innovations, for example through cryo-focusing of the resulting $N_2$ sample gas, allows for much reduced sample sizes (~25-40 nmol N) (Polissar et al., 2009). Specific organic compounds, amino acids and other polar compounds typically require a derivatization process for gas chromatograph-based analysis. Amino acids can also be isolated using liquid chromatography followed by micro-combustion to $N_2$, which is commonly applied prior to the IRMS analysis of the resulting $N_2$ (Ohkouchi et al., 2017; Ishikawa et al., 2022).

In addition to $N_2$, $N_2O$ can be analysed by IRMS for N isotopic analysis. As $N_2O$ has a much lower atmospheric background relative to $N_2$, the sample size requirement can be further reduced when $N_2O$ is the final analyte. Methods have been developed for the conversion of nitrate ($NO_3^-$) and nitrite ($NO_2^-$) to $N_2O$, preceded by chemical procedures to convert different N forms to $NO_3^-/NO_2^-$ (Sigman et al., 2001; McIlvin and Altabet, 2005; Weigand et al., 2016). The $N_2O$-based method, using denitrifying bacteria, has a typical sample size requirement of 2-5 nmol N, with analytical precision better than 0.2‰ (Weigand et al., 2016). This technique has expanded the range of sample types accessible to isotopic analyses and has successfully been applied to analyze a range of fossil-bound N, including diatoms, foraminifera, corals, among others.

Nitrogen cycling is closely linked to the dissolved oxygen content in the ocean. In ODZs, where water column oxygen concentrations are lower than roughly <5 μmol kg$^{-1}$, the bacterial reduction of $NO_3^-$ to $N_2$, also known as denitrification, removes $NO_3^-$ from the ocean (Sigman and Karsh, 2009). Denitrification strongly discriminates against the heavier isotope, progressively increasing the $\delta^{15}N$ of the remaining $NO_3^-$ pool. Culture and field studies show an isotope effect between 15‰ and 25‰ for denitrification (Sigman and Fripiat, 2019). In addition, recent work suggests that anaerobic ammonium oxidation (anammox), where $NO_2^-$ is used to oxidize ammonium ($NH_4^+$) to $N_2$, could also lead to an increase in the N isotope composition





of $NO_3^-$ ($\delta^{15}N_{nitrate}$) from suboxic environments, where extensive $NO_2^-$ oxidation co-occurs with anammox (Brunner et al., 2013; Casciotti, 2016). Multiple factors influencing the net-effect of anammox on $\delta^{15}N$ are still under investigation (Kobayashi et al., 2019). However, since denitrification and anammox typically coexist in suboxic environments, isotope effects estimated from previous field studies should include both processes. For the purpose of this review, we will only refer to denitrification, but note that denitrification likely coexists with anammox. Because of the low-oxygen threshold for denitrification and anammox, the two processes only occur in ODZs.

In contrast to water column denitrification, sedimentary denitrification leads to little increase in the $\delta^{15}N_{nitrate}$ in the overlaying water column, because the isotopic discrimination is minimized by nearly complete consumption of $NO_3^-$ at the site of denitrification within sediment pore waters, leading to a small overall isotope effect (less than 3‰) (Sigman and Fripiat, 2019). In highly productive ocean margin environments such as subarctic and Arctic shelves, sedimentary denitrification has been reported to have a somewhat greater effect (up to 8‰) (Granger et al., 2004), albeit the effect on water column $NO_3^-$ remains small.

In summary, denitrification/anammox leaves a strong imprint on the isotopic composition of the residual $NO_3^-$ in the water column of ODZs (Cline and Kaplan, 1975; Liu and Kaplan, 1989). In those regions, isotopically enriched $NO_3^-$ can be upwelled and taken up by primary producers. The organic N produced in the surface ocean is exported, part of which is remineralized below the euphotic zone, spreading the high $\delta^{15}N$ signal of denitrification/anammox further away from the ODZs and suboxic environments through lateral transport (Sigman et al., 2009). The organic N that is not respired can carry the resulting isotopic enrichment into the underlying sediment and, if preserved well during sinking and burial, can be used as a recorder for changes in $\delta^{15}N_{nitrate}$ and thus changes in the extent of ODZs in the past (Altabet et al., 1995; Ganeshram et al., 1995).

The $\delta^{15}N$ of sedimentary archives is a semi-quantitative proxy for oxygen content in the water column. Several challenges must be overcome to quantitatively calibrate the proxy. This will require understanding of (i) uncertainties in the denitrification/anammox isotope effect, (ii) controls on denitrification rates, and (iii) influences of other processes such as $NO_3^-$ consumption and $N_2$ fixation on $\delta^{15}N_{nitrate}$. First, the current estimate of the isotope effect associated with denitrification/anammox has a large range. Earlier field-based estimates for the isotope effect often led to overestimation by comparing $NO_3^-$ from dysoxic environments with deep ocean $NO_3^-$ (Marconi et al., 2017). This can be improved with new analyses with the assistance of modelling and culture studies. Second, controls on the overall rate of denitrification in dysoxic environments are not fully constrained, although it is most likely tied to organic carbon supply (Ward et al., 2008). Field and modelling studies are needed to understand the relationship between the rate of denitrification/anammox, the size of the ODZs, and the oxygen content in the OMZ. Third, other biological processes occurring in the upper water column, especially $NO_3^-$ consumption and $N_2$ fixation, can affect $\delta^{15}N_{nitrate}$ when it is upwelled from dysoxic waters to the surface ocean (e.g., Farrell et al., 1995). The three major modern ODZs in the Eastern Tropical North and South Pacific, and the Arabian Sea are all





currently characterized by strong upwelling, high productivity, and in many cases, incomplete $NO_3^-$ utilization in the surface ocean. Because phytoplankton preferentially assimilate $^{14}N$ relative to $^{15}N$, incomplete utilization of surface $NO_3^-$ would elevate surface $\delta^{15}N_{nitrate}$ independent of changes associated with denitrification (e.g., Studer et al., 2021). In addition, upwelled

waters from dysoxic environments have lower N:P ratios due to N loss from denitrification. This may encourage biological $N_2$ fixation in the surface ocean, which brings in newly fixed N with lower $\delta^{15}N$ values (-1 to 1‰) (Sigman and Karsh, 2009; Ryabenko, 2013). These processes could modify the $\delta^{15}N$ of nitrate upwelled from dysoxic waters, especially in regions above and downstream of ODZ cores (e.g., Knapp et al., 2016; Wang et al., 2019). Careful selection of paleoceanographic study sites and modelling studies constrained by analyses of modern $\delta^{15}N_{nitrate}$ from dysoxic environments are needed to separate

influences from $NO_3^-$ consumption and $N_2$ fixation. Finally, various processes can influence the incorporation of the $\delta^{15}N$ signal into different organic N pools in the surface ocean and its preservation during sinking and burial. These processes are specific for each N archive, and will be discussed in the following sections.

### 6.4.2 Bulk sedimentary N isotopes

There is generally a good correlation between the upper ocean $\delta^{15}N_{nitrate}$, sinking particulate $\delta^{15}N$, and surface sediment $\delta^{15}N$

($\delta^{15}N_{bulk}$) in or near ODZs (Tesdal et al., 2013). As a result, $\delta^{15}N_{bulk}$ in marine sediments at or near an ODZ has been used to reconstruct changes in past water column denitrification and oxygen content, with most applications during the recent glacial/interglacial cycles (e.g., Altabet et al., 1995; Ganeshram et al., 1995; Galbraith et al., 2008; Galbraith, et al., 2013). Interpretations of $\delta^{15}N_{bulk}$ records rely on the assumption that the total N preserved in the sediments represent the total N generated and exported out of the surface ocean. This assumption has been challenged in two main aspects (Robinson et al.,

2012 and references therein). First, it has been widely observed that $\delta^{15}N_{bulk}$ is modified during sinking and incorporation into marine sediments. Sinking particles collected in the water column often show decreasing $\delta^{15}N$ with water depth (Altabet et al., 1991), which may be due to disaggregation of large particles into smaller particles, and thus infers a preferential loss of isotopically heavy N forms. Upon burial, microbial degradation processes may increase $\delta^{15}N_{bulk}$ in surface sediments with respect to sinking particles (by 2-5‰) (Altabet and Francois, 1994). Diagenetic effects on $\delta^{15}N_{bulk}$ have been considered small

in ODZs, because of low-oxygen content and high sedimentation rates, but this requires further validation. Second, $\delta^{15}N_{bulk}$ integrates the $\delta^{15}N$ signal of different N compounds in marine/terrestrial organic matter and clay-bound inorganic nitrogen. As such, it can be biased by the contribution of organic and inorganic N derived from terrestrial or distal marine (e.g., shelf) sources (Schubert and Calvert, 2001; Meckler et al., 2011). Thus, $\delta^{15}N_{bulk}$ is often used in combination with the total N content, carbon (C) to N ratio, and organic carbon isotope composition ($\delta^{13}C_{org}$) to quantify different endmember contributions using a

simple mixing model. However, uncertainties using these calculations can be significant. Other efforts have been made to measure the $\delta^{15}N$ of total sedimentary N and clay-bound inorganic N separately to calculate the $\delta^{15}N$ value of the organic N ($\delta^{15}N_{org}$) (e.g., Schubert and Calvert, 2001). Despite these challenges in applying and interpreting $\delta^{15}N_{bulk}$, it remains one of the most used $\delta^{15}N$ archives in ODZs, especially in sediment cores with lower carbonate content. It can be analysed relatively quickly and easily, and at a lower cost compared to other $\delta^{15}N$ archives.





### 6.4.3 Foraminiferal calcite-bound N isotopes

N isotopes in planktic foraminifera (FB-$\delta^{15}$N) have emerged as a novel approach to reconstruct past ocean oxygenation (Ren et al., 2009; Kast et al., 2019; Studer et al., 2021; Auderset et al., 2022; Wang et al., 2022). When foraminifera build their chambers, they form an organic sheet between calcite layers to facilitate the calcification process (Oscar et al., 2016). These organic sheets are mainly composed of proteins and polysaccharides, which are encased within the shells after calcification (Hemleben et al. 1989). Thus, foraminiferal-bound organic matter is protected by the calcite shells and less prone to diagenesis than bulk sediments. The amino-acid composition of the foraminiferal-bound organics of modern foraminifera appears to be very similar to that of fossil foraminifera from millions of years ago (King and Hare, 1972; Kast et al., 2019), suggesting there is little breakdown of more labile amino acids. Laboratory experiments have also shown that neither the N content nor its $\delta^{15}$N vary with oxidative degradation, fossil dissolution, and thermal alteration, confirming the robustness of FB-$\delta^{15}$N with respect to diagenesis (Martínez-García et al., 2022).

Planktic foraminifera incorporate N primarily by feeding on primary producers and zooplankton (Hemleben et al., 1989). Assimilation of $NO_3^-$ and $NH_4^+$ from the environment by photosynthesizing symbionts hosted in foraminifera could occur, but this is considered to be negligible in much of the oligotrophic oceans where the concentrations of both dissolved inorganic nitrogen (DIN) forms are low (Ren et al., 2012a, b). However, the symbionts could contribute to the internal N cycle by assimilating the recycled $NH_4^+$ excreted by the foraminifera host. This process may reduce the overall ammonium leakage and lower the expected trophic enrichment associated with it (Ren et al., 2012a, b). These processes propagate the $\delta^{15}N_{nitrate}$ signal through the food web into the biomass of foraminifera. It has been shown that modern planktic FB-$\delta^{15}$N provide a robust record of $\delta^{15}N_{nitrate}$ in thermocline waters (e.g., Ren et al., 2009, 2012a), which allows use of foraminifera from ODZ-influenced regions for the reconstruction of past ocean suboxia (e.g., Kast et al., 2019; Studer et al., 2021; Auderset et al., 2022; Wang et al., 2022).

### 6.4.4 Other potential archives for N isotopes in dysoxic environments

Diatom-bound $\delta^{15}$N ($\delta^{15}N_{db}$) is a potential, yet unexplored, proxy for past ocean oxygenation. Diatoms are siliceous phytoplankton that assimilate $NO_3^-$ as their main N source with an isotopic offset of roughly 5‰ (Waser et al., 1998). During biomineralization, a fraction of that biomass is incorporated into their silica shell, termed frustule, which is thought to be protective as in foraminifera tests (Sigman et al., 1999). While the exact controls on the isotopic fractionation between diatom biomass N and frustule-bound N are not yet fully explored, $\delta^{15}N_{db}$ has been shown to correlate with the $\delta^{15}$N of the $NO_3^-$ source to the diatoms (Horn et al., 2011; Jones et al., 2022). Since diatoms thrive in areas of high nutrient supply, most paleoceanographic applications of the $\delta^{15}N_{db}$ proxy have so far focused on the Southern Ocean and the Subarctic/Arctic oceans (Robinson and Sigman, 2008; Studer et al., 2012, 2015). While diatom opal also accumulates in the EEP upwelling region, diatoms are difficult to separate from radiolarians in those sediments, which can bias the $\delta^{15}N_{db}$ signal (Studer, 2013; Robinson



et al., 2015). Furthermore, incomplete $NO_3^-$ consumption in the EEP surface complicates the interpretation of $\delta^{15}N_{db}$ as an oxygen proxy, as it is influenced by both changes in denitrification and $NO_3^-$ assimilation. As such, $\delta^{15}N_{db}$ has the potential to record past ocean oxygenation only in regions of complete surface $NO_3^-$ consumption. Often, these oligotrophic environments are not hot-spots for diatom accumulation on the seafloor. Nonetheless, a recent global compilation of opal flux records indicates that sedimentary opal concentrations reach >10% in the northeastern Atlantic and equatorial Indian Ocean (Hayes et al., 2021). Those areas could be potential targets for future studies investigating past ocean oxygenation using diatom-bound N isotopes.

Most marine sediment records do not have the temporal resolution to capture the seasonal, annual or decadal climate variability required for direct comparison with historical observations. In contrast, reef-building, scleractinian corals are unique environmental archives that have been used extensively to reconstruct climate variability during past centuries at high resolution (Wang et al., 2016). Typical growth rates in most coral species used in paleoclimatic reconstructions are in the range of ∼1–2 cm per year, allowing monthly sampling resolution in most cases. Methods have been developed to analyse the $\delta^{15}N$ of coral skeleton-bound organic matter (CS-$\delta^{15}N$) requiring as little as 5 mg of coral carbonate (Wang et al., 2015). Modern ground-truthing has demonstrated that CS-$\delta^{15}N$ follows that of the $NO_3^-$ assimilated across a wide range of isotopic compositions and environmental settings (Wang et al., 2014, 2016). As a result, corals living close to the major ODZs are ideal candidates to study past changes in water column denitrification and ocean oxygenation at high temporal resolution. For example, a monthly-resolved CS-$\delta^{15}N$ record from the Oman margin shows values as high as 10-11‰, apparently recording the ODZs-sourced $NO_3^-$ signal in the Arabian Sea (Wang et al., 2016). Despite their slower growth rates, deep-sea corals (both scleractinian and proteinaceous) can also be used to investigate the marine N cycle and ocean deoxygenation in the past (McMahon et al., 2015; Sherwood et al., 2014; Wang et al., 2014).

### 6.4.5 Case study N-isotopes in the Eastern Tropical Pacific

An increase in overall denitrification rate may be expected to result from an increase in anoxic water volume, as a result of global warming and decline in the whole ocean oxygen concentration. Sedimentary $\delta^{15}N$ records could help constrain the various factors governing changes in water column denitrification rates and the oxygen content in the ocean. Below we discuss nitrogen isotope case study, focusing on the dynamics of the Eastern Tropical Pacific ODZ across different time intervals and climatic backgrounds.

While other Cenozoic records using $\delta^{15}N_{bulk}$ show nearly no changes (Algeo et al., 2014), Auderset et al. (2022) and Kast et al. (2019) found a decline in FB-$\delta^{15}N$ during prolonged warm periods indicating a reduction in water column denitrification and better ventilation of the Pacific ODZs (Fig. 7). Two possible mechanisms are proposed to explain the FB-$\delta^{15}N$ trend: (i) a reduction of equatorial upwelling as a result of reduced atmospheric pressure gradient during the warm periods and/or 2) a



decline in the biological pump efficiency resulting in less regenerated nutrients in the deep ocean. Both processes would lead
to a better ventilated EEP and reduction of the ODZs.

In addition, a FB-$\delta^{15}$N record from the Eastern Tropical South Pacific (Wang et al., 2022) showed a significant long-term
increase in $\delta^{15}$N since the Miocene (Fig. 7). A $\delta^{15}$N$_{bulk}$ record from the California margin (Liu et al., 2008) also showed an
increase since ~2.1 Ma, indicating ODZs expansion in the North Pacific. These records indicate expansion of the ODZs in the
Eastern Tropical Pacific, when global climate was cooling since the mid-Miocene (Herbert et al., 2016; Westerhold et al.,
2020). Although decreasing sea surface temperatures may have supplied more oxygen to the surface ocean since the late
Miocene, Wang et al., (2022) argue that the rising oceanic nutrient content and resulting higher productivity appear to have
overwhelmed the solubility effect and driven ocean deoxygenation over the past 8 Myrs.

On glacial/interglacial timescales, most published $\delta^{15}$N$_{bulk}$ records from the Eastern Tropical Pacific generally show lower
$\delta^{15}$N$_{bulk}$ values during the last ice age compared to the Holocene warm period, which has been interpreted as lower water
column denitrification and thus, higher oxygen content in the water column (Ganeshram et al., 1995; Galbraith et al., 2004;
Dubois et al., 2011). However, two recent FB-$\delta^{15}$N records from the EEP show similar $\delta^{15}$N values during the LGM and the
Holocene, indicating comparable water column oxygen content during those periods (Studer et al., 2021) (Fig. 7). The
difference between FB-$\delta^{15}$N and $\delta^{15}$N$_{bulk}$ may be attributed to lower diagenetic alteration and/or higher foreign N input during
ice ages which may have altered the $\delta^{15}$N$_{bulk}$ signal (Robinson et al., 2012; Studer et al., 2021). Using a box model, Studer et
al. (2021) argue that multiple processes may have stabilized the oxygen content. A glacial shoaling of the Atlantic meridional
overturning circulation, enhanced iron fertilization in the Subantarctic, and global cooling would have raised mid-depth oxygen
content, whereas a more efficient biological pump would have led to an accumulation of regenerated nutrients and thus a
decrease in deep Pacific oxygen content. This signal would have been mixed/upwelled into the mid-depth Pacific, leading to
little net LGM-to-Holocene change in the Eastern Tropical Pacific ODZ extent (Hain et al., 2010; Studer et al., 2021). The FB-
$\delta^{15}$N data are supported by the independent oxygen proxy I/Ca on planktic foraminifera (Hoogakker et al., 2018, Section 6.5)
and challenges the previous views on reduced Eastern Tropical Pacific water column denitrification during ice ages based on
$\delta^{15}$N$_{bulk}$ (Ganeshram et al., 1995; Galbraith et al., 2004).

By combining different $\delta^{15}$N records from the Eastern Tropical Pacific across various time scales, a novel hypothesis has
emerged that temperature-driven changes in mid-ocean oxygen content may not be the dominant control for ODZ evolution
and changes in water column denitrification rate. Instead, ocean circulation and biological activity are important additional
controls (e.g., Robinson et al., 2014). These applications highlight the need for multiple N proxy applications in the same
region or even the same sedimentary record, in combination with other oxygen proxies. In particular, we note an interesting
recurring pattern that $\delta^{15}$N$_{bulk}$ records tend to be similar to the FB-$\delta^{15}$N when $\delta^{15}$N and denitrification rate increase, but they
often fail to record $\delta^{15}$N decreases across a climate event (Fig. 7 and references therein). As we generally assume better





preservation conditions under low-oxygen conditions, the high $\delta^{15}N$ values recorded by the bulk sediment during intervals of decreased FB-$\delta^{15}N$ and increases in water column oxygen content could then be due to an increase in the diagenetic effect on

$\delta^{15}N_{bulk}$ (Robinson et al., 2012). The $\delta^{15}N$ difference between bulk sediment and fossil-bound N may in turn hold interesting information on past changes in oxygen content in the water column and sediments. Finally, as physical and biological processes are both important in understanding the history of the ODZs, an oxygen-realistic biogeochemical model embedded with nitrogen isotopes would be important to advance our interpretation of these $\delta^{15}N$ records.

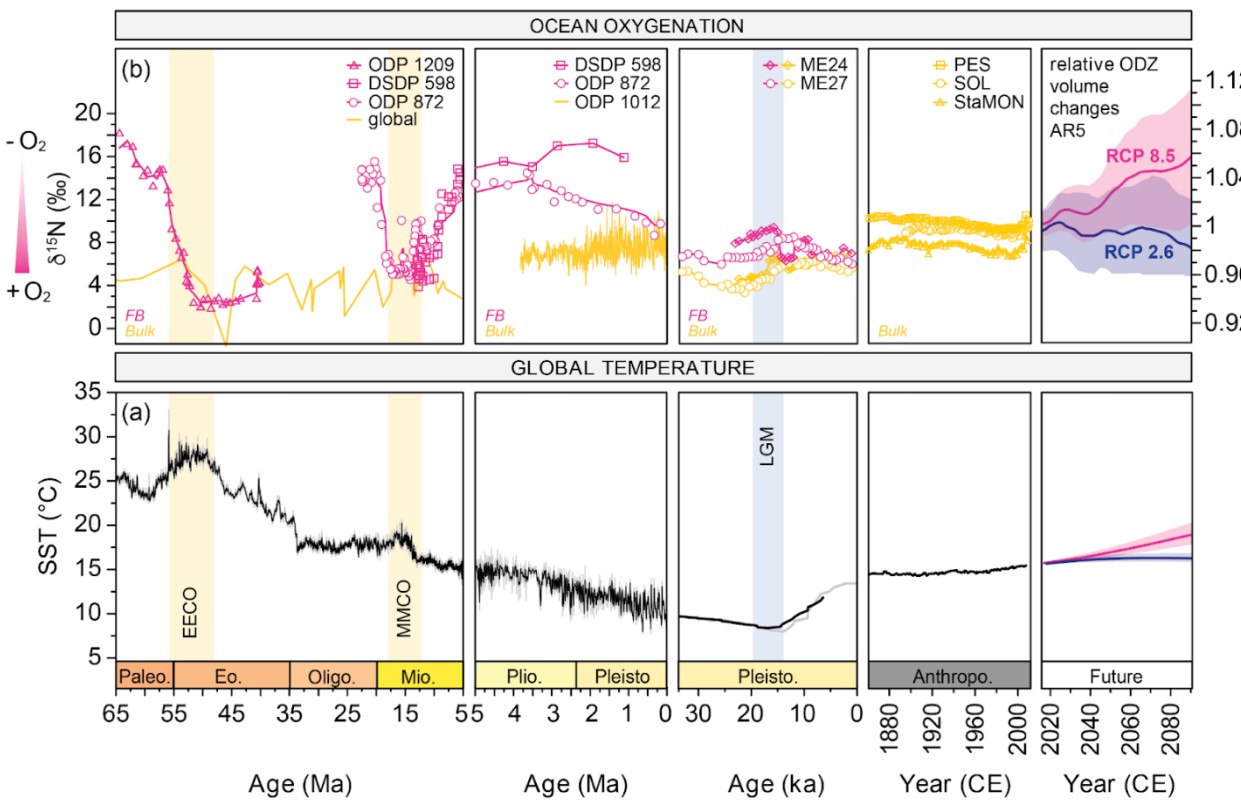


**Figure 7: Compilation of proxy-reconstructions of Equatorial Pacific anoxia from (FB-) $\delta^{15}N$. From left to right for the timespan 65–5 Ma, 5–0 Ma, 35–0 ka, 1860–2010 CE and simulated global ODZ volume for RCP2.6 and RCP8.5 projections 2020-2100 from IPCC AR5 (Bindoff et al., 2019). A) Sea surface temperature compilation by Hansen et al. (2013). B) Analyses of $\delta^{15}N$ in bulk sediment (yellow) and foraminifera-bound organics (FB; pink) in Pacific sedimentary archives. $\delta^{15}N$ is used as a qualitative oxygen-**

**sensitive proxy due to the strong isotopic fractions of denitrification by ODZ dwelling bacteria. Data compiled from the following sources: ODP 1209 (Kast et al., 2019); DSDP 598 (Wang et al., 2022); ODP 872 (Auderset et al., 2022); ODP 1012 (Liu et al., 2008); ME24 and ME27 (Studer et al., 2021; Dubois et al., 2011); PES, SOL, StaMON (Deutsch et al., 2014); global Paleocene-Miocene compilation from Algeo et al. (2014).**



## 6.5 Foraminifera trace elements

### 6.5.1 Introduction to proxy/geochemical system

#### 6.5.1.1 History of development of use

In addition to sedimentary trace elements, foraminifera trace elements can provide information about environmental redox conditions. In this section we will focus on foraminiferal I/Ca, Mn/Ca, and U/Ca, which are commonly used proxies to track seawater redox conditions and relative dissolved oxygen concentrations. Specifically, these proxies are thought to track dissolved iodate ($IO_3^-$), manganese (Mn(II)), and uranium (U(VI)), respectively. One frontier of the carbonate lattice-based proxies lies with determining the potential for their differential application to benthic and planktic foraminifera to quantify depth gradients in the water column. We discuss the proxy sensitivities within the context of the redox ladder, to avoid confusion with nomenclature (e.g., Canfield and Thamdrup, 2009). Below we discuss the distribution of $IO_3^-$, Mn(II), and U(VI) in the water column and pore waters, their incorporation into foraminiferal calcite, and evaluate their potential use as proxies for dissolved oxygen concentrations and seawater redox conditions (Fig. 8).





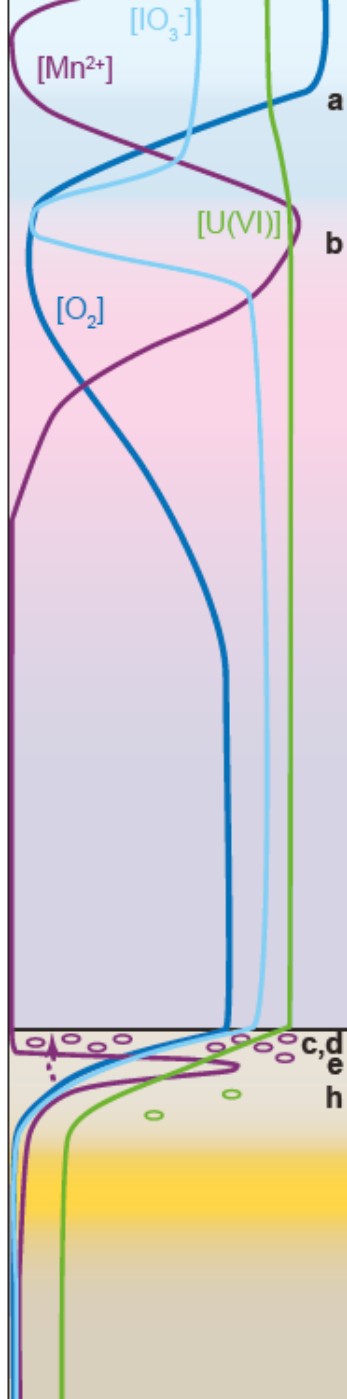

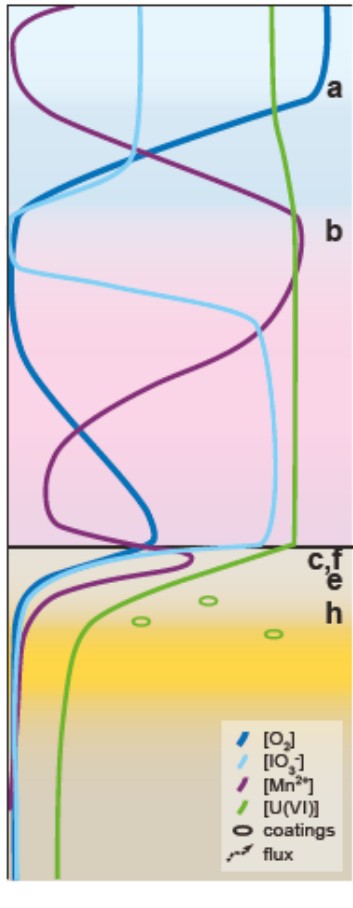

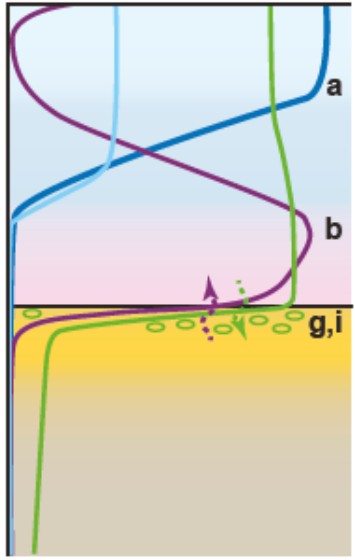

a: $IO_3^-$ incorporated in planktonic foramin-ifera. $I^-$ oxidation is relatively slow in ODZ waters mixing upwards, lower $[IO_3^-]$ above ODZ than in deep ocean

b: $IO_3^-$ reduction to $I^-$ in the upper ODZ

c: $IO_3^-$ reduction to $I^-$ in anoxic pore waters

d: $Mn^{2+}$ oxidation to Mn-Fe-(oxy)hydrox-ides, precipitation of Mn-carbonate coatings

e: Mn reduction to soluble $Mn^{2+}$, deep infaunal foraminifera incorporate Mn

f: Shallow $O_2$ penetration, shallower $Mn^{2+}$ front than in open ocean, deep and intermediate infaunal foraminifera incorporate Mn

g: Low-$[O_2]_{bottom\ water}$ allows $Mn^{2+}$ to escape sediments, less Mn for foraminifera to incorporate

h: U filtering in from overlying water column precipitates gradually in areas with deeper $O_2$ penetration

i: U filtering in from overlying water column precipirates rapidly in Fe-reduction zone





**Figure 8: Illustration of chemical profiles and key processes (a-i) affecting the I/Ca, Mn/Ca, and U/Ca in foraminifera oxygenation proxies in a theoretical transect through an ODZ from A) a deep ocean site with overlying ODZ to B) an intermediate-depth ODZ-adjacent site to C) a shallow-water site with sediments in the ODZ. Based on data and concepts from Anderson et al. (1989), Thamdrup et al. (1994), Morford et al. (2009), Singh et al. (2011), Koho et al. (2015), Mariyasu et al. (2020), and Singh et al. (2023).**

### 6.5.1.2 Seawater elemental cycling tracked via paleoredox proxies

Multiple published seawater transects from the Eastern Tropical North and South Pacific document the geochemistry of Mn and I (Rue et al., 1997; Cutter et al., 2018; Rapp et al., 2020; Moriyasu et al., 2020, 2023). These studies demonstrate similarities and differences in seawater I and Mn speciation that can be exploited for defining water column redox conditions. For example, redox thresholds for $IO_3^-$ and Mn-oxide reduction overlap (Fig. 8), and both display potentially rapid reduction kinetics, sluggish oxidation kinetics, and an important benthic component of accumulation within ODZs (Cutter et al., 2018;

Rue et al., 1997). Specifically, in ODZs, dissolved iodate ($IO_3^-$) is reduced to dissolved iodide ($I^-$) at and below the oxycline (Fig 6.5.1 areas labelled b). Estimates of the iodate oxygen reduction threshold range from up to 50 μmol kg$^{-1}$ (Lu et al., 2020) to potentially less than 1 μmol kg$^{-1}$ (Hardisty et al., 2021). Thresholds less than 1 μmol kg$^{-1}$ may be possible, but iodine cycling has yet to be evaluated alongside oxygen sensors with sub-μM detection limits (e.g., Thamdrup et al., 2012). Regardless, the lowest $IO_3^-$ has been demonstrated to nearly exclusively occur in low-oxygen waters, thus, defining a threshold for 'iodinous'

conditions (Fig. 8) (Hardisty et al., 2021).

At similar water column depths to where $IO_3^-$ reduction occurs in ODZ cores (i.e., oxygen minima), dissolved Mn(II) also begins to accumulate (Fig. 8). Mn(II) is formed through the reduction of suspended manganese oxyhydroxides (III, IV) minerals driven by organic matter mineralisation and/or oxidation of reduced components (e.g., dissolved iron, sulphides, and

ammonium, Burdige, 1993; Thamdrup et al., 1994a, 1994b; Kristiansen et al., 2002). Potential oxygen thresholds for Mn accumulation have been evaluated using Switchable Trace Oxygen Sensor (STOX) microsensors, which indicate that Mn(II) may be re-oxidized at oxygen levels as low as 100 nM (Clement et al., 2009). Ultimately, the accumulation of both Mn(II) and $I^-$ are well-defined features in ODZs, and these are increasingly used alongside $NO_2^-$ (product of nitrate reduction) maxima to define ODZ cores and so-called 'functional anoxia', which are zones where these and potentially more reducing metabolisms

such as $SO_4^{2-}$ reduction may occur within micro-niches (e.g., Canfield et al., 2010; Raven et al., 2018).

Uranium cycling is a useful comparison to I and Mn cycling in that it is sensitive to more reducing ferruginous conditions (Fig. 1.1). U(VI) is dissolved in oxygenated seawater as uranyl carbonate (U(VI)) complexes and behaves conservatively (Ku et al., 1977; Chen et al., 1986; Dunk et al., 2002; Not et al., 2012). Uranium has a long residence time in seawater (~400 kyr), and therefore its concentration is homogenous and relatively constant on timescales of $10^5$ yrs (Dunk et al., 2002). While uranium

exhibits conservative behaviour in oxic settings, in oxygen-deficient/ ferruginous seawater conditions, characteristic of some ODZ bottom waters, U(VI) is reduced to insoluble U(IV) and may precipitate on settling marine particles. Some of these U-containing particles ultimately reach the sediment to either be scavenged or re-oxidized to a soluble uranyl-carbonate complex



at the sediment/water interface (Anderson et al., 1989). Under sufficiently reducing bottom- and pore water redox conditions (dysoxic to anoxic), U will precipitate in the sediments as uraninite (U(IV)O$_{2(s)}$; Fig. 8 areas labelled i).


Reduction of IO$_3^-$, Mn-oxide, and U(VI) takes place through bacteria catalysation. Reduction of IO$_3^-$ has been associated with IdRA genes (Uyamas et al., 2021). Soluble U(VI) is most likely reduced to insoluble uraninite U(IV) by iron-reducing bacteria (e.g., Lovley et al., 1991). Abiotic reduction is important as well. I and Mn are rapidly reduced via abiotic reactions with Fe and sulphide (Luther et al., 2023). Because ferruginous and sulphidic conditions are common in modern sediments, rapid

reduction of IO$_3^-$, Mn-oxides, and U(VI) creates large gradients that can control diffusive fluxes to and from seawater. For IO$_3^-$ and Mn-oxides in sediments, rapid reduction creates elevated [I$^-$] and [Mn(II)]—which can be further exacerbated for [I$^-$] due to organic matter remineralization (Kennedy and Elderfield, 1987a; 1987b) - that drives benthic fluxes into the overlying seawater (Fig. 8 areas labelled c and g). These fluxes are large within ODZ water columns, where low-oxygen concentrations enable the persistence of the reduced I$^-$ (e.g., Cutter et al., 2018; Moriyasu et al., 2020) and Mn(II) (e.g., Froelich et al., 1979;

Sundby and Silverberg, 1985; Metzger et al., 2007; Mouret et al., 2009) (Fig. 8). U behaves the opposite, where the formation of insoluble U(IV) under ferruginous conditions removes U from pore waters. This causes a concentration gradient to form between high [U] overlying bottom waters and low [U] pore waters and leads to a diffusive flux of seawater U into the sediments (Barnes and Cochran, 1990; Klinkhammer and Palmer, 1991), causing sediment authigenic U (aU) enrichment, at a rate established by the diffusive flux (Fig. 8 area labelled i).

**6.5.1.3 Incorporation: how, when, and where are elements incorporated?**

The I/Ca, Mn/Ca, and U/Ca proxies track iodinous, manganous, and ferruginous conditions respectively (Fig. 8). Below we consider what is known for each proxy. Importantly, recent studies indicate that, whether applied to planktic or benthic foraminifera, each proxy may, at least in part, reflect the geochemistry of bottom waters, but more work is needed.

It has been demonstrated that only the oxidized I species, IO$_3^-$, can be incorporated into both abiotic calcite (Lu et al., 2010; Zhou et al., 2014; Podder et al., 2017; Kerisit et al., 2018) and dolomite (Hashim et al., 2022) (Fig. 8 inset labelled a). Therefore, the I/Ca of planktic and benthic foraminiferal calcite has traditionally been used to infer [IO$_3^-$] as a proxy for changes in subsurface and bottom water oxygen concentrations, respectively (Glock et al., 2014; 2016; Hardisty et al., 2014; 2017; Hoogakker et al., 2018; Lu et al., 2010; 2016; 2020; Zhou et al., 2014; 2016). However, the incorporation of IO$_3^-$ has not been

directly tested within foraminifera, which may include vital effects not currently recognized. A recent study found that I/Ca values of planktic foraminifera sampled from plankton tows showed little-to-no relationship with the dissolved [IO$_3^-$] of ambient seawater (Winkelbauer et al., 2023). In fact, this work showed that planktic foraminiferal I/Ca was about ten times lower in plankton tows compared to that in sediment core-tops and as would have been expected from abiotic calcite precipitation experiments (Winkelbauer et al., 2023). Winkelbauer et al. (2023) suggest that planktic foraminifera may gain

iodine during gametogenesis or post-mortem, either when falling through the water column, or through burial. Thus, core-top





and downcore planktic foraminiferal I/Ca may be representative of an integrated $IO_3^-$ signal from across the water column and sediment, instead of the depth that they occupy during their life cycle.

While both benthic and planktic foraminiferal I/Ca data from core-top samples support a relationship with bottom and subsurface water dissolved oxygen (Fig. 9), the species-specific and/or mineralogical controls for incorporation of $IO_3^-$ into biogenic carbonates are still not well-understood. Globally, the highest core-top I/Ca values are found to be ~9 µmol mol$^{-1}$ for planktic foraminifera in the Walvis Ridge region (Lu et al., 2020) and ~22 µmol mol$^{-1}$ for benthic foraminifera in Little Bahamas Bank region (Lu et al., 2022), both from well-oxygenated environments. Assuming the $IO_3^-$ concentrations in these oxic waters range between 0.5 and 0.65 µmol L$^{-1}$, the partition coefficient (Kd) for $IO_3^-$ incorporation (Kd = $[I/Ca]_{foram}$ / $[IO_3^-]_{sw}$, with a unit of [µmol mol$^{-1}$]/[µmol L$^{-1}$]) can range from 14 to 44, or even higher if the seawater $IO_3^-$ concentration is lower

than 0.5 µmol. These Kd estimates are much higher than those reported in abiotic calcite synthesis experiments (~10) (Lu et al., 2010), suggesting a potential biological control on the I incorporation in the calcite. The strong association of iodine with organic heterogeneities in the calcite of benthic foraminifera might be another challenge to consider within future studies on foraminiferal I/Ca (Glock et al., 2019).


Interpretations of redox-conditions based on the foraminiferal Mn/Ca proxy may be derived from calcite lattice-bound Mn ($Mn/Ca_{foram}$) and Mn bound in post-depositional authigenic coatings of foraminifera tests (e.g., Barras et al., 2017; Chen et al., 2017). Foraminifera can incorporate soluble Mn(II) into their calcite tests (Fig. 8 inset labelled e). Barras et al. (2018) observed a linear correlation between $Mn/Ca_{sw}$ and $Mn/Ca_{foram}$ for two different species (*Ammonia* T6 and *Bulimina marginata*) of

benthic foraminifera in controlled laboratory conditions. Although it seems that the partition coefficient increases when concentrations are lower than ~10 µmol L$^{-1}$ of $Mn_{sw}$. Because of the link between Mn(II) and ODZs, Mn/Ca in benthic foraminifera has been linked to dissolved oxygen in the bottom and/or pore waters of their microhabitat (e.g., Klinkhammer et al., 2009; Koho et al., 2015). Concentrations of benthic foraminiferal (lattice-bound) Mn/Ca from OMZ or low-oxygen environment and culture experiments can be in a range of 0.1 to >150 µmol mol$^{-1}$ (Lea, 2003; Glock et al., 2012; Koho et al.,

2017; Barras et al., 2018; Brinkmann et al., 2021) depending on oxygen in bottom/pore waters. Planktic foraminiferal (lattice-bound) Mn/Ca ratios from plankton tows and sediment traps also seem to be linked to seawater oxygen, with higher Mn/Ca relating to lower oxygen (Steinhardt et al., 2014; Davis et al., 2023). The advantage of foraminiferal calcite-bound Mn/Ca ratios compared to a bulk sediment proxy such as Mn/Al is that once precipitated, the Mn concentration remains fixed in the foraminiferal shell and should not be subject to diagenetic reduction or oxidation (Koho et al., 2015; McKay et al., 2015).


Nevertheless, in the case of fossil tests, post-mortem Mn-rich contaminant secondary coatings (e.g., Mn oxides or Mn carbonate; Fig. 8 inset labelled d) may obscure the Mn/Ca signal of the pristine calcite signal (Barker et al., 2003; Ni et al., 2020). Authigenic Mn mineral formation can occur on the outside and/or inside of the foraminiferal tests and pores. Recrystallization or banding within foraminiferal test laminations can interfere with the application of this proxy for the



reconstructing redox conditions when the foraminifera were formed (e.g., Detlef et al., 2020; Ni et al., 2020). In addition to the primary foraminiferal Mn proxy, several studies suggest authigenic foraminiferal U/Mn in coatings as a proxy for sedimentary post-deposit redox conditions (Gottschalk et al., 2016; Chen et al., 2017; Detlef et al., 2020). The formation of Mn-rich authigenic carbonates potentially responds to the microbial activity in the pore water which is linked to the sedimentary redox environment (Detlef et al., 2020; Ni et al., 2020).


The U/Ca proxy does not target carbonate lattice-bound U. Instead, this utilizes the formation of aU which precipitates onto foraminifera tests buried in marine sediments, forming a U-rich (post-depositional) coating on their carbonate tests (Boiteau et al., 2012) (Fig. 8 area labelled i). The rate of authigenic enrichment is established by the U diffusive flux between overlying bottom waters and pore waters and follows similar dynamics to aU precipitation in bulk sediments (Boiteau et al., 2012). The

diffusive flux, in turn, depends on how reducing the conditions are within the sediments (Barnes and Cochran, 1990; Klinkhammer and Palmer, 1991). Therefore, higher U/Ca concentrations are indicative of reducing oceanic bottom water conditions.

Concentrations of foraminiferal U/Ca can reach 300-700 nmol mol$^{-1}$ (Boiteau et al., 2012; Gottschalk et al., 2016; 2020;

Skinner et al., 2019; Chen et al., 2017). This exceeds the foraminiferal lattice-bound [U], with shell matrix U/Ca ranging from ~1-23 nmol mol$^{-1}$ (Russell et al., 1994; Raitzsch et al., 2011; Boiteau et al., 2012; Yu et al., 2008; Chen et al., 2017). Therefore, lattice-bound U has a negligible impact on the measured U/Ca values of a diagenetically altered shell. Furthermore, the post-depositional accumulation of aU as foraminiferal authigenic coating means that any species can be measured, including planktic foraminifera which tend to be much more abundant in open ocean settings.

**6.5.1.4 Additional impacts on proxy values**

Preservation and diagenetic effects on I/Ca ratios from biogenic calcite in sediments is currently unexplored, although it is well-known that I/Ca in bulk carbonate is susceptible to diagenetic alterations, specifically declining values due to diagenesis in reducing IO$_3^-$ free pore waters (Hardisty et al., 2017; Lau and Hardisty, 2022). Diagenetic impacts on local IO$_3^-$ availability in pore fluids relative to overlying seawater may also make benthic foraminiferal signals particularly susceptible to recording

diagenetic conditions. Specifically, a combination of excess I, related to I input from organic remineralization, and reducing conditions which can impact IO$_3^-$ availability, are possible. For example, I/Ca values as high as 20 µmol mol$^{-1}$ have been observed in infaunal foraminifera from the PETM. This may be due to higher IO$_3^-$ near the sediment-water interface driven by oxic organic remineralization (Zhou et al., 2016; Kennedy and Elderfield, 1987a,b), or higher-than-modern total iodine concentrations in the seawater during the PETM. In the modern Peruvian OMZ, shallow infaunal species (*Uvigerina striata*

and *U. peregrina*) show I/Ca values ~1 µmol mol$^{-1}$ lower than epifaunal species (*Planulina limbata*) (Glock et al., 2014). If oxygen was the only control on pore water IO$_3^-$, these lower I/Ca values in modern infaunal species would reflect lower IO$_3^-$ concentrations in pore water, linking to rapid oxygen decrease within a few centimetres or millimetres of sediments. Thus, it



remains unclear how pore water IO$_3^-$ may influence I/Ca of infaunal foraminifera. Attention should be given to possible contamination through organic bound iodine in foraminiferal calcite (Glock et al., 2016, 2019), which might significantly

impact measured I/Ca ratios, thus requiring intense oxidative cleaning. Additional oxidative cleaning steps can result in considerably lower I/Ca ratios (Winkelbauer et al., 2021, 2023; see also Fig. 9).

Like many other foraminiferal trace element proxies, several additional environmental parameters (e.g., temperature, salinity, and carbonate ion concentrations) may impact the elemental incorporation as well. In the Little Bahama Bank region where bottom water oxygen concentrations are similarly high between 150 and 200 µmol kg$^{-1}$, seawater temperature, salinity, and

carbonate ion concentration show negative correlation with benthic foraminifera I/Ca (Lu et al., 2022). It is thus speculated that benthic foraminifera may preferentially incorporate IO$_3^-$ at lower temperature and/or lower salinity. Additionally, or alternatively, when ambient seawater has less carbonate ion availability, IO$_3^-$ may be used as an alternative substitute for calcite structure by foraminifera. It is not clear whether the negative correlation between I/Ca and temperature could still be found under lower bottom water oxygen conditions, as bottom water oxygen is often anti-correlated with temperature. Further studies

are needed to disentangle such effects. Lastly, one epifaunal species, aragonitic *Hoeglundina elegans,* shows lower I/Ca values than observed in other benthic foraminifera (*Planulina limbata*, *Uvigerina peregrina* and *Uvigerina striata*, Glock et al., 2014), suggesting different IO$_3^-$ incorporation mechanisms for differing mineralogies (e.g. aragonite versus calcite), or an effect of different microhabitat preferences. Future work is needed to clarify these differences.

Factors such as carbonate chemistry, metabolic effects, ontogenetic effects, and species-specific effects could also have potential impacts on Mn incorporation into the foraminiferal test. Culture experiments with different partial pressure of carbon dioxide (*p*CO$_2$) levels show Mn/Ca of larger benthic foraminifera increased under high *p*CO$_2$ conditions, which can be mainly ascribed to Mn speciation changes in seawater for Mn incorporation (van Dijk et al., 2020). Mn incorporation can also be affected by Mg incorporation in hyaline species. Mg and Mn are coupled during foraminiferal calcification and are correlated

on specimen and species level (van Dijk et al., 2020). In the case of symbiont-bearing species grown under low-oxygen conditions, Mn mapping using electron probe microanalysis (EPMA) on the cross-section of the test highlighted that layers of calcite are enriched with Mn compared to ones grown under high-oxygen conditions (van Dijk et al., 2019). This may be caused by the influence of the day/night (light/dark) cycles, meaning that symbiont activity (photosynthesis/respiration) or other diel shifts in physiology may directly or indirectly impact Mn concentration/speciation at the site of calcification.

Different chamber-to-chamber trends (such as between proloculus and last chambers) for *Ammonia* and *Bulimina* species show ontogenetic effects (Barras et al, 2018, Brinkmann et al., 2023). Ontogeny-driven (life strategy) preferences may influence Mn/Ca in initial chambers (incl. proloculus) of *Bulimina*, as indicated by in-field observations (Brinkmann et al., 2023). Species-specific biomineralization processes and microhabitat effects could also impact Mn incorporation in small benthic foraminifera (Koho et al., 2015; Barras et al, 2018, Brinkmann et al., 2023, Groeneveld and Filipsson, 2013; Groeneveld et al.,

2018). Culture experiments demonstrate species-specific ontogenetic effects on Mn/Ca with opposite chamber-to-chamber trends in the last three chambers of *Ammonia* and *Bulimina* species (Barras et al., 2018). Diagenetic effects, including



secondary mineral coatings, can significantly interfere with the Mn/Ca measurements of primary calcite under hypoxic/anoxic burial conditions (e.g., Detlef et al., 2020; Ni et al., 2020). The formation of inorganic carbonates with highly elevated Mn on the internal and external surfaces of foraminiferal tests, and especially in the pores, is difficult to eliminate through standard foraminiferal trace element cleaning procedures (Ni et al., 2020).

By contrast, post-depositional alterations form the basis of the U/Ca proxy. Authigenic U accumulated in sediments may be remobilized, due to a deepening of the anoxic boundary, driven by an increase in bottom water oxygen, decreased $C_{org}$ flux, or bioturbation (McManus et al., 2005; Zheng et al., 2002). Such a change in oxygen penetration depth could lead to a reversed pore-to-bottom water [U] gradient, causing an efflux (remobilization) of dissolved U from the sediments back into the overlying bottom water, therefore eliminating the primary U signal (McManus et al., 2005; Zheng et al., 2002). It has been suggested that the U/Ca of foraminiferal coatings from marine sediment cores with high sedimentation rates ultimately record aU formed in steady state with bottom water, and that the diagenetic aU loss is minimal (Gottschalk et al., 2020; Jacobel et al., 2020).

### 6.5.2 How does the proxy relate back to oxygen?

Quantitatively relating proxies to oxygen content is an important goal for understanding paleo-redox evolution. This is extremely challenging because each proxy has variable oxygen thresholds and vital, mineralogical, and diagenetic effects that can distort its signal in the geologic record. Empirical comparisons of plankton tow and core-top proxy values to subsurface or bottom water oxygen can provide important constraints on proxy-oxygen relationships applicable to the geologic record. Toward this goal, Figures 6.5.2, 6.5.3, and 6.5.4 provide proxy-oxygen syntheses, and below we provide both mechanistic and empirical discussions for relating proxy records back to specific oxygen levels. For each of the proxies this means understanding at least two tiers of oxygen relationships: oxygen values allowing for an initial change in the proxy from baseline conditions and subsequent scaling of changing proxy values to dynamic oxygen conditions.

I/Ca values recorded in planktic and benthic foraminifera are lower in areas with lower subsurface and bottom water oxygen, respectively (Fig. 9-11 and references therein). Of the three foraminiferal proxies discussed here, I/Ca has the best empirical constraints on oxygen thresholds, but at the same time the mechanistic understanding of factors driving these relationships remain unclear (see Section 6.5.1). Recent studies provide two ways to interpret I/Ca relative to oxygen content or redox conditions. The first is the simple presence/absence of carbonate-associated iodine, and hence the presence/absence of $IO_3^-$. Iodide ($I^-$) and $IO_3^-$ have a similar redox potential and thus $IO_3^-$ is quantitatively reduced to $I^-$ in productive anoxic settings. This implies that the simple presence of $IO_3^-$ or carbonate-associated iodine may be indicative of oxygen at some level (e.g., Hardisty et al., 2014). We note that a global compilation of $IO_3^-$ in ODZs demonstrates that $IO_3^-$ may persist when oxygen is below detection; however, this is interpreted to reflect sluggish reduction of $IO_3^-$, not in situ $IO_3^-$ production (Hardisty et al., 2021; Moriyasu et al., 2020; Cutter et al., 2018). Also, as discussed earlier, the CTD oxygen sensor detection limits, which are



typically near 1 μmol kg⁻¹, are currently a limitation for understanding $IO_3^-$-oxygen thresholds, as nmol kg⁻¹ oxygen levels may support active I redox cycles, as has been demonstrated for N and Mn (Clement et al., 2009; Thamdrup et al., 2012).

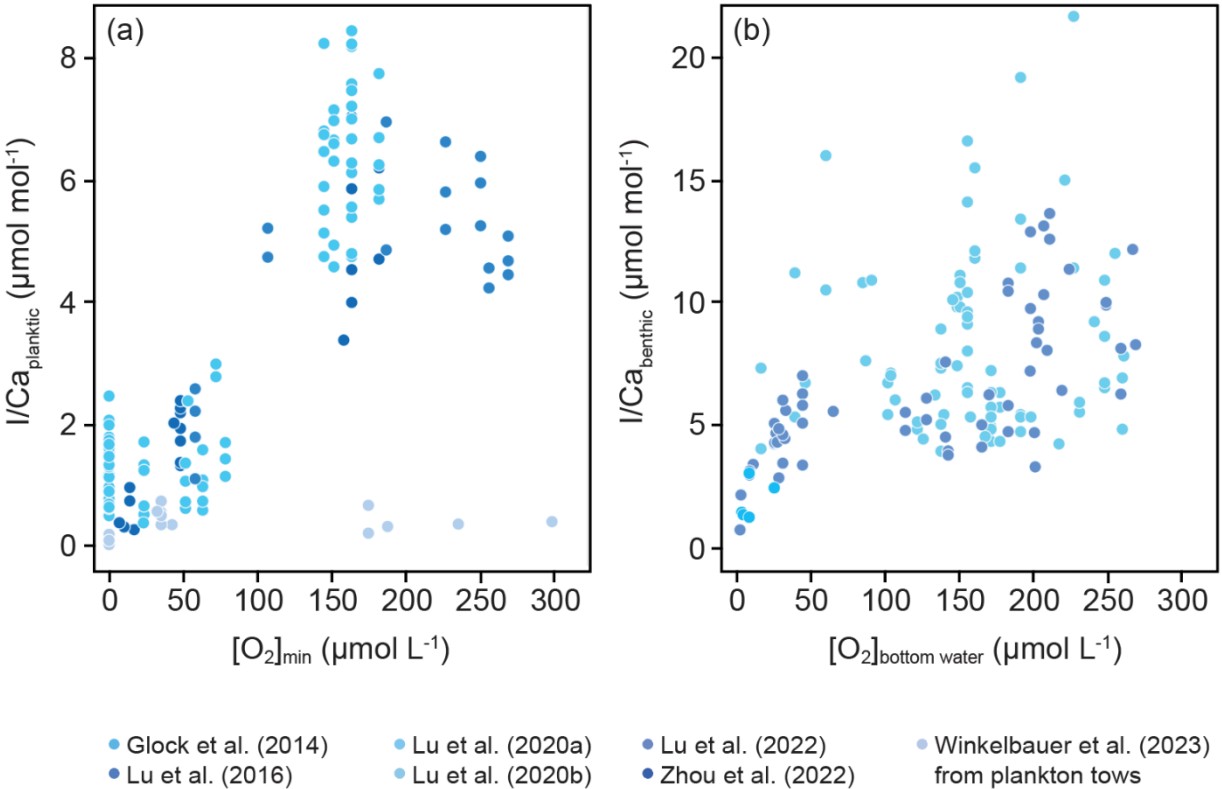

**Figure 9: I/Ca from core-top (top 10 cm) and plankton tow foraminifera compared to oxygenation. A) I/Ca in planktic foraminifera versus minimum oxygen concentration in the water column. Zhou et al. (2023) data were corrected for reductive cleaning. B) I/Ca in epifaunal and infaunal benthic foraminifera versus bottom water oxygen concentration.**

A major challenge in interpreting I/Ca data quantitatively is that the oxidant(s) responsible for I⁻ oxidation to $IO_3^-$ is unknown. Indeed, unambiguous *in situ* $IO_3^-$ production has not been observed under normal marine conditions. Oxygen is not directly responsible for $IO_3^-$ formation, at least abiotically, as demonstrated by the long residence time - estimated to range from 40 to
1715 <0.5 yrs - of I⁻ in oxygenated seawater (Chance et al., 2014). Some recent culture studies propose that superoxide or ammonia-oxidizing bacteria may be responsible for catalysing $IO_3^-$ production, but results have yet to be confirmed in natural marine settings or in cultures without iodine in excess of seawater (Hughes et al., 2021; Li et al., 2014). While I⁻ oxidation is a limitation for models used to interpret mechanisms and distributions of ancient $IO_3^-$, constrained via I/Ca (e.g., Lu et al., 2018), $IO_3^-$ reduction is clearly linked to declining oxygen, thus bolstering proxy applications.

Beyond presence/absence, I/Ca values above and below a threshold range may be used to define 'iodinous' conditions typical of ODZs. The 'iodinous' framework allows for I/Ca interpretations in the context of the redox ladder alongside other proxies



reflecting specific reduction potentials (Canfield and Thamdrup, 2009; Lau and Hardisty, 2022; Algeo and Li, 2020) (Fig. 1). I/Ca < 3 µmol mol$^{-1}$ can be related back to the IO$_3^-$ range <300 nM common to ODZ settings (Lu et al., 2016; Hardisty et al., 2021; Lu et al., 2022). Lastly, specific oxygen thresholds have been recommended for the recognition of 'iodinous' conditions. I/Ca values <3 µmol mol$^{-1}$ have been demonstrated in benthic foraminifera with a bottom water oxygen concentration <50 µmol kg$^{-1}$ (Fig. 8; Lu et al., 2022). I/Ca variations>3 µmol mol$^{-1}$ are unlikely directly related to oxygen, but instead likely reflect combinations of biologically mediated transformations during primary production and physical mixing and advection processes (Chance et al., 2014; Campos et al., 1996; Truesdale, 2000; Hepach et al., 2020).

The application of Mn/Ca is mainly limited to trace bottom water oxygen conditions. Whilst water column Mn cycling systematics are well understood, the direct relationship of Mn/Ca values to specific oxygen levels is restricted in comparison to I. Fundamentally, the highest Mn/Ca are found in benthic foraminifera from manganous environments, allowing for high dissolved Mn$^{2+}$ beneath intermediately oxic water columns that limit benthic Mn fluxes out of the sediments (Fig. 10).This is because Mn/Ca tracks the reduced Mn endmember, contrarily to I/Ca, and because abiotic and benthic cycling can contribute to Mn(II) accumulation. For Mn, the diffusion of dissolved Mn from pore waters into bottom water, under prolonged anoxic conditions, prevents a linear relationship between Mn/Ca in foraminiferal calcite and bottom water oxygen concentrations (Koho et al., 2015). Recently, Brinkmann et al. (2023) found that an upper limit of around 130 µmol L$^{-1}$ exists in a fjord setting above which the linear correlation between foraminiferal Mn/Ca and bottom water oxygen no longer exists, confirming earlier work by Guo et al. (2019) from the Yangtze River Estuary. This proxy would also be difficult to apply in environments with very low dissolved Mn content as is the case for example in the Santa Barbara Basin (Brinkmann et al., 2021). In the case of a depleted Mn pool in the sediment, the changes in Mn speciation according to oxygen concentrations would not be significant enough to be recorded in the foraminiferal shell. On the other hand, the vicinity of continental inputs or other sources of Mn into the sediment, independent from oxygen conditions, could also hamper the proxy robustness (Klinkhammer et al., 2009).



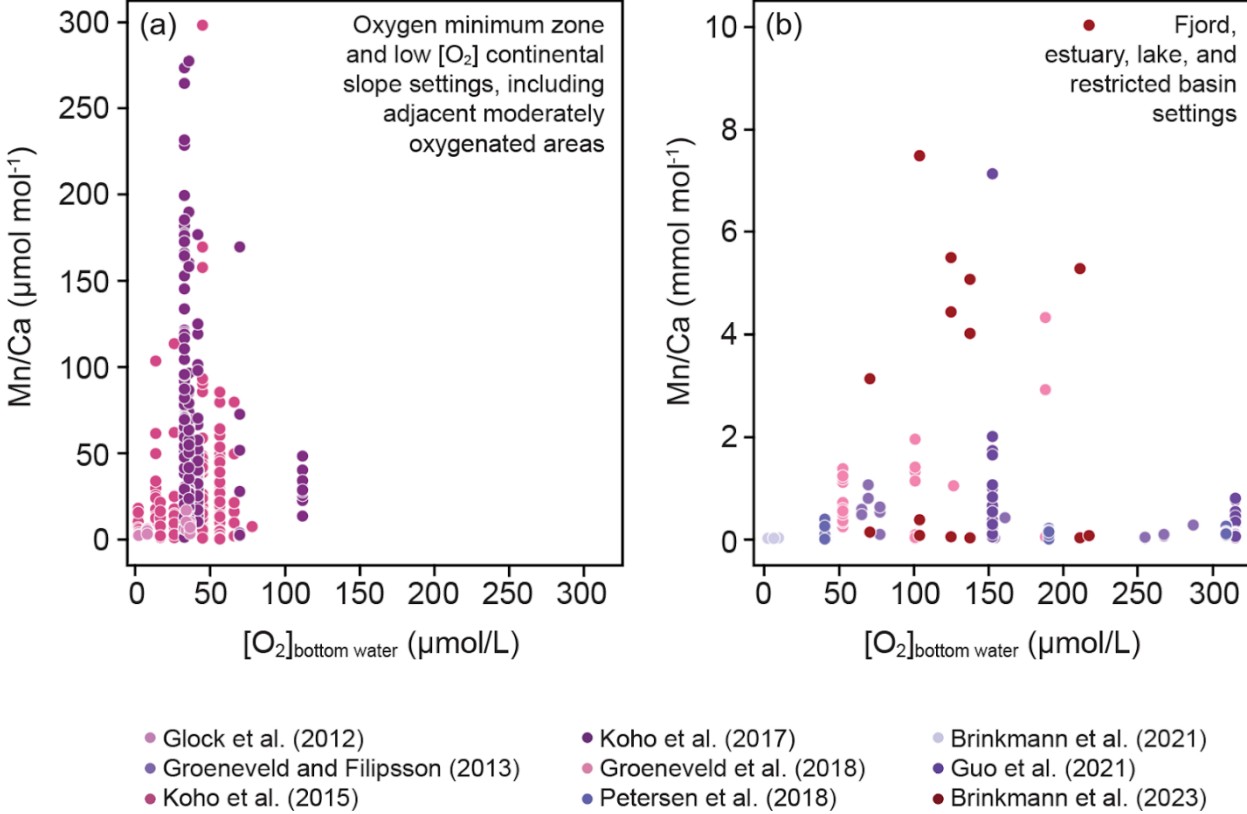


**Figure 10: Mn/Ca data from core-top benthic foraminifera compared to oxygenation in different settings. Note the differences in scale between A and B. A) OMZ and low-oxygen continental slope and B) fjord, estuary, lake, and restricted basin settings. Note two-tailed distribution from right to left, with (1) low Mn/Ca values where a deeper oxycline in pore waters results in less Mn$^{2+}$ available at foraminifera depth habitats, (2) high Mn/Ca values where low-oxygen bottom water prevents Mn$^{2+}$ from leaving pore**
**waters, and (3) low Mn/Ca where Mn$^{2+}$ is lost to water column with low-oxygen bottom water or precipitated as Mn-carbonate.**

The Mn/Ca proxy is best related to relative oxygen levels, with highest Mn/Ca encountered under hypoxic conditions. The Mn/Ca proxy will reflect the oxygen conditions in the microenvironment surrounding the foraminifera during calcification, i.e., bottom water, pore water, or water column conditions in the case of planktic foraminifera. Benthic foraminifera species
considered as epifaunal or shallow infaunal should therefore better record bottom water conditions than intermediate and deep infaunal taxa. Moreover, under oxic conditions epifaunal species would incorporate less Mn. With decreasing bottom water oxygen concentrations, the redox boundary would migrate towards the sediment water interface and Mn incorporation would increase for the shallow infaunal species. However, under anoxic/hypoxic bottom water conditions, dissolved Mn diffuses out of the sediment resulting in less free Mn available for incorporation in the foraminiferal shell (e.g., Groeneveld et al., 2018).
Intermediate and deep infaunal taxa are already living at or below the redox boundary and could migrate in the sediment accordingly, potentially changing their microhabitat.





Similar to Mn/Ca, the U/Ca proxy is best related to relative increases/decreases in oxygen, rather than absolute oxygen values. Higher U/Ca concentrations in foraminifera indicate more reducing sedimentary conditions, which are driven by low-oxygen concentrations in bottom and pore waters (Fig. 11). However, these are driven by multiple processes of physical or biological nature. Specifically, high organic C fluxes can lead to low-oxygen environments in pore waters and bottom waters, which ultimately causes the in-situ precipitation of aU as foraminiferal coatings.

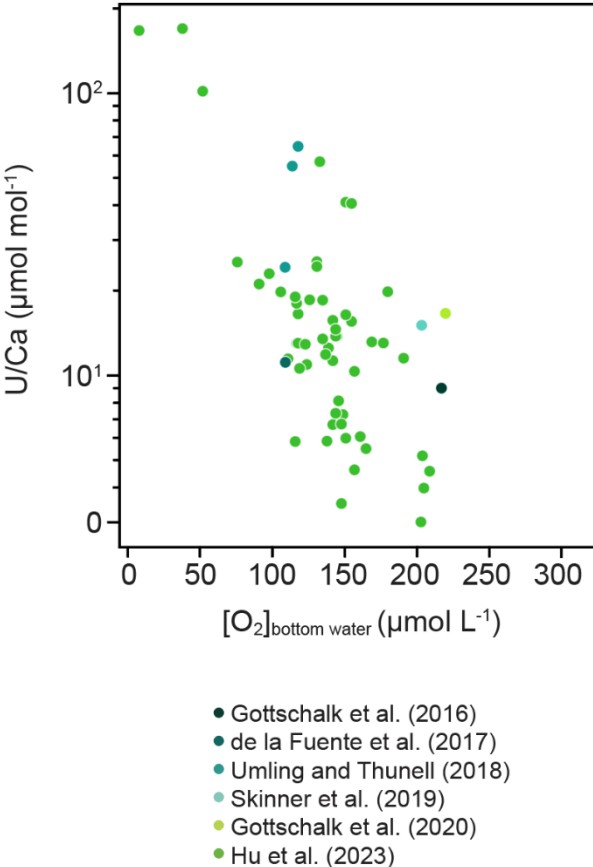

**Figure 11: U/Ca from planktic foraminifera deposited in sediments <10 ka BP compared to current bottom water oxygenation from World Ocean Atlas 18. Modified from Hu et al. (2023).**

### 6.5.3 Description of analyses and resources required

Foraminiferal elemental analyses require careful species selection, preparation and cleaning, and analytical procedures, which can vary element-to-element. For example, as outlined in Table 1, the I/Ca and Mn/Ca paleo-redox proxies specifically target lattice-bound I and Mn, while the U/Ca paleo-redox proxy targets aU, meaning differential cleaning procedures are required. In addition, while recent advances have been made (Zhou et al., 2022; Cook et al., 2022), the matrices used for analyses can vary between elements, thus requiring careful consideration and in some cases inhibiting multi-elemental analyses. Below, we provide an overview of cleaning and analytical procedures for I/Ca, Mn/Ca, and U/Ca.



The foraminiferal cleaning procedures for both I/Ca and (calcite-bound) Mn/Ca are adapted from Mg/Ca protocols (Boyle and Keigwin, 1985; Barker et al., 2003). Samples are first gently crushed with cleaned glass slides to open all chambers, then

cleaned by ultrasonication in deionized water to remove clays, followed by a 10-20 min boiling-water bath in NaOH-buffered 1% $H_2O_2$ solutions to remove organic matter, and lastly rinsed thoroughly with deionized water. It is recommended that samples with high organic matter should use additional oxidative cleaning steps (Glock et al., 2016; Winkelbauer et al., 2021). However, Mn and I analyses differ in that a reductive cleaning step is required for targeting lattice-bound Mn, as it is needed to remove authigenic Mn-oxide coatings on the exterior of the test. For I/Ca, the reductive cleaning step is not required as the

iodine content in Mn/Fe oxides is minimal (Zhou et al., 2014). Further, the reductive cleaning step has been demonstrated to cause a systematic offset in I/Ca values, and perhaps even Mn/Ca (Fritz-Endres and Fehrenbacher, 2021), so is not recommended (Zhou et al., 2022). Notably, Mn-carbonate coatings are formed under reducing conditions and cannot be removed with the existing cleaning techniques. Accordingly, increased Mn/Ca values in foraminiferal calcite may either be part of the test calcite itself or present as a coating. Diagenetic contamination on the outer or inner surface of foraminiferal

tests like Mn carbonates and Mn-rich oxyhydroxides can be identified with LA-ICP-MS, EMP and XRF mapping. However, Mn overgrowth inside foraminiferal pores is difficult to eliminate using LA-ICP-MS (Ni et al., 2020) due to the material inside the pores being ablated through the whole depth profile. We also need to consider that the Mn oxides can be removed through a reductive cleaning step whereas authigenic Mn carbonates cannot be eliminated using standard solution-based cleaning methods including the reductive cleaning step (Boyle and Keigwin, 1985).


In order to preserve authigenic foraminiferal coatings, only a 'weak chemical cleaning' protocol is used for U/Ca foraminiferal analysis, whereby only the first step of clay cleaning from the standard cleaning protocol from Barker et al. (2003) is carried out. Cleaning experiments have shown that the 'Mg-protocol' cleaning method (i.e., the addition of oxidative cleaning) removes the authigenic coating and therefore produces systematically lower U/Ca offsets, in comparison to clay cleaning only

(Boiteau et al., 2012; Chen et al., 2017).

While I/Ca, Mn/Ca and U/Ca are all commonly measured via solution-based ICP-MS or ICP-OES, for the most accurate I/Ca results, it is best to analyse samples in a mildly basic solution (Cook et al., 2022). For example, while carbonate dissolution via 1-3% nitric acid ($HNO_3$) is suitable for other trace elements/Ca ratios, samples for I/Ca measurements are typically diluted

in a 0.1% or 0.5% tertiary amine, tetramethylammonium hydroxide (TMAH), or ammonium hydroxide matrix to stabilize iodine in solution before analyses (Winkelbauer et al., 2021). Other trace elements are commonly diluted in a 2% $HNO_3$ matrix for analysis, which may be advantageous if the goal is to compare elemental data from the same foraminifera or to reduce the required sample mass and time/resources needed to separately pick, clean, and measure two sets of foraminifera in order to get separate I/Ca data. Importantly, a recent study calibrated a TMAH-nitric based matrix that allows for simultaneous

measurement of I/Ca, Mn/Ca, U/Ca, and a suite of other trace elements (e.g., Li, B, Na, Mg, Al, Mn, Fe, Zn, Sr, Cd, Ba, U, and Ca) (Cook et al., 2022). The drawback of simultaneously measuring a large suite of trace elements is that the ideal cleaning





and analysis procedures may vary element-to-element and thus, some elements may be measured under less-than-ideal circumstances. Differences in cleaning or analytical procedure may also make comparison between datasets measured using different protocols difficult.


The most commonly used reference material for I/Ca analyses is coral JCp-1 with reported values ranging between $3.70 \pm 0.27$ µmol mol$^{-1}$, but with values as high as $4.33 \pm 0.36$ µmol mol$^{-1}$ (compiled in Lu et al., 2020), suggesting the potential for some heterogeneity and/or the need for a multi-lab intercalibration. A limestone standard (ECRM752-1) that is commonly used for Mg/Ca for inter-laboratory comparison may have a similar potential for Mn/Ca, although extensive datasets are still missing

(Greaves et al., 2008).

Beyond solution-based analysis, nanoSIMS and SIMS have been used to understand I/Ca distribution in foraminiferal tests (Glock et al., 2019; Glock et al., 2016), but there has been limited subsequent application. For Mn, in situ measurements include laser ablation (LA-)ICP-MS and SIMS for quantitative measurements, while µXRF and conventional EPMA are usually used as semi-quantitative or relative measurements. For such high spatial resolution measurements, cleaning processes

are minimalistic. Ca$^{43}$ is used as the internal standard and NIST SRM 610 glass standard as the external calibration material (using established values from Jochum et al., 2011) for LA-ICP-MS. NIST SRM 612 glass (Jochum et al., 2011) and calcium carbonate pellets of MACS-3 (Jochum et al., 2012), JCp1, JCt1 and now NFHS (Boer et al., 2022) can be used as quality control material. For U, Skinner et al. (2019) showed that core-top and downcore measurements of U/Ca using LA-ICP-MS are also possible. We note that high resolution ICP-MS is not required, and a more accessible quadrupole ICP-MS may also

be used. However, due to the low concentration of uranium in foraminiferal coatings (compared to, for example, Mg or Sr), ICP-OES is not sensitive enough to carry out U/Ca measurements.

Table 1. Summary of cleaning methods and analytical techniques.

| Trace Element | Cleaning Method | Instrument for measurement | Lattice Bound/ Coating | References |
|---|---|---|---|---|
| **I/Ca** | Clay and organic matter | Solution ICP-MS | Lattice bound | Lu et al, 2010; Glock et al., 2014; Winkelbauer et al., 2021; Cook et al., 2022; Zhou, Hess et al., 2023 |
| **Mn/Ca** | Clay and organic matter | Solution ICP-MS/ICP-OES LA-ICP-MS | Lattice bound | Boyle and Keigwin, 1985; Groeneveld & Filipsson 2013; Ni et al., 2020; Brinkmann et al 2023 |
| **U/Ca (Mn/Ca) (U/Mn)** | Clay clean only | Solution ICP-MS/ LA-ICP-MS (coatings) | Coating | Boiteau et al., 2012, Gottschalk et al., 2016; 2020, Skinner et al., 2019; Chen et al., 2017; Umling and Thunell, 2018 |





### 6.5.4 Future directions

There are important knowledge gaps for each of the proxies discussed above. Addressing these gaps requires an overview of water column and pore water geochemistry obtained through oceanographic and sediment and pore water sampling. Though multi-element applications show potential (Hu et al., 2023), cross proxy calibration comparisons of I, Mn, and U in sediments or water column transects are currently lacking. We also note that there are gaps in the calibration of the various element/Ca ratios relative to oxygen. For planktic and benthic foraminifera, I/Ca ratio gaps are found around oxygen levels between 80 and 140 µmol kg$^{-1}$ respectively (Fig. 9). Mn/Ca measurements from oceanic settings are concentrated at oxygen levels below 70 µmol kg$^{-1}$ (Fig. 10). The U/Ca proxy is specific to coatings, and thus is bottom water specific, but there are only a few calibrations in low-oxygen environments (Fig. 11). Finally, the application of Mn/Ca to planktic foraminifera has hardly been explored, but some recent studies show promise that the Mn/Ca could record redox conditions in OMZs (Vedamati et al., 2015, Davis et al., 2023).

It is important to understand the impacts that vital effects may exert on element/Ca ratios. Controlled culture experiments allow the assessment of direct relationships between foraminifera element/Ca (e.g., I/Ca and Mn/Ca) and ambient sea- and pore water elemental concentrations. Planktic foraminifera migrate through the water column during their life cycle (Schiebel and Hemleben, 2017). As such, the element/Ca signal represents an integrated signal from environments with potentially variable elemental compositions. In order to constrain pathways of element incorporation into planktic foraminifera calcite, we need to study the life cycle of living planktic foraminifera (i.e., from plankton tows), as well as settling dead planktic foraminifera (i.e., from sediment traps), and planktic foraminifera in sediments (e.g., from core-top samples), all across the variable ambient element concentrations. This integrated framework allows us to track the proxy from seawater elemental values to the fossil record, thus integrating vital effects and diagenesis. Benthic foraminifera can actively migrate between the top of sediments and different redox zones within the sediments. To monitor the effect of their migratory behaviour on element incorporation, controlled culture experiments are needed, tracking benthic foraminifera depth habitat, calcification and bottom- and pore-water chemistry.

On a final note, analysis of successive chambers from single foraminiferal shells has shown promise for Mn/Ca when using high resolution techniques such as LA-ICP-MS (Guo et al., 2019; Petersen et al., 2018; Brinkmann et al., 2023). This approach may be further developed for I/Ca, U/Ca and other redox proxies to reconstruct short-term (seasonal to annual, since foraminifera can live up to 1-2 years) variations in paleo-records.



## 6.6 Foraminifera assemblages

### 6.6.1 Introduction


Like all organisms, foraminifera thrive when environmental conditions match their requirements. Among the environmental parameters that drive benthic foraminifera species presence and abundance, the most important are export productivity (supply of nutrients to the seafloor) and bottom water oxygenation (Jorissen et al., 1995). The tolerance of specific species to different

oxygen levels has made benthic foraminifera assemblages an especially useful tool for qualitative and quantitative reconstructions of past bottom water oxygenation variability (Sen Gupta and Machain-Castillo, 1993; Kaiho, 1994; Alve and Bernhard, 1995; Bernhard et al., 1997;  Baas et al., 1998; Nordberg et al., 2000; Jannink, 2001; Schmiedl et al., 2003; Leiter and Altenbach, 2010; Ohkushi et al., 2013; Tetard et al., 2017; Sharon et al., 2021; Erdem et al., 2020; Tetard et al., 2021).

### 6.6.2 Historical perspective of benthic foraminifera

The use of benthic foraminifera assemblages as indicators of environmental conditions began with their application as proxies for paleobathymetry (Bandy, 1953). Since then, numerous studies have established connections between the distribution of benthic foraminifera and various environmental parameters, with particular emphasis on oxygenation in open ocean settings (e.g., Phleger and Soutar, 1973; Mackensen and Douglas, 1989; Sen Gupta and Machain-Castillo, 1993; Bernhard et al., 1997; den Dulk et al., 1998; Jannink et al., 1998; Levin et al., 2002; Schumacher et al., 2007; Cardich et al., 2012, 2015; Mallon et

al., 2012; Caulle et al., 2014). Certain species inhabit regions characterized by sustained low-oxygen concentrations, such as the OMZs in the Pacific Ocean (Smith, 1964; Bernhard et al., 1997; Bernhard and Sep Gupta, 1999; Cardich et al., 2012; 2015; Mallon et al., 2012; Erdem et al., 2020; Castillo et al., 2021; Tavera Martínez et al., 2022), and Arabian Sea (Hermelin and Shimmield, 1990; Caulle et al., 2015; Gooday et al., 2000; Jannink et al., 1998), as well as in restricted basins and fjords (Bernhard and Alve, 1996; Leiter and Altenbach, 2010; Nordberg et al., 2000; Bouchet et al., 2012; Fontanier et al., 2014).

Culture studies have confirmed the ability of certain species of both benthic and planktic foraminifera to survive (Alve and Bernhard, 1995; Moodley and Hess, 1992; Moodley et al., 1997; Geslin et al., 2004, 2014; Bernhard and Alve, 1996), calcify (Kuroyanagi et al., 2013; Nardelli et al., 2014), and thrive (Bernhard, 1993, Enge et al., 2016, Piña-Ochoa et al., 2008; Orsi et al., 2020) under very low-oxygen and even euxinic conditions (Fig. 12).

Oxygen-depleted marine environments are often characterized by a high organic carbon rain rate, and these factors collectively

influence the habitat depth, abundance, and assemblage composition of benthic foraminifera (e.g., Lutze and Coulbourn, 1984; Corliss and Emerson, 1990; Loubere, 1994; Altenbach et al., 1999; Gooday and Rathburn, 1999; Geslin et al., 2004). Living benthic foraminifera are found both on the sediment surface (epifaunal), and throughout at least the top 10 cm of the sediment (infaunal), although the proportion of the total population decreases rapidly with increasing depth (Corliss, 1985; 1991). Moreover, some benthic foraminifera can migrate vertically within sediments, with their habitat depths and position in the

sediment column influenced by the organic matter flux and availability of resources such as oxygen (Bernhard, 1992; Barmawidjaja et al., 1992; Loubere et al., 1993; Linke & Lutze, 1993; Jorissen et al., 1995; Geslin et al., 2004). This notion





plays an important role in the Trophic-Oxygen, or TROX, model which was the first to consider food and oxygen availability as key factors in determining benthic foraminifera assemblages and microhabitat (Barmawidjaja et al., 1992; Jorissen et al., 1995). The oxygen concentration in pore water is influenced not only by bottom water oxygen concentration but also by respiration in pore waters, which is proportional to the organic matter content in the sediments. Thus, bottom water oxygen concentration is only one of the drivers for infaunal foraminifera living deeper in the sediments (Jorissen et al., 1995), particularly because several infaunal species have been shown to denitrify (i.e., respire nitrate instead of oxygen; Risgaard-Petersen et al., 2006). A more detailed discussion of this phenomenon will follow.

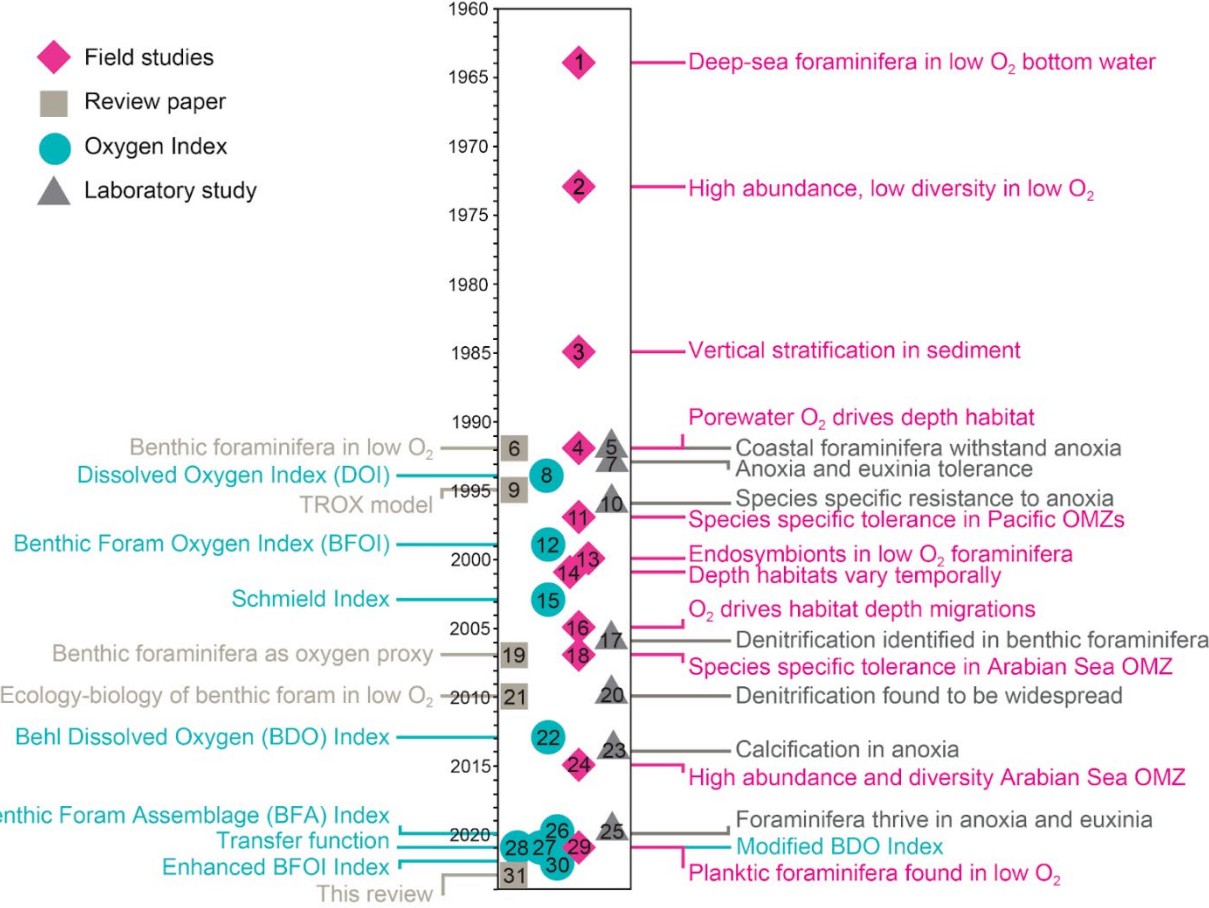

**Figure 12: Timeline of key discoveries and advances regarding physiology, ecology and proxy development based on benthic and planktic foraminifera since the 1960's including field and laboratory studies, oxygen indices and review papers. 1. Smith (1964), 2. Phleger and Soutar (1973), 3. Corliss (1985), 4. Bernhard (1992), 5. Moodley and Hess (1992), 6. Sen Gupta and Machain-Castillo (1993), 7. Bernhard (1993), 8. Kaiho (1994), 9. Jorissen et al. (1995), 10. Bernhard and Alve (1996), 11. Bernhard et al. (1997), 12. Kaiho (1999), 13. Bernhard et al. (2000), 14. Alve and Murray (2001), 15. Schmiedl et al. (2003), 16. Fontanier et al. (2005), 17. Risgaard-Petersen et al. (2006), 18. Schumacher et al. (2007), 19. Jorissen et al. (2007), 20. Piña-Ochoa et al. (2010a), 21. Koho and Piña-Ochoa (2010), 22. Ohkushi et al. (2013), 23. Nardelli et al. (2014), 24. Caulle et al. (2015), 25. Orsi et al. (2020), 26. Erdem et al. (2020), 27. Sharon et al. (2021), 28. Tetard et al. (2021a,b), 29. Davis et al. (2021), 30. Kranner et al. (2022), 31. This review.**



Ecological studies have identified benthic foraminifera species that can serve as indicator taxa for low-oxygen conditions, and the association of specific species with quantitative ranges of bottom water oxygen. Several studies have developed indices to reconstruct paleo-oxygen levels based on benthic foraminifera assemblages (e.g., Kaiho, 1994, 1999; Jannink, 2001; Schmiedl et al., 2003, review in Jorissen et al., 2007; Ohkushi et al., 2013; Tetard et al., 2017; Sharon et al., 2021; Erdem et al., 2020; Tetard et al., 2021, Kranner et al., 2022), which are summarized here (Fig. 12). The benthic foraminifera oxygen index (BFOI)

developed by Kaiho (1994, 1999) considers the relative proportion of taxa indicative of low bottom or pore water oxygen conditions compared to the total fauna. Another index developed by Jannink (2001) focuses on the presence of oxyphilic species, which consistently inhabit the top centimetre of sediment. Schmiedl et al. (2003) proposed a method based on a combination of the relative proportion of low-oxygen tolerant species and a diversity index. Ohkushi et al. (2013) developed a new index using thresholds of oxygen tolerance for different species and paleoenvironment assessments through principal

component analysis (PCA) of assemblages. Tetard et al. (2017, 2021a) also used PCA and calibrated transfer functions to semi-quantitatively reconstruct benthic oxygen offshore California during the last 80 kyrs. Sharon et al. (2021) presented downcore applications using a modification of the Ohkushi et al. (2013) index, updated through comparison with modern assemblages from core-top studies and cross-checked with redox-sensitive trace metals. Erdem et al. (2020) employed a transfer function approach, using living Rose Bengal stained benthic assemblages and prevailing oxygen concentrations as a

regional analogue for downcore environments. More recently, Tetard et al. (2021) proposed the use of the relative abundance of *Eubuliminella exilis (Buliminella tenuata)* as a proxy, calibrated with average bottom-water oxygenation and core-top samples predominantly from the Pacific Ocean. The most recent paleo oxygen quantification up until now updates the original BFOI from Kaiho (Kranner et al., 2022). The enhanced BFOI (EBFOI) considers multiple indicator species and transfer functions for both bottom water and pore water oxygen concentrations. Continued improvement and implementation of these

approaches are active areas of research.

All of these indices are considered valid, and the choice of index depends on factors such as the location, available material, and the specific scientific question being addressed. Identifying in which environments whole assemblages or indicator taxa are most useful will also continue to improve reconstructions. A comparison of the different indices applied to a sediment core from the Californian Margin is shown in Fig. 13.






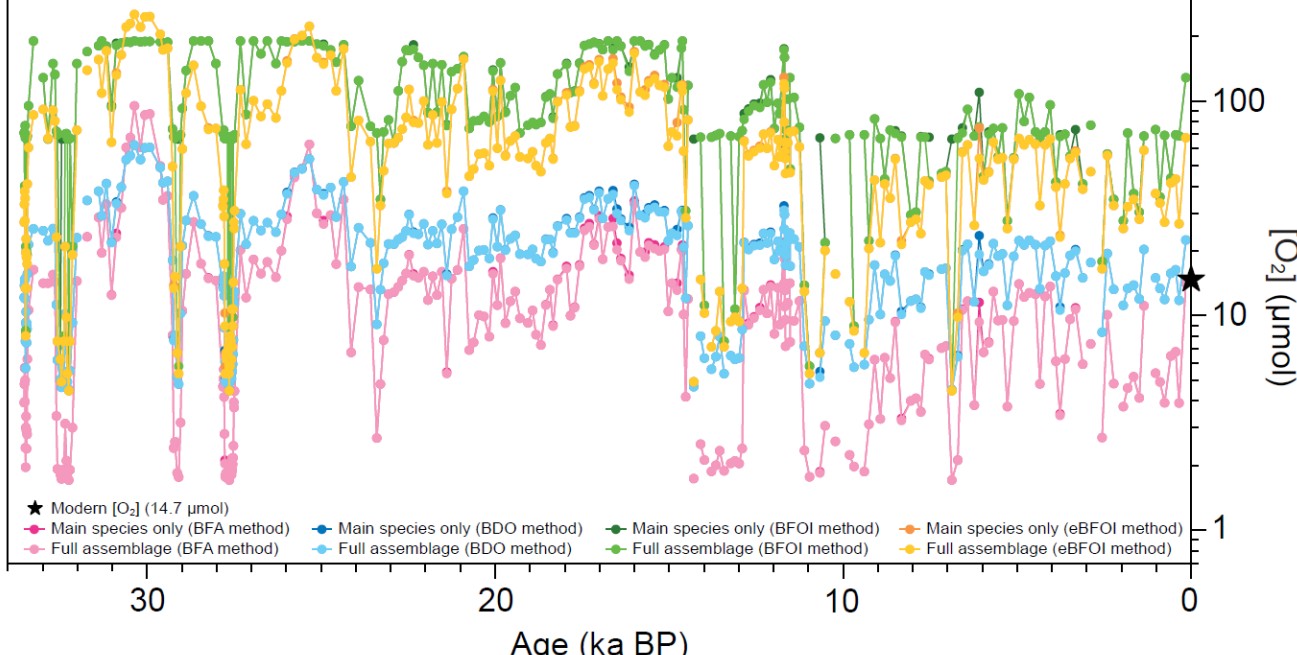

**Figure 13: Comparison of past oxygen reconstructions for core site MD02-2503 (Okhushi et al., 2013) comparing main species (determined by a PCA on the census data, usually corresponding to species with an averaged relative abundance >5%) with the complete assemblage. This comparison was performed using 4 dissolved oxygen estimation methods: the BFOI (Kaiho, 1994), the BDO (Okhushi et al., 2013), the BFA (Tetard et al., 2021), and the eBFOI (Kranner et al., 2022). Black star shows the modern oxygen value at the core site.**

There is active debate whether it is advantageous to use full assemblages (e.g., Sharon et al., 2021), only the dominant species (e.g., Tetard et al., 2017), or only indicator species (see also Fig. 13). Using the whole assemblages might be more time consuming and it is likely that different approaches are appropriate for different environments and questions. For example, in environments with a high degree of temporal or spatial averaging, use of indicator species may be more successful in positively identifying low-oxygen events, but may be limited in the range of oxygen conditions reconstructed. However, these hypotheses deserve more rigorous testing.

### 6.6.3 Historical perspective of benthic foraminifera

Planktic foraminifera assemblages differ from benthic in that they represent a collection of foraminifera inhabiting the water column above the seafloor. As most species occupy a near-surface habitat, planktic foraminifera assemblages have primarily been used to reconstruct sea surface or mixed layer temperature and, to a lesser extent, conditions such as productivity, salinity, and stratification (e.g., Imbrie & Kipp et al., 1971; Kipp, 1976; Cayre et al., 1999; Kucera et al., 2005; Morey et al., 2005; Kucera, 2007; Sicca et al., 2009). However, certain extant species of planktic foraminifera, including *Hastigerinella digitata* and *Globorotaloides hexagonus* have been observed in low-oxygen pelagic environments (Hull et al., 2011, Davis et al., 2021), with the latter taxon subsequently used to infer past distribution of low-oxygen water in the open ocean (Davis et al., 2023).



Some extinct species of planktic foraminifera, particularly those with digitate morphologies such as the clavigerinellids, have also been associated with low-oxygen environments (reviewed in Coxall et al., 2007). As planktic foraminifera assemblages integrate across multiple distinct depth habitats, the use of indicator species rather than relative abundance is generally more appropriate for interpreting paleo-oxygenation. One notable example is the biserial chiloguembelinids, interpreted as OMZs-dwellers based on isotope records (Boersma et al., 1987, Boersma and Premoli Silva, 1989, Luciani et al., 2020). The presence of their shells has been used to infer the presence of a pelagic OMZ in Paleocene to Oligocene sediments (e.g., Corfield & Shackleton, 1988; Pardo et al., 1997; Luciani et al., 2020).

### 6.6.4 Analyses and Required Resources

The majority of foraminifera assemblages relevant to Cenozoic oxygen reconstructions are obtained from sedimentary records, with a smaller portion coming from lithified outcrops. For the purposes of this review, we will focus on the sedimentary records. Collecting samples from subtidal stations, ranging from coastal to abyssal environments, often requires the use of seagoing vessels. However, the study of foraminifera assemblages is relatively inexpensive once sediment samples have been collected. Prior to analysing assemblages, sediment samples typically undergo various laboratory treatments. These may include disaggregation using hexametaphosphate, wet or dry sieving to separate different size fractions, and sample splitting. The subsequent steps involve 'picking', isolating and sorting foraminifera tests using a fine brush and black tray, and gathering the specimens in micropaleontological slides. Standard collection and counting procedures have been reviewed by Schönfeld (2012).

It is important to highlight that conducting robust counts requires an even split of the sample. Typically, a split containing 200–300 specimens is considered sufficient for evaluating species assemblages with a reasonable degree of accuracy (Patterson & Fishbein, 1989). However, this number could be lower if only the most abundant species are being considered or insufficient if rare species are important to the analyses (Fatela & Taborda, 2002). This work is carried out using an optical stereomicroscope equipped with a light source. Best practices also require documenting taxonomic attribution using either a high-resolution camera or a Scanning Electron Microscope (SEM). Ultimately, the most crucial resource needed for these analyses is a trained researcher or technician, skilled in taxonomic identification of foraminifera species. It is worth noting that acquiring such ability is relatively accessible compared to the taxonomy of macrofauna.

### 6.6.5 Recent Advancements

Several recent advances in analytical methods have shown promise in automating labour-intensive aspects of foraminifera assemblage analysis. This is exemplified by the utilization of high through-put imaging and artificial intelligence (AI) neural networks to generate species-level identifications (Mitra et al., 2019; Hsiang et al., 2019; Marchant et al., 2020). These methods have been combined with mechanical shell sorting to further enhance efficiency (Mitra et al., 2019; Richmond et al., 2022). While the current applications of automatization techniques have been primarily focused on planktic foraminifera, Marchant





et al. (2020) have also included a few benthic taxa. Although these approaches are still in their early stages, the use of AI to generate assemblages holds tremendous potential, especially as models can be trained on an increasing number of species.

Another important advancement is the increased availability of imaging technology. This includes the use of high-resolution digital cameras, enabling high-quality, true-colour imaging (e.g., Erdem and Schonfeld, 2017; Wilfert et al., 2015), as well as the growing popularity of desktop or environmental SEMs. Both approaches enhance the speed and cost-effectiveness of generating and sharing of assemblage data. Alongside improved data storage capabilities, both internally and in online repositories, the sharing of images can and should continue to increase the transparency and reproducibility of assemblage

work. The best practice for assemblage work is that all, or at least a representative subset, of images should be published alongside manuscripts.

### 6.6.6 Proxy driver(s)

#### 6.6.6.1 Benthic foraminifera

Most benthic foraminifera have the ability to survive at lower oxygen concentrations of around 1-2 ml $L^{-1}$ or 44.6 - 89.3 µmol

$kg^{-1}$ compared to other benthic microorganisms such as nematodes or ciliates (Jorissen et al., 1995; Bernhard et al., 1997; Van der Zwaan et al., 1999; Levin et al., 2001; Murray, 2001; Geslin et al., 2011). Thus, it is only below this threshold that oxygen is expected to be a key driver of assemblage composition and robustly quantifiable. While oxygen concentrations and the tolerance of individual species are important factors, designating species as low-oxygen indicators is an ecological oversimplification, but a useful one. Other factors such as food supply, biogeography, differing metabolic strategies, habitat

depth, and other environmental gradients including temperature and salinity, all contribute to defining ecological ranges. Some species may be opportunistic rather than specifically adapted to low-oxygen conditions. Other species may possess a unique tolerance to very low-oxygen or have metabolic adaptations that allow them to utilize alternate electron receptors. Although low-oxygen adapted species may survive in oxic environments, their relative abundance increases when oxygen becomes a limiting factor and, thus, competition with other species decreases.

There is a particularly complex interplay between organic matter flux and oxygenation of bottom and pore waters. In general, higher organic matter content leads to higher remineralization rates, i.e., through decomposition of organic carbon, and subsequent oxygen consumption, resulting in lower dissolved oxygen concentrations. This covariation can make it challenging or impossible to differentiate between the two drivers. Indeed, many indicator species can be viewed as specialists adapted to high productivity that also happen to tolerate low-oxygen conditions. This is likely the case for taxa such as *Bulimina, Bolivina,*

and *Nonionella* (Margreth et al., 2009), which are rare in oligotrophic low-oxygen conditions, but found in highly productive, yet oxic, regions.

Discrepancies can also be observed between geographical areas with similar bottom water oxygenation, particularly in defining specific indicator taxa. For example, *Uvigerina peregrina* is typical of OMZ environments in the Pacific and Arabian Sea, where it can be used in oxygen reconstructions (e.g., Moffitt et al., 2014; Schumacher et al., 2007). However, the same species



is considered a high productivity indicator in the relatively well oxygenated waters of the Northeast Atlantic (e.g., Fontanier et al., 2005; Mojtahid et al., 2017). The same observation applies for *Nonionella stella* (Moffitt et al., 2014; Diz and Francés, 2008). In some cases, these geographic differences may highlight the complexity of food and oxygen as co-drivers, or they may suggest the presence of unknown environmental drivers, cryptic species, or adaptive differences between populations.

Benthic foraminifera are distributed throughout the upper centimetres of the sediment column, and their depth habitat is influenced by the availability of food, oxygen, ecosystem stability, bioturbation, and competition for resources (Jorissen et al., 2007 and references therein). In oligotrophic conditions, particularly open ocean settings, most species tend to occupy the uppermost sediment levels to maximize access to food exported from the photic zone (Linke and Lutze 1993). However, other factors such as the quality of food, with fresh phytodetritus sometimes preferred, can also have an impact (Smart et al., 1994; Gooday & Hughes, 2002). In eutrophic conditions, where oxygen levels are low, the depth habitat of benthic foraminifera is determined by the capability of the species or assemblage to tolerate low-oxygen (Jorissen et al., 1995). A key consequence of variability in habitat depth is that not all benthic foraminifera at a particular site will experience the same environmental conditions (Tetard et al., 2021a). This variability in sediment depth habitat not only impacts assemblage composition, but also plays a fundamental role in other oxygen proxies such as $\Delta\delta^{13}C$ (Section 6.8) and determines the chemical microenvironment that is recorded by trace metal proxies (Section 6.5).

While oxygen concentration may be limiting for many species, certain adaptations, including anaerobic metabolisms like denitrification, enable some foraminifera to persist even in euxinic conditions (Orsi et al., 2020). Deep-infaunal *Globobulimina* spp. (i.e., *G. turgida, G. pacifica, G. affinis*), and *Chilostomella* spp. (i.e., *C. ovoidea, C. oolina*) inhabit anoxic sediments below the oxygen and nitrate penetration depth (e.g., Jorissen et al. 1998; Schönfeld 2001; Fontanier et al. 2002, 2003, 2005; Koho et al. 2007, 2008a; Glud et al. 2009). These species are capable of storing nitrate and respiring nitrate in absence of oxygen (Risgaard-Petersen et al., 2006; Piña-Ochoa et al., 2010a, b; Koho and Pina-Ochoa, 2012). They can be considered facultative (an)aerobes. The foraminiferal denitrification pathway is a rare example of eukaryotic denitrification (Woehle et al., 2018; Gomaa et al., 2021), and while foraminifera appear to perform only incomplete denitrification innately, the missing steps are likely complemented by bacterial symbionts (Bernhard et al., 2010; Woehle et al., 2022). Similar abilities, such as intracellular nitrate accumulation have been found in species including *Nonionella* cf. *stella*, *Uvigerina akitaensis, and Bolivina spissa* (Høgslund et al., 2008; Glud et al. 2009). Some species even exhibit a metabolic preference for nitrate over oxygen with nitrate concentrations rather than oxygen concentrations being limiting (Glock et al., 2019; Suokhrie et al., 2020). Dormancy has also been identified as an adaptive response to short-term oxygen depletion (LeKieffre et al., 2017; Ross and Hallock, 2017). A comprehensive review about the survival strategies of foraminifera under low-oxygen conditions is given in Glock (2023). However, due to the limited number of species that have evolved these adaptations, oxygen-depleted conditions are characterized by lower species diversity and higher dominance compared to oxygenated settings (Phleger and Soutar, 1973; Sen Gupta and Machain-Castillo, 1993; Gooday et al., 2000, Levin, 2003; Koho and Ochoa, 2011).





### 6.6.6.2 Planktic foraminifera

The role of oxygen as a driver of planktic foraminifera abundances is not as well-established as in benthic foraminifera, however it is likely that the key environmental drivers in low-oxygen environments are similar. A combination of drivers including accessibility of food, and either tolerance of or adaptations to low-oxygen are probable. Some planktic low-oxygen specialists, such as *H. digitata* and *G. hexagonus*, do not seem to occur in oxic environments, and may possess special adaptations to low-oxygen. It remains unknown whether some more widely distributed deep-dwelling species such as *Globorotalia scitula* or *Globorotalia truncatulinoides*, whose habitat may intersect low-oxygen waters, can opportunistically persist across both oxic and low-oxygen environments.

### 6.6.7 Marine archives and limitations

Reconstructions based on foraminifera assemblages can be applied in any marine environment where foraminifera shells are deposited and preserved in relatively undisturbed sediments. This includes fjord, estuarine, supratidal shelf, and deep-sea (including continental shelf) environments above the lysocline. While foraminifera archives are widely available, they are most useful as oxygen indicators in low-oxygen environments, where oxygen limitation is a meaningful driver of assemblage composition. The use of foraminiferal assemblages may be more limited in settings where it is more difficult to deconvolve oxygen and productivity as drivers.

Low carbonate saturation state environments also pose a limitation, where living calcareous species are rare or absent due to unfavourable conditions (Bernhard et al., 2009; Dias et al., 2010; Petit et al., 2013; Martinez et al., 2018). Taphonomic processes in such environments can further complicate interpretations. Corrosive bottom or pore waters can lead to preferential dissolution of small, thin-walled species, or those with a spiral arrangement of chambers (Berger, 1973; Hecht et al., 1975; Nguyen et al., 2009, 2011, 2014), or even lead to the complete loss of calcareous foraminifera (Gutiérrez et al., 2009), potentially altering the preserved assemblage. Thus, while dissolution could impact different methods differently (i.e., use of only indicator species vs diversity metrics), it should always be considered. Dissolution can pose limitations, particularly in very low-oxygen or high productivity sites where high respiration rates can result in both low carbonate saturation states and oxygen-depleted conditions.

Fossil assemblages contain individuals that lived over a period of time, depending on factors such as sedimentation rate and degree of sediment mixing. Within a single centimetre of sediment, it is possible to find foraminiferal species that lived contemporaneously or thousands of years apart. This complication is common to all marine sedimentary records, but it poses specific concerns for foraminifera. One important factor to consider is the vertical migration of infaunal species within the sediment column, which can occur over several centimetres (Alve and Murray, 2001; Jorissen et al., 1995; Duijnstee et al., 2003). As a result, older fossil epibenthic foraminifera can be recovered alongside younger infaunal specimens. This vertical movement of species contributes to the phenomenon of time-averaging in fossil assemblages. In environments with seasonal or short-term variation in oxygen levels, the presence of both oxic and low-oxygen taxa as part of the same death assemblage





can limit high temporal resolution reconstructions using "complete" foraminifera assemblages. However, this time-averaging effect can also offer opportunities to investigate seasonal contrasts. For instance, when species from contrasting environmental settings occur together in the same samples, it could indicate large seasonal (or other short-term) differences in oxygenation at the site.

Another challenge in interpreting foraminiferal assemblages is the occurrence of "no-analogue" fauna, where taxa that are not
observed together in modern environments coexist in the sediment record. This poses a potential limitation as most identification and quantification of relationships between assemblages or indicator species and oxygenation are based on observations in the modern ocean. The presence of no-analogue fauna can have different explanations. Time-averaging can contribute to the mixing of taxa from different time periods, creating assemblages that do not reflect the present-day co-occurrence patterns. In other cases, the limited range of modern sites used for calibration may not capture the full spectrum of
environmental conditions that have existed throughout space and time. Consequently, there may be gaps in our understanding of the relationships between foraminiferal assemblages and oxygenation in certain environments. This highlights the need for further research and a broader spatial and temporal sampling of modern environments to improve our understanding of foraminiferal assemblage dynamics and their relationship with oxygenation.

### 6.6.8 Future Directions and General Recommendations

Advances in automation and imaging technology have made it easier to capture and analyse high-resolution images of foraminiferal assemblages. As a result, the  routine inclusion of images should become a standard practice in the field. Furthermore, efforts should be made to ensure that images are uploaded to publicly available archives or repositories and stored as supplementary media whenever permitted by journals. The inclusion of images in publications and their archival in digital repositories offer several benefits to the scientific community. Firstly, it increases reproducibility, as other researchers
can refer to the images for verification, validation, and comparison. Secondly, it facilitates the resolution of taxonomic and nomenclature issues related to key indicator species. This is particularly important considering the potential taxonomic revision and updates in the future.

### 6.6.8.1 Open Questions: Ecology and Metabolism

To improve reconstructions of past oxygen levels based on foraminiferal assemblages, it is crucial to enhance our
understanding of foraminifera ecology. By doing so, we can distinguish between opportunistic species driven by food availability and true low-oxygen specialists. Exploring ecological questions related to seasonality and physiological oxygen tolerances in key species is also important. Understanding the constraints and variations in seasonality can provide insights in the temporal dynamics of foraminiferal assemblages and their response to changing oxygen conditions. Investigating the physiological oxygen tolerances of different species, while considering interactions between oxygen, food availability, and
carbonate chemistry stresses, will contribute to a more comprehensive understanding of their ecological responses.





The exploration of metabolic adaptations and the role of oxygen and nitrogen respiration in different species and environments is another avenue of research. New technologies, such as the increasing availability of genetics and genomics' analyses hold promise for addressing these metabolic questions (Woehle et al. 2018; Orsi et al., 2020; Gomaa et al., 2021), and explaining the variations in oxygen ranges observed among geographically disparate populations. These advanced techniques can provide valuable insights into the underlying mechanisms and adaptations that allow foraminifera to thrive in low-oxygen environments. It is important to note that while this discussion has focused primarily on benthic foraminifera, addressing the same suite of questions for planktic foraminifera is equally important and will improve the use of planktic assemblages for reconstructing pelagic oxygenation.

Continued focus on foraminifera ecology can facilitate the integration of "traditional" assemblage-based proxies with emergent proxies, leading to more comprehensive reconstructions. Understanding the ecological and physiologic drivers of foraminiferal oxygen tolerance can help researchers to determine where regionally-specific or global calibrations are appropriate and which species should be considered. This will be crucial when applying calibrations developed in specific times and regions, such as the modern Pacific, to other times and places. It is especially relevant when comparing restricted basins and fjord environments to open ocean OMZs. Understanding the species-specific responses to varying oxygen content and co-stressors can be achieved through advances in culturing techniques, enabling analyses of assemblage-level changes (Bernhard et al., 2021). Furthermore, denitrifying species capable of living across large oxygen gradients can serve as target species for complementary proxy approaches, such as foraminifera morphometrics and geochemistry. These approaches can provide additional insights into environmental conditions and further enhance our ability to reconstruct past oxygen levels.

### 6.6.8.2 Open Questions: Methods for expanding oxygen ranges

While foraminifera assemblages form the basis of some of the more established approaches for constraining marine oxygenation, a lack of sensitivity to oxygen higher than 1-2 ml L$^{-1}$ (44.6-89.3 µmol kg$^{-1}$) remains a key limitation. The inclusion of other fossilizing organisms in addition to foraminifera is one solution for extending oxygen reconstructions into more oxic conditions. This potential is reviewed briefly in Gooday et al., (2009) and discussed in Mhyre et al. (2017). By incorporating groups such as brachiopods, mollusks, ostracods, brittle stars, sponges and other organisms with hard parts recoverable in marine sediments, a composite index could be developed that is sensitive to intermediate oxygen concentrations. This expanded approach would provide a more comprehensive understanding of past oxygen levels.

The use of foraminifera assemblages for reconstructing oxygenation has a robust, decades-long history. Unfortunately, due to the elevated time consumption of this approach and recent advances in genetic methods, traditional morphology-based taxonomists are getting rarer. This is a disadvantage, since careful distinction of morphologically similar species is often crucial also for geochemical approaches. However, studying foraminifera assemblages remains an area of active research. Advancements in both knowledge and technological applications continue to expand the utility of foraminifera assemblages as an oxygen proxy. Lessons from this approach have already and will continue to inform other foraminifera-based proxies and it is likely that it will remain in the toolkit of paleoceanographers for years to come.





### 6.6.9 Contribution to Morphological Proxies

The principles behind the use of foraminifera assemblages have sometimes been distilled into a few morphological traits, primarily size and shape. Most extant low-oxygen tolerant species share several features. The calcareous hyaline foraminifera, which outnumber agglutinated and porcelaneous species in low-oxygen levels, often have thin, porous test walls such as those seen in *Bolivina* and *Globobulimina* species (e.g., Kaiho, 1994; Sen Gupta & Machain-Castillo, 1993; Bernhard & Sen Gupta, 2003; Gooday, 2003; Caulle et al., 2014). Conversely, larger species like *Dentalina*, *Lagena*, and *Nodosaria* species are associated with well-oxygenated conditions. In terms of shape, more circular epifaunal species like *Cibicides* and *Planulina* species dominate during oxygenated conditions, while elongated infaunal species like *Eubuliminella*, *Bolivina*, and *Brizalina* species migrate to the water-sediment interface, and dominate the benthic foraminifera record during oxygen-impoverished periods (Jorissen et al., 1995, 2007; Palmer et al., 2020). These observations of size and shape have been crucial to the development of indices (see Section 6.7.2), based on test morphology and individual size. The recent MARIN index (Tetard et al., 2021a) (see Section 6.7.2.2 for details) also relies on size and shape measurements of complete benthic assemblages, using semi-automated methods. By measuring size and shape using image analysis software, such as ImageJ (https://imagej.nih.gov/ij/download.html and https://doi.org/10.5281/zenodo.4740079), past oxygen values for OMZ conditions can be estimated without the need for species-level identification. An assemblage characterized by small, elongated tests would indicate low bottom water oxygen conditions, while an assemblage dominated by large, spirally-arranged tests would indicate well-oxygenated conditions. The impact of changing oxygen conditions on foraminiferal morphology is discussed in more detail in the next section.

## 6.7 Foraminifera morphometrics

### 6.7.1 Introduction

The morphology of foraminifera shells, both within and across species, is widely understood to reflect environmental conditions, including oxygenation (Fig. 14 and Tab. 6.7.1). Specific features include those influenced by in situ calcification environment, such as shell porosity, size, ornamentation features (e.g., spines, costae and keels), shape, and coiling direction. It also includes features influenced by the carbonate saturation state of the depositional environment where the shells are deposited. These can manifest as shell density changes, pitting or other dissolution features. Some features like shell thickness or size-normalized weight can be affected by both growth and depositional environments. Out of these metrics, porosity (percentage of pore area of total shell surface) and pore density (number of pores per surface area) have received particular attention as proxies for redox conditions and they will be the primary focus of this section (Petersen et al., 2016; Rathburn et al., 2018; Glock et al., 2011, 2018 & 2022; Tetard et al., 2017; Richirt et al., 2019; Lu et al., 2022). Shell size and circularity, especially within the context of the Major Axis and Roundness INdex (MARIN; Tetard et al., 2021), will be discussed as well.



**Figure 14: Cartoon showing the response of different foraminiferal morphometrics to changes in oxygen concentrations. Typical conditions at a continental margin OMZ are used as an example. For an overview of methods, see Tables 2 and 3.**

### 6.7.2 Historical Perspective

#### 6.7.2.1 Porosity and Pore Density

Porosity is defined as the percentage of pore area relative to the surface area of the foraminiferal shell. Pore density is defined as the number of pores per surface area. The two measures are distinct, but tightly linked: larger pores at constant pore density result in a higher porosity. A detailed review of pores as an oxygen proxy in foraminifera can be found in Glock et al. (2012), the history of which we summarized here. The first studies on the morphology and fine structure of pores in foraminiferal tests go back to the middle of the 20th century. Advances in electron microscopy during this period allowed scientists to describe "pore plates" or "sieve plates" that cover the pores of many benthic foraminiferal species (Le Calvez, 1947; Jahn, 1953; Arnold



1954; Angell 1967; Sliter, 1974; Berthold, 1976, Leutenegger, 1977). There was also early evidence for strong environmental influences on pore density and other morphological features in *Bolivina spissa* and other closely related bolivinids (Lutze, 1962; Harmann, 1964). In the 1980s-1990s more studies described the correlation between pore size and pore density in benthic foraminifera and ambient oxygen concentrations and species with higher porosity having been suggested as indicators for oxygen-depleted conditions (Bernhard, 1986; Perez-Cruz and Machain-Castillo, 1990; Moodley and Hess, 1992; Sen-Gupta and Machain-Castillo, 1993; Kaiho, 1994). Since then, several studies have shown that in some species of benthic foraminifera, individuals living in oxygen-depleted environments have higher pore density and porosity than conspecifics in well-oxygenated conditions (Glock et al.,2011; Kuhnt et al., 2013, Petersen et al., 2016, Rathburn et al., 2018; Richirt et al., 2019, Lu et al., 2022). Over the past decade, the porosity and pore density of foraminifera has evolved from a qualitative indicator for redox conditions towards a quantitative proxy.

Shell porosity as an oxygen proxy has received less attention in planktic foraminifera. Bé (1968) initially interpreted the porosity of planktic foraminifera as a potential temperature proxy due to a significant correlation between the porosity and latitudinal temperature gradients. A correlation between temperature and porosity was more recently validated in culture, with higher porosity interpreted as the result of the increase in metabolic rates with increasing temperature (Burke et al., 2018). Differences in porosity have also been observed with oxygen (Kuroyanagi et al., 2013; Davis et al., 2021). Notably, Kuroyanagi et al. (2013) report smaller pores in the final spherical chamber of the shallow, symbiont-bearing foraminifer *Orbulina universa* cultured under low-oxygen conditions. Davis et al. (2021) report higher porosities in the youngest chambers of deep-dwelling, low-oxygen affiliated species *Globorotaloides hexagonus*. The differences in these observations could reflect either inter-species variability, ecological specificity, or a difference in the metric (pore size vs porosity) used.

**Table 2 Foraminiferal morphometrics that can be assessed to estimate past oxygen concentrations. Morphometrics are divided by benthic and planktic foraminifera. The column "Low-oxygen conditions" refers to the response of the correspondent morphometric under low-oxygen concentrations. "Other controlling factors" list other environmental parameters that might influence the morphometric.**

| Foraminifera | Morphometric | Low-oxygen conditions | Other controlling factors | Example species |
|---|---|---|---|---|
|  |  |  |  |  |



| BENTHIC | Pore Density/ Porosity | High | Temperature (affecting metabolic rates); nitrate availability; age (decrease with older chambers) | Infaunal: *Bolivina spissa, Bolivina seminuda,* (Glock et al., 2011); Epifaunal: *Cibicidoides* and *Planulina* spp. (Rathburn et al., 2018; Lu et al., 2022) |
|---|---|---|---|---|
| | Size | Small (between different species); small or large (within same species) - respondent but variable direction of effect | Nitrate and food availability; depth in sediment (increase with depth) | The small *Nonionella stella* dominate low-oxygen sediments of the Santa Barbara Basin (Bernhard et al., 1997); *Bolivina spissa* adults (larger size) tolerate lower oxygen concentrations deeper in Peruvian margin sediment than juveniles (smaller size) (Glock et al., 2011). *Bolivina* (*seminuda, spissa, argentea, subadvena*), *Takayanagia delicata*, off California (Tetard et al., 2017; Moffitt et al., 2014; Ohkushi et al., 2013) |
| | Circularity | Size and circularity used for MARIN index to reconstruct oxygen: Low-oxygen: small size and elongated specimen (low MARIN); High-oxygen: big size and round specimen (high MARIN). | | Low-oxygen: elongated specimen (bolivinids, buliminids); high-oxygen: Cibicidids, planulinids…(Tetard et al., 2021) |
| | Surface Area:Volume Ratio | High or low - respondent but variable direction of effect | Age (increase with older chambers) Pollution | *U. peregrina, B. tunata,* and *L. psuedobeyrichii* have lower SA:V during low-oxygen period in Gulf of Alaska (Belanger, 2022) |



| | Size-Normalized Weight | Uncertain | | |
|---|---|---|---|---|
| | Thickness | Unknown | acidification | *Elphdium clavatum* (Choquel et al., 2023) |
| **PLANKTIC** | Pore Density/ Porosity | High or low - respondent but variable direction of effect | Temperature (affecting metabolic rates) | Low in *Orbulina universa* (Kuroyanagi et al., 2013)*;* High in *Globorotaloides hexagonus* (Davis et al., 2021) |
| | Size | Small or large (within same species) - respondent but variable direction of effect | | |
| | Circularity | Unknown | | |
| | Surface Area:Volume Ratio | High or low - respondent but variable direction of effect | | Lower for *G. hexagonus* in lowest oxygen conditions of eastern tropical North Pacific (Davis et al., 2021) |
| | Size-Normalized Weight | Low (high dissolution) | Mainly controlled by carbonate chemistry | *G. sacculifer*, *P. obliquiloculata*, and *N. dutertrei* (Broecker and Clark, 2001) |
| | Thickness | Unknown | | |





Several equations are available to calculate environmental oxygen and [NO$_3^-$] concentrations, using the porosity or pore density of benthic foraminifera. Rathburn et al. (2018) found that the porosity (Pore%) of *Cibicides* spp. and *Planulina* spp. within a 5000 μm$^2$ window at the center of the ultimate or penultimate chamber can be used to calculate bottom water oxygen concentrations ([O$_2$]$_{BW}$) according to equation 6.7.1:

*Equation 6.7.1.: [O$_2$]$_{BW}$ = $e^{((Pore\%-47.237)/(-8.426))}$*

In the same way the pore density (PD; in pores per μm$^2$) of *Cibicides* spp. and *Planulina* spp. within a 5000 μm$^2$ window at the center of the ultimate or penultimate chamber on the spiral side can be used with slightly lower accuracy (see Eq. 6.7.2; Glock et al., 2022):

*Equation 6.7.2.: [O$_2$]$_{BW}$ = $e^{((PD-0.008[+/-0.0002])/(-0.00142[+/-0.00006]))}$*

Within a limited [O$_2$]$_{BW}$ range of 2 - 14 μmol kg$^{-1}$, the PD of *Planulina limbata* on a size normalized area of the older part on the spiral side can be used to calculate [O$_2$]$_{BW}$ with higher accuracy according to eq. 6.7.3 (Glock et al. 2022):

*Equation 6.7.3: [O$_2$]$_{BW}$ = -6027[+/-652]·PD + 22.0[+/-1.7]*

Glock et al. 2011 & 2018 found that the PD of some denitrifying benthic foraminifera can be used to calculate bottom water NO$_3^-$ concentrations ([NO$_3^-$]$_{BW}$). The most recent equations to reconstruct [NO$_3^-$]$_{BW}$ have been found for *Bolivina spissa* and *Bolivina subadvena* (Govindankutty-Menon et al., 2023). [NO$_3^-$]$_{BW}$ can be calculated using the PD from a size-normalized area that includes the ~10 oldest chambers according to eq. 6.7.4:

*Equation 6.7.4: [NO$_3^-$]$_{BW}$ = -3896[+/-350]·PD + 61[+/-1]*

### 6.7.2.2 Size and Morphotype

A predominance of smaller benthic foraminifera species in low-oxygen environments has frequently been observed (Bernhard, 1986; Sen Gupta & Machain-Castillo, 1993; Bernhard & Sen Gupta, 1994). From the perspective of gas exchange, a decrease in size is an efficient way to increase shell surface area/volume ratio and thus, maximize the relative surface available for gas diffusion. However, this trend is not universally observed within species. Some studies report no consistent relationships between specimen size and oxygen levels (Keating-Bitonti & Payne et al., 2017). Some low-oxygen adapted species, such as *Uvigerina peregrina* and *Buliminella tenuata* even seem to increase in size and decrease in surface area/volume ratios in lower oxygen settings (Keating-Bitonti & Payne et al., 2017; Davis et al., 2021; Belanger, 2022). This counter-intuitive observation may be explained by reliance on nitrogen rather than oxygen for respiration (Glock et al., 2019), or the influence of high food availability (Belanger, 2022). Starting in the 1990s, multiple authors mention the relationship between test morphotype or shape of benthic foraminiferal tests and environmental parameters such as bottom water oxygenation (e.g., Corliss, 1991; Kaiho, 1994; Kaiho et al., 2006). These observations were combined with size metrics to develop the Major Axis and Roundness INdex (MARIN; *Equation 6.7.5* ; Tetard et al., 2021).

*Equation 6.7.5:          MARIN = Major Axis x Roundness*





The "Major Axis" corresponds to the primary axis of the best fitting ellipse. The
Roundness can be calculated according to Eq. 6.7.6:

*Equation 6.7.6:*        *Roundness = 4 x Area x $\pi^{-1}$ x Major Axis$^{-2}$*

Tetard et al. (2021) calibrated the MARIN as an oxygen proxy for the Eastern North Pacific according to Eq. 6.7.7:

*Equation 6.7.7:*        *$[O_2]_{BW} = 0.0000266$ x $exp^{0.0354\ x\ MARIN}$*

### 6.7.2.3 Size-Normalized Weight and Dissolution

Size-normalized weight (SNW) is a metric derived from normalizing the weight of individual or pooled shells by their length, area, or volume. It serves as a measure of how heavily calcified a foraminifera is and has been frequently linked to carbonate ion concentration in both pelagic and benthic environments. High input of organic matter results in higher oxygen consumption and release of dissolved inorganic carbon (DIC) by remineralization. This resulting coupling between oxygen and the carbonate system in many marine environments makes SNW worth mentioning here. While the direct driver of SNW (and shell dissolution) is likely carbonate chemistry, it could act as a viable supporting proxy. Lohmann (1995), followed by Broecker & Clark (2001), proposed the use of planktic foraminiferal shell weights to assess dissolution, and therefore bottom water carbonate chemistry. The use of shell weights as a carbonate chemistry (and, indirectly, oxygen) proxy, however, was rapidly complicated by evidence which shows that carbonate ion concentration influences SNW of planktic foraminifera during growth as well (e.g., Bijma et al., 1999; Barker and Elderfield, 2002; de Moel et al., 2009; Moy et al., 2009; Manno et al., 2012; Marshall et al., 2013). Results from size-normalized weight studies in foraminifera from low-oxygen, high-carbon environments have been equivocal in benthic foraminifera (Davis et al., 2016), but show some promise when applied to planktic foraminifera (Davis et al., 2021). Use of micro-CT to differentiate dissolution distinguishable from initial weight (Iwasaki et al., 2019) may facilitate the application of SNW as an indirect proxy for oxygen more feasible in the future.

### 6.7.3 Analyses and Required Resources

Sample preparation normally commences comparable to that of foraminiferal assemblages (see Section *6.6.4*). Morphometric analyses then proceed using one of several microscope and imaging approaches. In most cases, analyses of basic morphology (size, circularity, ornamentation, and sometimes porosity, pore size and pore density) can be carried out using a lighted stereo microscope, equipped with a camera and/or micrometer. This approach can also be automated, using a motorized microscope stage and image acquisition and processing software (e.g., NI Vision software, ImageJ, the R package *forImage*, Freitas et al., 2021) for automatically reconstructing and measuring pore density, pore surface area, volume and various test measurements (e.g., ImageJ, the MorFo_.ijm plugin available at https://doi.org/10.5281/zenodo.4740079 (Tetard et al., 2021; Freitas et al., 2021). Higher resolution analyses, or precise measurements of small features such as pores, sometimes require



SEM imaging. The SEM images usually include an image of the entire specimen, and in some processing methods, also need to include higher magnification images of a specific chamber. Processing the SEM images can be performed using open-source software ImageJ (https://imagej.nih.gov/ij/download.html) (Petersen et al., 2016) or Adobe Photoshop and ArcGIS software

(Rathburn et al., 2018). A detailed manual for semi-automated pore measurements of benthic foraminifera can be found in Petersen et al. (2016).

As a result of the light resource requirements, these analyses can be made both relatively low-cost and highly accessible. In addition, because most SEM analyses are non-destructive, specimens can be re-used in geochemical analyses, which provides potential for a multi-proxy approach to paleo-oxygen reconstruction.

**6.7.4 Recent Advancements**

While interest in studying foraminiferal morphology in 3D emerged in the mid-20th century (Bé et al., 1969; Schmidt, 1952), quantifications and paleoclimatic reconstructions using such methods have become possible only recently. Since the pioneering µCT work of Speijer et al. (2008), the number of studies dealing with 3D reconstructions of foraminiferal tests is increasing towards a variety of ends such as taxonomy and ontogeny (e.g., Briguglio et al., 2011, Görog et al., 2012, Schmidt et al., 2013,

Caromel et al., 2016, 2017; Burke et al., 2020), ocean acidification and test dissolution processes (e.g., Johnstone et al., 2010, 2011, Iwasaki et al., 2015, 2019, Prazeres et al., 2015, Ofstad et al., 2021, Kuroyanagi et al., 2021, Charrieau et al., 2022, Choquel et al., 2023), effects of temperature (e.g., Kinoshita et al., 2021, Titelboim et al., 2021), and paleoclimate reconstruction (e.g., Fox et al., 2020, Zarkogiannis et al., 2021, Todd et al., 2020, Schmidt et al., 2018). These studies have been focused mainly on planktic and tropical larger benthic foraminifera, with a recent study by Choquel et al. (2023)

expanding this research to benthic foraminifera.

Foraminiferal 3D reconstruction is a promising, non-destructive approach for accessing the morphology of the entire shell (inner and outer walls). However, it remains technically and methodologically challenging as well as costly to scan a large number of shells, with high resolution (< 1µm). To date "conventional" µCT scanners have primarily been used, supplemented with particle accelerator facilities, and Atomic Force Microscopy (AFM). Studies using 3D foraminifera constructions with

scanner-based or synchrotron light-based µCT were reviewed in Choquel et al. (2023). Most studies reconstructing foraminifera in 3D are performed with costly software such as Avizo (e.g., Fox et al., 2020), Amira (e.g., Schmidt et al., 2013), Molcer Plus and ConeCT express (e.g., Iwasaki et al., 2015). These software packages are adapted to the problems of test reconstruction but also can induce limitations on access to data processing and lack guidelines on how to analyse test morphometrics. Some authors are developing 3D post-data analyses with free software such as ImageJ/Fiji (Belanger et al.,

2022), Meshlab (Choquel et al., 2023) or Gwyddion (Giordano et al., 2019).

Reconstruction of pore patterns in 3D is challenging. Few studies have addressed porosity from µCT images and only from planktic foraminifera (Burke et al., 2018; Davis et al., 2021). Davis et al. (2021) demonstrated that in *Globorotaloides hexagonus* the porosity of the most recent chamber measured by CT scans and light microscope images capture the same trend, however the porosity from CT images is higher. This difference could be due to the accuracy of the pore segmentation (the




delineation of the automatic pore contour) related to the lower resolution of the µCT images compared to conventional microscopy.

Many efforts have been made in the last decade to automate or semi-automate the acquisition of pore measurements (number of pores, pore density, and porosity). Automated pore measurements have the advantage of acquiring data rapidly, facilitating the production of a large amount of representative data (Kuhnt et al., 2014). These automatic methods are often made from
SEM images of a small part, or fragments of the test, to limit the pore deformation linked to the test curvature. The difficulty lies in standardizing pore measurements between individuals of the same species, e.g., *Ammonia tepida* (Petersen et al., 2016; Giordano et al., 2019), *Orbulina universa* (Morard et al., 2009), or *Bolivina seminuda* (Tetard et al., 2017). The pores are mainly studied with a 2D view but 3D analyses allow access to more parameters such as pore depth and roughness (Giordano et al., 2019). Interest in studying 3D pore patterns from CT scans is increasing (Burke et al., 2018; Davis et al., 2021). Indeed,
the 3D reconstruction of the test is a promising tool that could give us access to the pore patterns chamber by chamber, and especially be adaptable to different shapes of the tests.

### 6.7.5 Proxy drivers

While porosity and/or pore density are empirically useful proxies in some species of foraminifera, understanding the parameters that directly drive this correlation continues to be a work in progress. Several authors have hypothesized that larger
pores act to facilitate increased gas exchange across the shell in low-oxygen environments (Leutenegger and Hansen, 1979; Corliss, 1985). A clustering of mitochondria behind the pores and the exchange of labelled $CO_2$ through pores suggest that pores are involved in respiratory processes in several species (Leutenegger and Hansen, 1979; Bernhard et al., 2010). Leutenegger & Hansen (1979) argues that the clustering of mitochondria below the pores will create a deficiency of oxygen, and thereby a diffusion gradient across the pores. *Patellina corrugata* has been shown to actively pump dissolved organic dyes
through its pores into the cytoplasm (Berthold, 1976). Moreover, pores have been found adequate for the exchange of gasses in both low and high oxygen conditions (Moodley & Hess, 1992). A gas exchange function would lead to a prediction of increased porosity and/or pore density under conditions of increased demand for oxygen diffusion due either to increased metabolic demand or decreased oxygen.

Complicating this interpretation, studies from the early 21st century revealed that not all foraminifera are reliant on oxygen
for respiration. Several benthic species have been shown to denitrify, either innately, or with the help of symbionts (Bernhard et al., 2012a; Bernhard et al., 2012b; Høgslund et al., 2008; Piña Ochoa et al., 2010; Risgaard-Petersen et al., 2006, Woehle et al., 2018&2022; See Section 6.6.6.1). Some denitrifying foraminifera such as *Bolivina spissa* most likely take up nitrate through pores and the pore density of denitrifying species can be used as a paleoproxy for nitrate, with which it correlates more strongly than oxygen (Glock et al., 2011; 2018; 2019). Other potential functions of surface pores include taking up dissolved
organic matter as food resources and releasing metabolic $CO_2$ (Glock et al., 2012).





Direct drivers of porosity may therefore be species and environment specific. For example, epifaunal or very shallow infaunal species such as *Cibicidoides* and *Planulina* spp. do not live or migrate to anoxic pore waters and no studies have suggested their use of nitrate for respiration. Thus, it can be inferred that bottom water nitrate does not influence the porosity of these species (Rathburn et al., 2018). Though, there is a possibility that some *Cibicides* spp. might be able to denitrify under severe

oxygen depletion, since they cluster next to the known denitrifiers within their phylogenetic tree (Woehle et al., 2022). Another recent study from the Arabian Sea found no significant correlation between surface porosity of *Cibicidoides* spp., dissolved organic carbon, and $CO_2$ concentrations in the seawater, suggesting that bottom water oxygen is likely the major control on surface porosity for *Cibicidoides* spp. (Lu et al., 2022).

**6.7.6 Influence of temperature, ontogeny, and dimorphism on morphological characteristics**

There are several other parameters that can influence porosity of foraminifera. Some environmental factors in addition to oxygen concentrations, that are known to correlate with the porosity or pore density of foraminifera, are temperature and nitrate concentration (Bé, 1968, Glock et al., 2011, Kuhnt et al., 2013, Burke et al., 2018). Temperature might influence the porosity via changes to metabolic rates and oxygen solubility in the environment, while nitrate is an alternative electron acceptor for denitrifying foraminifera (Risgaard-Petersen et al., 2006; Piná-Ochoa et al., 2010, Woehle et al., 2018, Glock et al., 2019).

Since these parameters often co-vary, it can be difficult to unravel the main factors that control porosity (see below).

Ontogeny can influence foraminiferal morphology in various ways. Usually, surface to volume ratios decrease as foraminifera grow. This results in an increase of pore density and porosity in the youngest chambers, as the organism compensates for decreased surface to volume ratio (Glock et al., 2011). During ontogeny, benthic foraminifera can migrate in the sediment (Glock et al. 2011) and planktic foraminifera in the water column (Hemleben et al., 1989; Schiebel and Hemleben, 2017;

Meiland et al., 2021). For example, the size (and likely age) of living *Bolivina spissa* from the Peruvian OMZ increases with sediment depth (Glock et al., 2011). Nevertheless, there is no significant correlation between pore density in *Bolivina pacifica* and its inhabited depth in sediments from the Arabian Sea, which indicates only minimal ontogenetic effects on the living depth of this species (Kuhnt et al., 2013). However, the study focused on analyzing porosity in small areas of the shell, which may reduce ontogenetic effects, if parts of the test from a similar ontogenetic state are analyzed (Kuhnt et al., 2013).

Generational dimorphism may also impact porosity. At least in some species there are systematic differences in the pore patterns between megalospheric and microspheric specimens (Fig. 15), and some studies exclusively focused on megalospheric specimens (Glock et al., 2011, 2018).

One final consideration when analysing foraminiferal porosity is shell stability (Richirt et al., 2019). As porosity increases ontogenetically (Glock et al., 2011), the last chamber usually shows the highest porosity. The last chamber is also usually the

thinnest, due to the laminar calcification mechanism in rotaliid species (Erez, 2009), and may be broken in many fossil specimens. There might be different strategies to preserve stability during increasing porosity. One strategy is to build larger but fewer pores (Richirt et al., 2019). Another strategy might be to increase wall thickness.



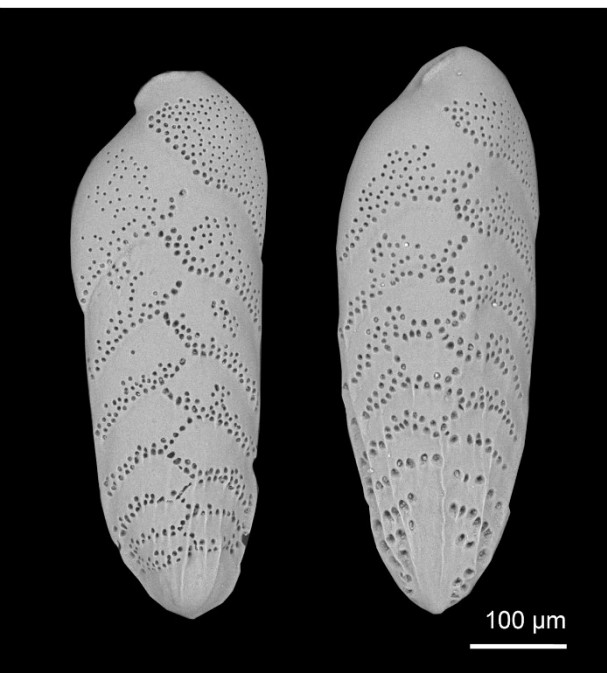


**Figure 15: Comparison of the pore patterns of a megalospheric (left) and microspheric (right) specimen of *Bolivina spissa*. Note the larger but fewer pores in the old parts of the test of the microspheric specimen (bottom part of the image, close to the proloculus). Location: Mexican Margin. Image taken on a Hitachi Tabletop SEM TM4000 series at Hamburg University.**

### 6.7.7 Marine archives and limitations


Similar to oxygen assessments based on benthic foraminiferal assemblages, morphometrics can potentially be used wherever marine carbonates are preserved. Features such as size and test morphology should be robust to minor dissolution, and image analyses typically should reveal if extensive dissolution has occurred. Environments corrosive enough to remove whole tests could lead to preferential dissolution of smaller tests and/or enlargement of pores, but this has yet to be directly tested in the context of low-oxygen proxies; proxies such as porosity, pore density and MARIN have not been directly assessed for susceptibility to dissolution.


The correlation between pore density/porosity and oxygen concentrations or other environmental parameters is species specific (Kuhnt et al., 2013; 2014; Glock et al., 2011; 2022). The distribution of foraminiferal species is often restricted to a certain oxygen range and thus, the species specific calibrations often cover only a limited oxygen range (Kuhnt et al., 2013 & 2014; Tetard et al., 2017 & 2021, Glock et al., 2011; 2022; Section 6.6.2). One potential solution to this limitation can be a multi-species pore density/porosity calibration. For example, the global multi-species calibration of *Cibicidoides* and *Planulina* spp. suggests a strong negative logarithmic relationship between porosity and bottom water oxygen from oxygen concentrations as






low as 2 µmol to completely oxygenated environments but the correlation is very flat at oxygen concentrations >100 µmol kg$^{-1}$ (Rathburn et al., 2018; Lu et al., 2022). The porosity of these species is usually >~10% when bottom water oxygen is < 100 µmol kg$^{-1}$. In oxic environments, few or no pores are found on the surface (Rathburn et al., 2018). The porosity proxy thus is most useful to indicate low bottom water oxygen conditions (Fig. 1).

The porosity of *Bolivina seminuda* has been suggested as an oxygen proxy for lower oxygen environments (~ 1 to 45 µmol after Tetard et al., 2017; 2021). Though, *B. seminuda* is a denitrifying species and it is possible that differences in oxygen concentrations play only a minor role regarding its porosity (Piná-Ochoa et al., 2010; Glock et al., 2019). Multiple environmental parameters that have been shown to correlate with porosity or pore density of foraminifera such as oxygen, nitrate and temperature often covary. This can result in significant correlation of the pore characteristics with multiple parameters (Glock et al., 2011; Kuhnt et al., 2013). Oxygen and nitrate are coupled to both denitrification and remineralization. Nitrate loss through denitrification is increased when oxygen is depleted, while remineralization consumes oxygen and increases nitrate concentrations (Andersen and Sarmiento, 1994; Johnson et al., 2019). Similar opposing trends are found for the correlation between oxygen and temperature. Oxygen solubility decreases with increasing temperatures and both parameters often covary with water depth (Keeling & Garcia, 2002; Schmidtko et al., 2017). In addition, temperature has an influence on metabolic rates, which might be a factor influencing foraminiferal porosity (Burke et al., 2018). For now the covariation of all these parameters limits interpretation of porosity proxies but future studies using controlled laboratory cultures might unravel their specific influences.

### 6.7.8 Future Directions

#### 6.7.8.1 Emerging Developments

High-throughput imaging and other forms of automation are rapidly increasing the scope of what can be done with morphological proxies, but bring new challenges as well. Automation reduces historically labour intensive work. This means that larger sample sizes, higher-resolution records, and attention to an increasing number of species and environments are becoming more feasible. However, key questions remain, such as how automated analyses compare to manual analyses, and what trade-offs are associated with accessibility versus accuracy. One concern is that the accuracy of high-throughput automated image analyses may suffer due to a decrease in human oversight, though this might improve with further development of algorithms. High-throughput methods and use of traditional equipment, such as optical microscopy, may also support different types of accessibility, with the former saving labour while the latter saving instrumental cost.

Other directions include wider application of existing methods and ground-truthing through laboratory culture. Application of morphological methods developed in benthic foraminifera to other organisms would also increase the number of environments from which oxygen can be constrained. This most readily applies to planktic foraminifera, which would expand proxies into the pelagic realm, but may also apply to other hard-bodied organisms such as ostracods, in which oxygen-morphology associations have not yet been systematically explored.





As mentioned above, the stability of tests is an important factor when studying foraminiferal porosity and pore density (Richirt et al., 2019). One factor, related to test stability, that is not analyzed yet, is the thickness of the walls. The effects of mechanical stress on the morphological characteristics such as porosity and wall thickness are currently unknown. Other factors to be considered in future studies include variations in mechanical stresses in the environment induced by varying sediment grain size, bioturbation and the intensity of bottom water currents. Future directions could include culturing experiments to isolate

the influence of different environmental parameters on the pore density/porosity of both benthic and planktic species, including both denitrifying and oxygen-respiring species. Previous culture studies have demonstrated pore plasticity responding to environmental conditions within the lifespan of a single individual in multiple benthic (Sliter, 1970; Moodley and Hess, 1992) and planktic (Bijma et al., 1990; Allen et al., 2008; Kuroyanagi et al., 2013; Burke et al., 2018) species. Porosity changes during the foraminifera lifetime in a potential metabolic response to environmental drivers (Kuroyanagi et al., 2013; Burke et

al., 2018).

    The evolution of 3D methods such as micro-computer tomography (μCT) provides access to morphological features of the full test. However, the resulting datasets are very large. This can be problematic, since full datasets cannot be easily shared or published and storage space can be a limited resource. Further work is needed to automate and streamline some data handling and processing.

Finally, little attention has been given to studying ecophenotypic variability, or the potential for adaptive responses to environmental forcing, in the frequency of ornamentations and test deformations in response to oxygen depletion. Lutze (1964) and Harman (1964) showed that bolivinids from lower oxygen sites in the Santa Barbara Basin typically have less ornamentations such as spines, costae and keels. In addition, growth disruption and test deformation can appear under unfavorable environmental conditions (Lutze, 1964). Future research might address this issue and include systematic studies

on the frequency and size of ornamentations and test deformation under variable oxygen concentrations.

### 6.7.8.2 Open Questions: Resolving methodological differences

    The literature is diverse for methods regarding the determination of foraminiferal pore characteristics, such as porosity, pore size and pore density. A relatively widespread method is focusing on the center of the ultimate or penultimate chambers and using a small window to approximate a flat surface (Kuhnt et al. 2013 & 2014; Petersen et al., 2016; Rathburn et al., 2018;

Richirt et al., 2019; Glock et al., 2022). An alternative method suggests using a larger size-normalized part of the older chambers (Glock et al., 2011, 2018, 2022). Another approach is automated image acquisition and analysis of shards after cracking open the shell, where penultimate chamber vs whole test porosity has been investigated (Tetard et al., 2017). Finally, one study used atomic force microscopy to automatically analyze foraminiferal morphometrics (Giordano et al., 2019). Each method has advantages and disadvantages (see 6.7.2). A recent study on the pore density of epifaunal *Planulina limbata* found

the best correlation between oxygen and pore density by using a size-normalized area on the older parts of the spiral side (Glock et al., 2022). In other epifaunal foraminifera, Rathburn et al. (2018) found that there is a better correlation between porosity and oxygen than between the pore density and oxygen. Finally, to minimize problems through dissolution effects or



overgrown pores of planktic foraminifera, Constandache et al. (2013) suggested breaking the shells and determining the pore characteristics on the inner surface. The wide variability in methods that have been used in existing studies shows a need for the development of a common approach. The ongoing automation of data acquisition may provide a suitable way to achieve this in future work. Though, the approaches may vary a bit, depending on the species specific shape of the shells (i.e., spatulate, planconvex, elliptic etc.).


**Table 3 Different methods to determine pore characteristics (e.g., shell porosity and pore density) of foraminifera with a list of different advantages and disadvantages of those methods.**

| Description of method | Advantages | Disadvantages | References |
|---|---|---|---|
| Focusing on small window with smooth surface in center of ultimate/penultimate chamber | Relatively fast; Minimizes artifacts due to curvature of the specimens; Normalizes regarding ontogenetic stage; Negates problems with overgrown pores | Dataset is limited due to small window size; Ultimate and penultimate chambers usually have highest porosity, which causes restrictions by the test stability | Kuhnt et al., 2013 & 2014; Petersen et al., 2016; Rathburn et al., 2018; Richirt et al., 2019; Glock et al., 2022 |
| Using a larger size normalized area on the older parts of the test | Normalization for ontogenetic effects; Larger datasets per specimen; Lower porosity in these parts causes test less stability restrictions and thus porosity might be better adapted to environmental changes | Possible artifacts by overgrown pores and curvature of the test; More effort to acquire the data | Glock et al., 2011, 2018 & 2022 |
| Automated image acquisition and analysis of shards from crushed foraminifera | Large datasets with relatively low effort | Method is "destructive" | Tetard et al., 2017 |
| Automated morphometric analyses using atomic force microscopy | Most metadata of all methods, including depth and 3D shape of the pores | Accessibility of atomic force microscopy | Giordano et al., 2019 |



| Analysis of porosity from the inside after breaking the test | Avoid problems with pores that are overgrown or show evidence for dissolution from the outside | Method is "destructive" | Constandache et al., 2013 |
| --- | --- | --- | --- |
| 3D image of the whole test using synchrotron x-ray | All pores of the test can be counted and calculated the surface area in 3D and pore sizes | Beamtime is constrained | Choquel et al., 2023 |


## 6.8 Benthic foraminifera carbon isotope offsets

### 6.8.1 Theory & proxy driver(s)

The carbon isotopic offset between specific benthic foraminifera species ($\Delta\delta^{13}C$) can be used to reconstruct bottom water oxygen concentrations. Application of the proxy relies on the principle that oxic remineralization of organic matter releases

isotopically light DIC into the pore waters. Within the top few centimeters of the sediment, aerobic respiration of organic matter dominates. This generates a $\delta^{13}C$ gradient between bottom and pore waters from the sediment-water interface to the anoxic boundary (McCorkle et al., 1985). It is thought that greater aerobic remineralization in pore waters, associated with higher bottom water oxygen concentrations, enhances this $\delta^{13}C$ gradient, and the availability of oxygen in marine pore waters is set by downward diffusion of bottom water oxygen across the sediment-water interface (McCorkle et al., 1985).


The basis of the $\Delta\delta^{13}C$ proxy is the difference in $\delta^{13}C$ between epifaunal and deep infaunal benthic foraminifera species, that theoretically reflect the carbon isotopic composition in their habitats (Eq. 6.8.1). The carbon isotopic composition of epifaunal foraminifera such as *C. wuellerstorfi* reflects that of DIC in bottom water, and the carbon isotopic composition of deep infaunal foraminifera of the genus *Globobulimina* reflects that of DIC in pore waters near the anoxic boundary (Duplessy et al., 1984;

Zahn et al., 1986; Fontanier et al., 2002; Geslin et al., 2004; Schmitter et al.,2017).

*Equation 6.8.1:*          $\Delta\delta^{13}C = \delta^{13}C_{C.\ wuellerstorfi} - \delta^{13}C_{Globobulimina\ spp.}$



Robust quantification of bottom water oxygen concentrations based on the $\Delta\delta^{13}$C rests on three fundamental assumptions. First, foraminiferal species respectively record the $\delta^{13}$C of the DIC of bottom waters and pore waters at the anoxic boundary within subsurface marine sediments. Second, degradation of organic matter above the sedimentary anoxic boundary and associated deviation of pore water $\delta^{13}$C of DIC from bottom waters is predominantly driven by aerobic respiration. Lastly, pore water $\delta^{13}$C gradients in DIC are largely determined by aerobic respiration of organic matter, which is mediated by the downward supply (i.e., diffusion) of dissolved oxygen from bottom waters, and thus directly scales with the availability (i.e., concentration) of dissolved oxygen in bottom waters. If overlying bottom waters are characterized by low-oxygen concentrations, a balance of consumption of oxygen through mainly aerobic respiration and diffusion of oxygen from the overlying bottom waters is reached at shallow sediment depths with small deviations of pore water $\delta^{13}$C of DIC at the anoxic boundary from the $\delta^{13}$C of DIC in bottom waters. Low $\delta^{13}$C gradients are thus associated with low bottom water oxygen levels (Fig. 16). The greater the oxygen concentration in overlying bottom waters, the greater the release of low $\delta^{13}$C DIC into pore waters during respiration of organic matter, leading to a larger $\delta^{13}$C difference between the DIC at the sediment-water interface and the DIC at the anoxic boundary, and a larger $\delta^{13}$C gradient.

There is a strong relationship between $\delta^{13}$C of seawater DIC and oxygen in the water column today (Hoogakker et al., 2016). However, epifaunal benthic foraminifera $\delta^{13}$C alone cannot be used to reconstruct past bottom water oxygen due to large uncertainties relating to preformed $^{13}$C, air–sea fractionation, mixing with other water masses and terrestrial biomass changes (Lynch-Stieglitz et al., 1995; Schmittner et al., 2013; Gruber et al., 1999; Curry and Oppo, 2005; Oliver et al., 2010). Instead, the carbon isotope gradient pairs epifaunal $\delta^{13}$C measurements with those of a deep infaunal species.

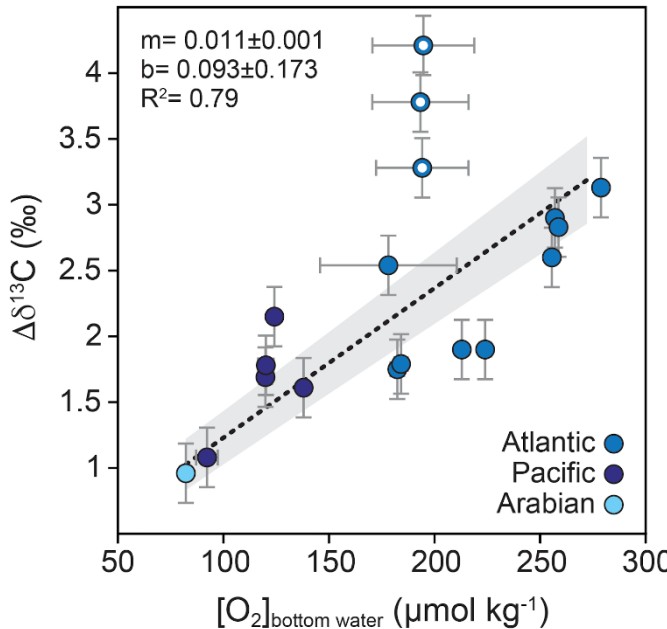




**Figure 16: Relationship between Δδ[13]C of *C. wuellerstorfi* versus *Globobulimina* (spp.) and bottom water oxygen. White circles**
**represent outliers that were excluded from the regression. Details can be found in the supplementary information.**

### 6.8.2 History of development and use

McCorkle and Emerson (1988) and McCorkle et al. (1990) first proposed that bottom water oxygen concentrations could be quantified using δ[13]C in benthic foraminifera. They recognized that as remineralization of organic matter in pore waters consumes oxygen and releases isotopically light DIC, in relatively fixed ("Redfield Ratio") proportions, δ[13]C in pore waters
relative to overlying bottom waters will reflect the amount of oxygen consumed in pore waters as a function of bottom water oxygen concentrations. They further suggested that paired specimens of benthic foraminifera that record bottom water and deep pore water carbon isotopes could be used to record this gradient, and thus bottom water oxygen concentrations in the past.

McCorkle et al. (1990) verified that the Δδ[13]C of deep infaunal living benthic foraminifera *Globobulimina affinis* and the overlying bottom water is similar to the Δδ[13]C between the pore water around anoxic boundary and overlying bottom waters. Schmiedl and Mackenson (2006) showed that the Δδ[13]C between *Globobulimina* species and bottom water DIC correlates well with bottom water oxygen concentrations, and applied the derived relationship to reconstruct past bottom water oxygen in the Arabian Sea (Schmiedl and Mackensen, 2006). Hoogakker et al. (2015) used existing (Mackensen and Licari, 2004; Schmiedl
and Mackensen, 2006; Fontanier et al., 2008) and new data to show that Δδ[13]C of epifaunal *C. wuellerstorfi* and deep infaunal *Globobulimina* species from six locations of the Atlantic, Indian and Pacific Oceans show a similar relationship as Δδ[13]C estimates between bottom water and pore water around the anoxic boundary from these basins (but at different locations)(Fig 6.8.1). They showed that Δδ[13]C between *C. wuellerstorfi* and *Globobulimina* spp. is linearly correlated with bottom water oxygen levels in the range of 50–235 μmol kg[-1], and used this new calibration to reconstruct bottom water oxygen variations
in the North Atlantic. The proxy has been applied in several studies to quantitatively reconstruct past bottom water oxygen concentrations in various ocean regions, with an estimated total error of 17 μmol kg[-1], including the North Atlantic (Hoogakker et al., 2016; Thomas et al., 2022), South Atlantic (Gottschalk et al., 2016; Gottschalk et al., 2020a), Indian Ocean (Schmiedl and, Mackensen 2006; Gottschalk et al., 2020b) and Pacific Ocean (Hoogakker et al., 2018; Umling and Thunell, 2018; Jacobel et al., 2020). Figure 6.8.1 shows the most up-to-date core-top calibration between Δδ[13]C obtained from *C. wuellerstorfi* and
*Globobulimina* spp. and bottom water oxygen, with bottom water oxygen and new data points from Umling and Thunnel (2018), Jacobel et al. (2020), and Thomas et al., (2022), as detailed in the supplementary information.

### 6.8.3 Description of analyses and resources required

Stable carbon isotope ratios from epifaunal (*Cibicidoides* spp.) and deep infaunal benthic foraminifera (*Globobulimina* spp.) can be measured using a mass spectrometer, commonly a Thermo Kiel V automated preparation device coupled with a Thermo
Delta V Plus or Thermo MAT253 Mass Spectrometer. Measurements are calibrated to the international VPDB (Vienna Pee



Dee Belemnite) standard. The sample mass required for analysis varies and depends on the mass spectrometer setup. Modern mass spectrometers typically require 40–80 µg CaCO$_3$, but can provide measurements for samples as small as 5 µg CaCO$_3$. Specimen size does not seem to matter for the δ$^{13}$C of *C. wuellerstorfi* (Franco-Fraguas et al., 2011), whereas size effects on *Globobulimina* spp. are not yet explored. *Cibicidoides* spp. specimens tend to weigh more compared to *Globobulimina* spp.

specimens. Selecting foraminifera from the larger than 250 µm-fraction, about one to four specimens of *Cibicidoides* and two to twenty specimens of *Globobulimina* spp. are needed for stable isotope analysis. Dirty specimens may need some treatment prior to analyses, including crushing, ultrasonication and/or methanol rinses to remove clay particles.

The most commonly used deep infaunal foraminifera of the genus *Globobulimina* for bottom water oxygen reconstructions

via the Δδ$^{13}$C proxy is *Globobulimina affinis*. In the absence of *G. affinis*, other *Globobulimina spp.* have been used as mono- or multispecific samples (e.g., Hoogakker et al., 2018), assuming all share a similar deep infaunal depth habitat: *G. pacifica*, *G. turgida*, and *G. auriculata* (Supplementary Fig. S6.8.1). A deep infaunal species of the genus *Chilostomella*, a potential candidate to monitor conditions at the anoxic boundary, does not form its test in equilibrium with pore water DIC (McConnaughey et al. 1997; Nomaki et al., 2021). When *C. wuellerstorfi* specimen are absent, other foraminiferal species

thought to live epifaunally such as *Cibicides mundulus* (synonymously used with *Cibicides kullenbergi*) have also been used to reconstruct bottom water oxygen levels using the Δδ$^{13}$C proxy  (e.g., Gottschalk et al., 2016a, 2020; Bunzel et al., 2017; Lu et al., 2022) with some notable caveats for their use (Supplementary Fig. S6.81).

### 6.8.4 Proxy limitations

The infaunal habitat of *Globobulimina* spp. may extend below the depth of sedimentary anoxia in some locations (and/or time

periods) (Geslin et al., 2004). This may potentially lead to the incorporation of additional isotopically-light carbon generated at depth via anoxic processes, including denitrification by *Globobulimina* spp. and/or sulphate reduction (McCorkle and Emerson, 1988). Variations in depth habitat are of concern because denitrification and sulphate reduction are known to play a significant role in the remineralization of organic matter, releasing isotopically light carbon to pore waters after oxygen has been consumed. Sulphate reduction and other early diagenetic reactions are of particular concern in margin settings that are

shallower than 1500 m (Sarmiento and Gruber, 2002) as, in general, more organic matter is deposited in these settings. Sulphate reduction can lead to the shallowing of the early diagenetic zones (Egger et al., 2018) and an increased diffusive flux of DIC into the zone of aerobic respiration. Thus, there may be a variable contribution of anaerobic processes to the pore water DIC from which *Globobulimina* spp. calcify. If one or more of these influences are at play, Δδ$^{13}$C is expected to be elevated and the calibration would overestimate bottom water oxygen concentrations.


Recent work has shown that at least four species of *Globobulimina* spp. (including *affinis*, *pacifica*, *turgida* and *pseudospinescens*) concentrate nitrate, for use as an electron acceptor in the absence of oxygen (Risgaard-Petersen et al., 2006; Nomaki et al., 2015; Piña-Ochoa et al., 2010a; Piña-Ochoa et al., 2010b). Metabolic and genetic data corroborates the capability





of *Globobulimina* species to denitrify, one of the main reasons they are successful in anoxic settings (Piña-Ochoa et al., 2010b,

Woehle et al., 2018 & 2022). These results may imply that *Globobulimina* spp. may thrive and calcify successfully well below

the anoxia boundary, meaning they could be influenced by the addition of low-$\delta^{13}$C DIC during sulphate and nitrate reduction,

although to date there is no direct evidence for this. Furthermore, if *Globobulimina spp.* contribute isotopically-light carbon to

the sedimentary pore waters, this could potentially decouple the relationship between bottom water oxygen and $\Delta\delta^{13}$C. The

influence of shallow denitrification has been invoked to explain observations of inconsistent $\Delta\delta^{13}$C between five

contemporaneous records of $\Delta\delta^{13}$C from the eastern equatorial Pacific Ocean (Jacobel et al, 2020), and between co-located

records of U/Ba and $\Delta\delta^{13}$C, where U/Ba yields consistently lower estimates of bottom water oxygen concentrations (Costa et

al., 2023). Significant inconsistencies between core-top bottom water oxygen concentrations, determined using $\Delta\delta^{13}$C, and

measured bottom water oxygen have also been identified at some equatorial Pacific sites (Jacobel et al, 2020). Separately,

higher $\Delta\delta^{13}$C values found in the OMZ of the Arabian Sea (Lu et al., 2022) have been attributed to sulphate reduction in

sediments. Specifically, $\Delta\delta^{13}$C-based bottom water oxygen estimates at these sites are more than 60 μmol kg$^{-1}$ higher than

reconstructed using other proxies (i.e., benthic surface porosity, benthic I/Ca, and aU) (Lu et al., 2022). Indeed, the three

outliers in the updated $\Delta\delta^{13}$C - bottom water oxygen relationship in Fig. 16 are from areas where sulphate reduction is known

to play an important role (Bradbury et al, 2021; Thomas et al., 2022).

Another factor that may drive variations in $\Delta\delta^{13}$C independent from bottom water oxygen is changes in the carbon isotopic

composition of organic material ($\delta^{13}C_{org}$) that is remineralized in the pore space of marine sediments (Hoogakker et al., 2015).

A decrease in $\delta^{13}C_{org}$ would enhance $\Delta\delta^{13}$C and cause an apparent increase in reconstructed bottom water oxygen. It is therefore

important to assess $\delta^{13}C_{org}$ alongside $\Delta\delta^{13}$C-based bottom water oxygen quantifications.

Finally, although previous work has shown a generally strong correspondence between $\delta^{13}$C of DIC in bottom waters and $\delta^{13}$C

of *C. wuellerstorfi* (e.g., Schmittner et al., 2017), there is some evidence that seasonal pulses of organic carbon (the

phytodetritus effect) may decrease the $\delta^{13}$C of epifaunal species by as much as 0.4‰ (Zarriess and Mackensen, 2011), perhaps

due to the development of benthic 'fluff' layers in which *C. wuellerstorfi* calcify (Mackensen et al., 1993). This effect has not

been found in all locations experiencing seasonally-variable production (Corliss et al., 2006) emphasizing the need for further

work to develop the regional and/or time-variant conditions under which these effects develop. Insights into this open question

could be derived from core-top samples where organic carbon flux, $\delta^{13}C_{C.\ org,}$ $\delta^{13}C_{C.\ wuellerstorfi}$, and bottom water oxygen are

directly measured to evaluate and quantify relationships.

### 6.8.5 Species relevant for calibration

Application of the $\Delta\delta^{13}$C proxy is limited by the availability of epifaunal (*C. wuellerstorfi* or *C. kullenbergi*) and deep-infaunal

(*Globobulimina* spp.) species in the sediment samples. However, some epifaunal species can adopt a shallow infaunal habitat,

and can therefore be influenced by the pore water environment (Gottschalk et al., 2016b; Wollenburg et al., 2021). Temporal

variations in the $\delta^{13}$C offsets of *Cibicides* species and *C. wuellerstorfi* could be an indication of a change in habitat (e.g.,

Gottschalk et al., 2016b). If there is an indication of temporal variations in this offset, or information about offsets is



unavailable, the application of the $\Delta\delta^{13}C$ proxy based on assumingly shallow infaunal species such as *C. kullenbergi* may lead to a bias of bottom water oxygen concentrations towards lower values (e.g., Gottschalk et al., 2016a).

Because *Globobulimina* spp. has a deeper habitat compared with *C. wuellerstorfi*, there is the possibility that the measurements are age-offset. Upon death, sediment stirring through benthic organisms will mix the sediments with a bioturbation depth unique to the sedimentary environment at the time of deposition. It is unlikely that the two species in the fossil record maintain the depth offsets observed in living specimens. The comparison of the stable oxygen isotope records of both species is thus critical for ruling out or determining the appropriate, possibly time-variant, depth offset between species (Hoogakker et al.,
2015).

### 6.8.6 Future directions and open questions

Direct comparison and correlation between living and dead *C. wuellerstorfi* and *Globobulimina* spp. $\delta^{13}C$, measurements of bottom and pore water oxygen, and the $\delta^{13}C$ of bottom and pore water DIC would be ideal for rigorously quantifying calibration uncertainties. The calibration could also be strengthened by quantitative assessment of co-varying environmental parameters
such as the flux of organic carbon, the $\underline{\delta}^{13}C$ of $C_{org}$, and the influence of sulphate reduction and denitrification at calibration sites. This could be achieved through coring campaigns and culturing studies following the methods of Wollenburg et al. (2015).

Several possibilities exist for expanding the use of the $\Delta\delta^{13}C$ proxy. For example, the improvement of analytical techniques now allows for the analysis of single specimens with respect to oxygen and carbon isotopes (e.g., Ganssen et al., 2011).
Analysis of single specimen $\delta^{13}C$ may provide insights into the natural variability of $\delta^{13}C$ of communities and improve our interpretation of $\Delta\delta^{13}C$. Additionally, application of the $\Delta\delta^{13}C$ proxy could be expanded if other deep infaunal species are found to record pore water $\delta^{13}C$ of DIC at the anoxic boundary.

### 7 Concluding summary statement and future directions

In this review, we summarize the current state-of-knowledge about proxies for reconstructing Cenozoic marine oxygen levels. Sediments are the carriers for all proxies associated with seawater oxygen reconstructions. Sedimentological and other non-destructive methods, as well as presence, relative abundance and potentially trace element compositions of pyrite provide important information about depositional redox conditions.

Bulk geochemical methods are described that can be used to reconstruct bottom water redox/oxygen conditions, as well as
methods that involve fossil foraminifera abundance, appearance, and geochemistry:

1) Redox-sensitive elements that are preserved under various redox potentials have provided key insights into deep ocean oxygenation on a variety of timescales. However, challenges remain and redox element research continues to



refine the interpretations of these proxies by constraining variations of other environmental variables (notably the rain of organic carbon) that affect the redox state of sediments. Recent technical advances have allowed for the development of novel 'non-traditional' stable metal isotope systems, which open new possibilities for more quantitative redox reconstructions and towards globally integrated estimates of ocean oxygenation through time.

2) Lipid biomarkers provide a wealth of paleoceanographic information. Their source specificity and excellent preservation potential allow the detailed and comprehensive reconstruction of water column (and sediment) redox conditions. Taxonomically specific biomarkers are available for a range of microorganisms thriving in different ecological redox niches, providing insights into past changes in the ocean's carbon, nitrogen, and sulphur cycles. Instrumental advancements and increased resolution continue to widen the analytical window, reveal novel biomarkers, and – in combination with (meta)genomics – aid identification of source organisms. Moreover, biomarker proxies are becoming more and more quantitative and the community strives to develop tools that allow inferring absolute oxygen concentrations.

3) Bulk nitrogen isotopes offer insights into bacterial denitrification processes that are closely linked to water column oxygen concentrations below <5 µmol kg$^{-1}$. The strong isotopic discrimination by denitrifying bacteria can be measured in bulk sediments. Isotopic discrimination by denitrifying bacteria can also be measured in foraminifera-bound $\delta^{15}N$, and this method shows great promise for understanding dynamics of OMZs. We highlight the need of integrating biogeochemical models to refine interpretations of the nitrogen isotopic records.

4) Foraminifera trace elements, especially I/Ca, Mn/Ca and U/Ca show promise as proxies for reconstructing past oxygen conditions, within the constraints of the complexities arising from various environmental factors and potential interferences. I/Ca values are linked to the presence of $IO_3^-$ and its reduction to $I^-$ in low-oxygen settings. U/Ca utilizes the formation of authigenic U coatings on foraminifera tests buried in marine sediments. Higher U/Ca concentrations are indicative of reducing oceanic bottom water conditions. Higher Mn/Ca in foraminiferal calcite indicates increased free $Mn^{2+}$ incorporation under low oxygen bottom/pore water conditions. Foraminifera trace element proxies require careful consideration of carbonate chemistry, variable oxygen thresholds, vital effects, ontogenetic effects, and potential diagenetic effects that can distort the signals in the geologic record. Further work is needed to establish robust calibrations for the relationships between proxies and oxygen conditions in different environmental settings and for different foraminifera species.

5) Foraminiferal assemblages as paleoproxies have a long tradition, due to their rapid response to changing environmental conditions. Benthic foraminiferal assemblages are very sensitive to changes in bottom water oxygenation, due to the specific adaptations of some benthic species to $O_2$ depletion. Recent advances in understanding the anaerobic metabolism of some species and indications that they can calcify when exposed to anoxia, make foraminiferal microfossils important archives for past periods of $O_2$ depletion. Though, in some cases it can be problematic to decouple changes in bottom water oxygenation from changes in organic matter input. One bottleneck for using benthic foraminiferal assemblages is the high workload for sample processing, including picking of





foraminifera and taxonomic classification for each sample. The main advantage of this method are the low instrumental and resource requirements for this approach. Future directions include AI-based automation of species recognition, which will severely reduce the time-intensity of this approach; and the more routine use of planktic
foraminiferal assemblages to reconstruct $O_2$ concentrations in the water column.

6)    Various morphological features of foraminifera shells also reflect the environmental conditions in which they were built, including oxygen concentrations. Shell porosity has received increasing attention recently and seems to reflect (a) the availability of oxygen for oxygen-respiring species, and (b) nitrate availability for species specialized in denitrification. Although a lot of focus has been placed on morphological features of benthic foraminifera, recent
advances in understanding planktic foraminiferal morphology opens new windows for oxygen reconstruction within different layers of the water column. The recent improvement on automated image analysis facilitates the quick generation of large data-sets, while the usually non-destructive methods for image acquisition preserve the analysed specimens for further analyses. In addition, the development of image acquisition methods and broader availability of μ- and nanoCT techniques allow 3D analyses of specimens to further provide access to morphological details that
were hidden before.

7)    There has been significant progress in employing the carbon isotope gradient of epifaunal and infaunal benthic foraminifera proxy to more quantitatively reconstruct bottom water oxygen concentrations, since its first proposal in the late 1980s. Multi-proxy work has been key in identifying sources of proxy uncertainty that are currently not well quantified, and in highlighting depositional environments that may not be well suited for the application of this proxy,
specifically areas where sedimentary denitrification and sulphate reduction are prevalent. Our review emphasizes that $\Delta\delta^{13}C$-based reconstructions are likely to provide estimates that represent an upper bound of past bottom water oxygen concentrations. Further research into uncertainties has the potential to improve the quantitative nature of the proxy. Specifically, we recommend future work focus on the fidelity of different species in recording the $\delta^{13}C$ values of bottom and pore waters, the role of variations in the carbon isotope composition of organic carbon, and the
significance of biases arising from contributions from anaerobic metabolic processes, and how changes in the rain rate of organic carbon may influence the proxy.

Proxies are by definition indirect measurements, each with their own sources of uncertainty, biases, limitations, and drivers as detailed in the sections above. For this reason, we recommend applying a multi-proxy approach, in which two or more proxy
records are generated in tandem from the sample set. Ideally, the design of a multi-proxy study should incorporate multiple proxies for the same or related parameters with different sources of uncertainty.

Multi-proxy approaches are particularly appropriate in the field of paleo-oxygenation where most available proxies are semi-quantitative and cover different ranges of redox chemistries (Fig. 1), and in part may have only been recently developed. They may also differ in their drivers, with some proxies having multiple drivers, which may be independent of oxygen. The layering
of multiple, semi-quantitative proxies can allow researchers to assign more quantitative paleo-oxygen estimates (e.g., FB-$\delta^{15}$N



and planktic foraminiferal I/Ca in Hess et al., 2023), an exercise that may be pivotal in generating paleo-oxygen reconstructions that can inform models. The rapid development of novel paleo-oxygen proxies has been highly beneficial to the field. However, the limitations and uncertainties of more recently developed proxies need to be further explored. Multi-proxy approaches can serve to validate these proxies (e.g., comparing benthic I/Ca, U, and foraminifera porosity as bottom water oxygen proxies in

Lu et al., 2022), and increase our understanding of their application in the sedimentary record (e.g., $\delta^{13}$C and U in Jacobel et al., 2020). Finally, inclusion of multiple proxies may allow researchers to disentangle multiple drivers in the paleorecord, not only constraining oxygen, but related environmental factors, such as export productivity or carbon fluxes, redox structure of the water column and sediment, or depositional settings.


## 8 Recommendations for data management and transparency

The interdisciplinarity of the communities generating oxygen proxy data and model outputs as well as the presently increasing number of applications make the implementation of FAIR data practices critical. The data availability from publications and data repositories (i.e., PANGAEA, NOAA) for proxy reconstructions (long-term) and validation of benthic and pelagic proxies

is increasing rapidly. Moving forward, it is important that data are easily accessible to the wider scientific community using the FAIR (Findability, Accessibility, Interoperability, Reusability) guiding principles (Wilkinson et al., 2016).

Proxy data produced for paleo-oxygen reconstructions are heterogeneous in terms of material (e.g., sediment, calcite, organic matter), methodology, chronology, and data formats. It is important that we standardize oxygen proxy data sets, including qualitatively, semi-quantitatively, and quantitatively, with error margins assessed and reported. As part of the FAIR principles,

it is important that meta-data (sample identifier, core name and sections, location, depth, etc.) as well as raw data (original analyses/counts, etc.) are reported. Through networking activities scientists can promote FAIR principles and several institutes and journals have already done so. It is important that this effort is also reflected in the paleo and oceanographic communities. Following FAIR principles in oxygen proxy data management will improve data accessibility for scientists from the same discipline, as well as other disciplines and policy makers.

Below we provide recommendations for data management that follow the guidelines for modern proxy validation data and reconstructions as proposed by Khider et al. (2019), Morrill et al. (2021), Jonkers et al. (2021), Mulitza et al. (2022), Muglia et al. (2023), and Paradis et al. (2023).

1.  Organize data and save the file in a format accessible with different operating systems (e.g., linux, windows, iOS). The file needs to be in a format that is accessible and can be edited without altering the order of the data, to adopt the **Interoperability ("I")** principle. We recommend files in csv (comma separated values). Files with csv format can be



easily opened/read by data visualization and statistical software as, e.g., Excel, Rstudio, python, PaleoDataView (PDW), OceanDataView (ODV). Provide auxiliary information (metadata, depth model, age, proxy data) for each site.

2. Organize files following the **findability principle ("F")** and provide a **unique identifier** for the files (see example in supplementary information).

3. Deposit data files in a public database/ repository to follow the **accessibility principle ("A")**. The most commonly used data repositories for paleoceanographic data are PANGAEA and NCEI, and Github for code. Recently Zenodo has also emerged as an alternative repository. To make data widely accessible, add the unique repository link. Data can be submitted to repositories at any time prior to paper submission and authors can determine when data becomes available to the public. Data should be made available upon publication of manuscripts.

4. To optimize data use, follow the principle of **reusability ("R")**. There needs to be a dataset descriptor containing all the necessary details to ensure re-usability and/or replication. The original references should be cited when re-using data.

## 9 Author contributions

Jorge Cardich, Catherine Davis, Babette Hoogakker, Katrina Nillson-Kerr, and Dharma Reyes Macaya organized the workshop in September 2022 that initiated this review, supported by PAGES. Babette Hoogakker, Catherine Davis, Yi Wang, Stephanie Kusch, Dalton S. Hardisty, Allison Jacobel, Dharma Reyes Macaya, Nicolaas Glock, Sha Ni, Julio Sepúlveda, Abby Ren, Alexandra Auderset, Anya Hess, Katrin Meissner, and Jorge Cardich took the lead of writing various sections of the manuscript, with Babette Hoogakker and Catherine Davis moderating the final text. All further authors (Robert Anderson, Christine Barras, Chandranath Basak, Harold Bradbury, Inda Brinkmann, Alexis Castillo, Madelyn Cook, Kassandra Costa, Constance Choquel, Paula Diz, Jonas Donnenfield, Felix Elling, Zeynep Erdem, Helena Filipsson, Sebastian Garrido, Julia Gottschalk, Anjaly Govindankutty Menon, Jeroen Groeneveld, Christian Hallman, Ingrid Hendy, Rick Henneham, Wanyi Lu, Jean Lynch-Stieglitz, Lelia Matos, Alfredo Martínez-García, Giulia Molina, Práxedes Muñoz, Simone Moretti, Jennifer Morford, Sophie Nuber, Svetlana Radionovskaya, Morgan Raven, Christopher Somes, Anja Studer, Kazuyo Tachikawa, Raúl Tapia, Martin Tetard, Tyler Vollmer, Shuzhuang Wu, Yan Zhang, Xin-Yuan Zheng, and Yuxin Zhou) contributed to the discussions, writing of text and creation of figures.



## 10 Competing interests

The authors declare that they have no conflict of interest.

## 11 Acknowledgements

We thank ICP14 and PAGES (https://pastglobalchanges.org/) for providing logistic and travel support that facilitated the writing of this manuscript. The manuscript benefitted from discussions with Trinity Ford, Philip Froelich, Zunli Lu, Tim Sweere and Qingchen Wang.

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
