# Peer review of "Reviews and syntheses: Review of proxies for low-oxygen paleoceanographic reconstructions"

_EGUsphere, 2023_

## Referee Comment (RC1)

[referee-annotated manuscript omitted]

---

## Referee Comment (RC3)

[referee-annotated manuscript omitted]

---

## Community Comment (CC1)

[supplement omitted: unrelated document]

---

## Author Comment (AC2)

Comments on Hoogakker et al. Reviews and syntheses: Review of proxies for low-oxygen paleoceanographic reconstructions.

This is a comprehensive review that covers all aspects of proxies for oxygen deficiency as well as a broader suite of related biogeochemical processes. Given current interest in marine deoxygenation, it is certainly timely and will be widely read by the extensive research community working on oxygen proxies. The structure of the review is quite complex, with a lot of sections and subsections, and would benefit from some reorganisation. I have made specific suggestions below, as well as some editorial suggestions and comments directly on the pdf.

Response: thank you for the thorough and constructive comments, including on the pdf. We especially appreciate the suggestions for improving structure and clarity throughout and will make sure these are implemented in a revision.

Suggestions for improving structure

1) The Introduction mainly deals with the state of oxygen in modern oceans, drivers of deoxygenation and the difficulties of modelling current and future oxygen concentrations. The final paragraph of the Introduction (lines 119-143) stands out because it discusses in some detail, and at some length, the difficulties of representing oxygen in models of ocean physics. This may be important background information, but it gives the impression that it is what the review is about. The actual topic of the review is referred to only in the final sentence, where it appears almost as an afterthought ('We also need proxy-based oxygen reconstructions….').

Response: based on this and the comments of Reviewer 2, we will remove lines 119-142.

The rationale for the review, and its structure, are only outlined later in sections 4 and 5. I would suggest some editing of lines 119-143 to make this paragraph a bit shorter and less detailed and then following it with the justification for the review and its format, currently given in sections 4 and 5 (on p. 8). The section on proxies (currently section 2) would then follow on logically from this introduction.

Response: we will reorganize sections 2, 4 & 5 as suggested.

2) Section 3, Figure 1. You introduce the term 'Oxygen Deficient Zone' (with capitals) for the first time in Fig. 1, whereas oxygen minimum zones (without capitals) are first mentioned in the Introduction; subsequently, the terms are referred in the text as ODZs (54 times) and OMZs (25 times), so together they figure quite prominently in the text. Fig. 1 shows the ranges of oxygen concentrations that each of these two kinds of zone is associated with, which would imply they are different. However, it's never explained what the terms mean and what, if any, are the differences between them. You often seem to use

ODZ as a synonym of OMZ (e.g., p.46-54). Section 3 would be the obvious place to clarify whether these terms refer to different things, or whether OMZs are a particular kind of ODZ.

Response: we will make sure both terms are capitalized at first use. In answer to the question, the two terms really come from different literatures, with ODZs being used primarily in the nitrogen literature with the implication that an ODZ is an OMZ with low enough oxygen levels that denitrification may occur. Hence the frequent usage in the nitrogen isotope section. We will include a short explanation of this in Section 3.

Perhaps you could also mention in Section 3 the different units used to measure oxygen levels and maybe also the conversion factors for switching between them. In this case, the section heading could be simplified to 'Terminology and units'. You seem to use μmol kg-1 consistently, except in a few places where other units creep in. These include nM in line 1527; μmol/L on Fig. 10 axis (p. 62); μmol on axis of Fig. 13 (p. 70).

Response: we will simplify the section heading. We will add a short discussion of units, and correct the axis on Fig 13 to μmol/L and alter the units on nM to nmol/L for consistency.. Unfortunately, the conversion factors between O2 units are not straightforward as the temperature, salinity, and/or density of seawater usually needs to be accounted for. Also see comment by Ellen Thomas.

3) p. 21 and 22 - Section 6.2.4 (Organic carbon and trace element burial) and Section 6.2.5 (Other factors controlling trace element metal preservation/metal isotope fractionation). Perhaps I'm misunderstanding, but given the title of section 6.2.5, I'm unclear why these two sections are given equal status. If you rename 6.2.5 'Factors controlling trace metal preservation and metal isotope fractionation' (i.e., delete 'Other'), then you could renumber this section as 6.2.4 and place the existing Section 6.2.4 as a subsection of it (6.2.4.1), along with those current numbered 6.2.5.1-6.2.5.5.

Response: we will do this.

4) Section 6.6. This whole section seems overly complicated and quite confusing. The short introduction (6.6.1) mentions only benthic taxa, whereas section 6.6 covers planktic taxa as well. The following subsections 6.6.2 to 6.6.6.2 switch twice from benthic to planktic with two sections on methods in between. I would suggest putting everything relating to benthic forams together under one heading (split into two subsections), followed by a section on planktic forams. So the arrangement would be something like this -

6.6. Foraminiferal assemblages

Under this main heading you could add one or two sentences (not a separate section) to introduce the topic.

6.6.1. Benthic foraminifera

6.6.1.1. Relationship to bottom-water oxygen and proxies (your 'Historical perspectives….' Section).

6.6.1.2. Factors influencing proxies and their interpretation. This would include your current section 6.6.6.1, which includes two biological topics: the interplay between oxygen and the organic matter supply, and nitrate respiration.

 6.6.2. Planktic foraminifera

6.6.2.1. Relationship to water-column oxygen. Perhaps this could combine your sections 6.6.3 and 6.6.6.2 since there only seems to be a tenuous distinction between them.

I'm not sure what to suggest for your sections 6.6.4 and 6.6.5 (which I think could be combined) and your section 6.6.7 (Marine archives and limitations). Are these just about benthic species or about both benthic and planktic species? If the former, then I would put them after the benthic part.  If the latter I would put them after the planktic part.

Response: we will make the section less complicated in our revision, following the various reviewers suggestions where possible.  We propose to include a paragraph on planktics at the end of each benthic-focused section.

Section 6.6.8. is also problematic. First, the remarks about the importance of images (lines 2105-2112), which come under this main heading, could be merged with lines 1995-2001 (section 6.6.5), with which they largely overlap. Second, I don't believe that 6.6.8.1 and 6.6.8.2 are sufficiently different to merit separate subsections. If lines 2105-2112 disappear, then you could combine 6.6.8.1 and 6.6.8.2 as section 6.6.8 with the title 'Future directions and open questions.

Sections 6.6.8.1 and 6.6.8.2 will be merged, and we will also merge all non-repeated comments under 6.6.8 into section 6.6.5

Section 6.6.9 Contribution to Morphological Proxies. I'm not convinced that this belongs in the section on foraminiferal assemblages as proxies. It reads more like an introduction to the next main section on foraminiferal morphometrics (6.7). I suggest you combine it with Section 6.7.1. It also overlaps to some extent with section 6.7.2.2.

We will combine this with the existing section 6.7.2.2.

Other points

Are the first three words of the title a statement of the manuscript type?  If not, then I suggest shortening it, e,g. – 'Proxies for low-oxygen paleoceanographic reconstructions: reviews and syntheses'.

Response: yes they are a statement of the manuscript type.

9. Section 6.1.1. I'm not sure that the statement - 'The presence of laminations is a key indicator of conditions that are inconsistent with the survival of benthic fauna beyond seasonal timescales' (line 255-256) is necessarily correct. Microbioturbation that is not visible to the naked eye has been described from laminated sediments in the Santa Barbara Basin (e.g., Pike et al., 2001 Geology, 29, 923– 926). This presumably reflects the presence of meiofaunal organisms such as nematodes and forams, which can survive on very little oxygen (or in the case of forams no oxygen).

This is a good point and this statement will be removed.

10. Please specify here whether you are referring to low resolution CT scanning or high resolution micro-CT (µCT) scanning.

Response: It is not low-resolution because some CT machines can go down to 30um resolution. It's not a micro-CT per se because it cannot accommodate the whole section. We can refer to it as standard-resolution.

Line 623 etc. The Carter et al. (2020) paper doesn't seem to be included in the reference list.

Thank you for catching this. It will be added.

34, Line 1006. You define BHP in line 1002, but you don't explain anywhere what BHT means, as far as I can see.

Thank you for catching this. It will be added in line 1002 ("a bacteriohopanetetrol (BHT) isomer with unknown stereochemistry, BHT-x…").

1. 70, Fig. 13 caption says that the figure compares 4 oxygen indices, in each case based on the main species and the complete assemblages. Data for different indices are shown in different colours and the legend indicates two shades of each colour, one for main species, the other for complete assemblages. However, the figure includes only four lines of data (green, yellow, blue, pink), one for each index, with no differentiation between main species and complete assemblages.

There are actually 8 lines here but substantial overlap makes them hard to see. Thank you for bringing this to our attention. We will play with altering symbols to make the figure clearer or be explicit about the overlap in the figure caption.

[Figure]

2. 71. Section 6.6.4. Analyses and required resources. You refer to 'wet or dry sieving to separate different size fractions'. It would be useful to say a bit more about the use of different size fractions (usually 63, 125, or 150 μm, sometimes 32 or 250 μm) because these have a strong influence on the composition of foraminiferal assemblages and so are an important issue when analysing them.

Response: we could add some statement, but note that size fraction can vary considerably and can influence the results? Reviewer 2 suggests shortening this section… We can recommend to use only the size fractions that have been used for the calibration of the transfer functions, because otherwise the results will be biased.

3. 72, 6.6.6. Section heading. Here and elsewhere, I'm not sure that 'Proxy drivers' is the best expression. It doesn't sound quite right. Perhaps 'Environmental influences (or 'controls') on proxies' would be better.

Response: we will rename headings using the term "Environmental influences" rather than drivers.

4. 72, Lines 2004-2004. Some metazoans can survive at very low oxygen concentrations. For example, high density, although low diversity, assemblages of nematodes flourish at 0.05 ml.L-1 off Costa Rica (Neira et al., 2018, Frontiers in Marine Science). A polychaete species is dominant at oxygen levels of 5-6 uM on in the Pakistan margin OMZ (e.g., Jeffreys et al. 2012, Marine Ecology Progress Series).

Response: good point We will remove the clause "compared to other benthic microorganisms such as nematodes or ciliates"

Lines 2044-2045. I'm not an expert in this, but from what I understand, the storage of nitrate allows them to live in the absence of nitrate as well as oxygen. So having stored the nitrate, they can migrate to even deeper sediment levels where there is even less competition and danger of predation. When the nitrate stored in vacuoles is exhausted, they move back up into the nitrate zone and refuel.

Response: yes, we can add this detail. We will follow the suggestion by Ellen Thomas, and add a section foraminifera before the relevant proxy sections. This will be a good place to clarify that detail.

Line 2090. I'm not sure what you mean by complete foraminiferal assemblages. For modern faunas, this term would refer to the live plus dead assemblage. Obviously, that can't apply to fossil assemblages. Perhaps you could call them 'mixed assemblages'

Response: we mean to refer to assemblages that take into account all species rather than only one of a few indicator taxa. This will be clarified in the text.

Lines 2116-2118. You could mention that foraminiferal populations can fluctuate over inter-annual, as well as intra-annular time scales, even in the deep sea. Also, it would be worth adding a few words to touch on the wider issue of temporal and spatial heterogeneity and the need to analyse replicate samples in order to provide a realistic assessment of the species-level composition of modern assemblages.

Response: this can be added.

6.6.8.2. Lines 2140-2145. There's also ancient DNA, which can reveal ecosystem changes over historical and longer time scales across a wide range of taxa. I think this will become an increasingly important tool. For example -

- Barrenechea Angeles et al. (2023). Encapsulated in sediments: eDNA deciphers the

 ecosystem history of one of the most polluted European marine sites. Environment

 International,172, 107738. https://doi.org/10.1016/j.envint.2023.107738.

- Pawlowska et al. (2022). Ancient foraminiferal DNA: A new paleoceanographic proxy. https://doi.org/10.5194/egusphere-egu22-9392 EGU General Assembly 2022

Response: thank you for this suggestion. We will add a line mentioning this emerging direction as well.

Table 2, pp. 79-81 occupies a lot of space. You could reduce the size by 1) deleting the left-hand column ('Foraminifera'), and 2) inserting an extra row at the beginning of the benthic entries, merging the cells into one cell stretching across the width of the table, and putting BENTHIC in bold centred in the middle of this cell. The other column headings would remain above this merged cell. The same could be done above the planktic entries. 3) You could then make the three right-hand columns wider, so that the entries in the cells take up less vertical space.

Response: we will work with the formatting as suggested.

Line 2373-2376. 'Nevertheless…..'. Is the intention here to contrast Bolivina pacifica with B. spissa?  In this case, it would be better to start the sentence with 'On the other hand….'

Response: It is not the intention to contrast B. spissa with B. pacifica but to contrast the pore density (which does not vary with sediment depth) with the size (which varies with sediment depth). This part will be adapted for clarification.

 Minor grammatical issues

I've made editorial suggestions directly on the pdf, which I hope will improve the clarity of the text.

In places, paragraph breaks are only indicated by a carriage return. Please indicate them by leaving a blank line, as you do elsewhere in the text.

Please hyphenate 'bottom water' when used as an adjective (bottom-water oxygenation')

'Foraminifera' is a noun. Please write 'foraminiferal' when using the taxon name as an adjective - e.g. 'foraminiferal species', 'foraminiferal assemblages' (not 'foraminifera species' etc). You could also use the phrase 'assemblages of foraminifera'.

Response: thank you for additional editorial comments. We will correct each of these category of grammatical error and take into account the suggestions made in the PDF as well.

---

## Author Comment (AC3)

Summary:

Hoogakker and colleagues provide a very extensive review of various proxy approaches (ranging from trace elements & their isotopes, over biomarkers, nitrogen isotopes, and foraminifera-based proxies) that can be used to reconstruct marine oxygen changes over the Cenozoic. The amount of information included in the paper is immense and has almost textbook dimensions (100 pages of text, including 16 figures, and >53 pages of references). In my opinion, this can be seen as a strength and a weakness of the paper – and I suppose it is an editorial decision if *Biogeosciences* wants to publish such an extended review study or if it would be better to split the review paper into multiple review studies to make it more manageable for readers and also reviewers (a lot of different subsections have a separate introduction already).

Given the paper's extensive nature, I will concentrate my (more detailed) comments on the initial sections, up to and including Section 6.2.6.3. I will provide more general comments, particularly on what I perceive as the manuscript's primary limitation - its structure.

Overall, I am convinced that the information provided by the manuscript will be of great value to the community, but, in my opinion, the text should be shortened significantly. I initially thought the manuscript would be a fantastic way to learn about different proxy approaches used to quantify paleo-oxygenation changes, but I got discouraged by the very long text and vast amount of subsections of the manuscript. This might be fine for general readers who can pick and choose the sections they are interested in (in contrast to a job of a reviewer). However, I still think that a more focused text and a better organization of subsections would improve the usefulness and approachability of the manuscript.

> Response: we thank the reviewer for their helpful comments and suggestions. The further suggestions here of which sections could be shortened is an especially helpful perspective. And we definitely appreciate the effort taken to review such a long and broad manuscript.

General comments:

Comment #1.1: Shortening the manuscript and combining duplicate information
Considering the long list of authors and diverse topics covered, I suppose that different groups of authors were responsible for different sections – which is absolutely fine and necessary – however, the manuscript would benefit from a few core authors reviewing the entire manuscript and combining/deleting overlapping information (as also suggested in the review by Ellen Thomas; with nitrogen-dynamics being discussed in multiple sections only being one example).

Response: we agree with the reviewer that a further review of the manuscript will help eliminate repetition and improve flow. We will do so as suggested by this reviewer and Ellen Thomas.

Comment #1.2:

At times, the manuscript is quite wordy and/or provides a lot of detail on topics that are not directly related to the understanding of the specific redox proxies. A few example parts (mainly of the first half of the manuscript that I looked at in more detail) that could (in my opinion) be shortened are: Sections 3; 6.1.1. (especially in 2nd and 3rd paragraph); 6.1.2.

  Response: both of these sections will be shortened.

Is the information of the "Materials/Methods" type sections really important for the review paper (e.g., 6.2.2, 6.3.7, 6.5.3, 6.6.4, 6.7.3)? I found these sections rather technical and not very informative/crucial for understanding the specific proxy (but that might, of course, be personal preference).
  Response: we feel that including some perspective on the resources needed for each of these approaches is important, however we will shorten and streamline each of these sections.

Some of the future directions sections are rather long (especially 6.2.6 & 6.6.8 & 6.7.8).
  Response: these sections will also be shortened.

The introduction to "6.5 Foraminifera trace elements" (6.5.1+6.5.2) consists of more than 7 pages (just text) plus 4 Figures. It should be possible to shorten this text (or combine figures) without losing too much relevant content.
  Response: we will condense this section.

Comment #2: Structure
The manuscript includes too many subsections (sometimes up to 5 levels—see, e.g., Subsection in 6.2.3), which is confusing and makes it challenging to pinpoint where the current information 'lives' in relation to the overall structure of the manuscript. I think a depth of 3 or 4 subsections should be enough; otherwise, the reader loses orientation.
  Response: we will restructure section 6 such that each proxy subsection is on its own, as well as eliminating the 5th level subsections as suggested also by other reviewers.

Section 6 consists of many, many subsections and forms the majority of the text. In contrast, the previous Sections 1 – 5 are very short and do not have any subsections. This should be better balanced. For instance, why are Sections 4 and 5 separate Sections at all – this information could be part of the general introduction. Section 6 could maybe be organized into multiple Sections of similar size (potentially just the current subsections of Sec. 6).
  Response: Sections 4 and 5 will be combined and moved upwards as suggested by Reviewer 3.

Comment #3: Introduction

The introduction does not introduce the topic of the review article. The second paragraph explains the causes for ocean deoxygenation and the second half of the introduction exclusively deals with problems in Earth system models to simulate ocean oxygen correctly. This is very surprising as Earth system models are not part of the review paper at all.

More relevant would be a general introduction to oxygenation changes over the Cenozoic and redox-proxies, and how they can help quantify the oxygenation changes. Information given in the different intros throughout the document could here be combined(e.g. such as the information given in 6.1, 6.2.1 and similar sections throughout the manuscript).

Response: we will remove the paragraph on modeling as we agree it is incongruent with the review in its current form and will consolidate repetitive introductory information throughout and move this earlier in the document. This reorganization should make clearer the topic and aims of the review.

Comment #4:

6.2. "Sedimentary redox trace elements and isotopes"
An overview table would be very useful that summarizes/compares the key characteristics, residence times and applications of the different proxies.

Response: this is a great idea - we will include such a table in this section.

Also, for Subsections 6.2.3: Why not combine the elements & their isotopes in one section?

Response: element and isotope sections can be consolidated as suggested.

Comment #5:

The large amount of references for some sentences (sometimes 6-12) makes it difficult to read the text (just a few examples: lines 261, 521, 561, 575, Sections 6.2.3.1.4, 6.2.5.2 in general, 628, … ). It would be helpful to shorten the references given, e.g., only provide the most important references are given or a few examples. Also, it is not necessary to cite the same paper multiple times in consecutive sentences (for instance, see 6.2.3.1.2; 6.2.3.1.4; 6.2.3.3.1)

Response: we will attend to the referencing to avoid repetition as much as possible.

More Technical Comments:

First two sentences of Section 2: Please rephrase. It sounds like seawater temperature, pH, and dissolved oxygen are environmental properties that can generally not be measured directly.

Response: we make sure to rephrase this so it is understood we mean proxies for these direct measures.

Fig 1: the caption says: "Proxy types shown in olive can be used to reconstruct oxygen from benthic settings, those in green can be used for pelagic settings." But I do not see olive and green proxies in the figure.
Response: thank you for catching this. This should read as "grey" and "blue" respectively and will be corrected.

Ln 389: "Fully digested" what does this mean – not clear for a non-data person.
Response: this will be rewritten as "fully or partially dissolved"

ln. 547: decomposition of organic matter is probably meant here. Organic carbon describes only the C itself contained in organic material. Please check the use of organic carbon throughout the document.
Response: you are correct. We will use "organic matter" here and check throughout the manuscript.

Some of the subsection titles are rather long and should be shortened, see e.g. Subsections 6.3.3; 6.3.6
Response: these will be shortened to "Biomarkers of microbial processes associated with oxygen deficiency" and "Non-specific/orphan biomarkers from oxygen-deficient depositional settings" respectively

Title 6.6.3 = 6.6.2 -- I suppose, 6.6.3 is Planktic foraminifera
Response: you are correct. Thank you.

Line 398: "This is especially true ..." please rephrase, it is unclear what 2 to 3 cm kyr-1 referes to.
Response: this will be rephrased as "This is especially true in environments with low accumulation rates less than 2 (Jung et al., 1997; Mangini et al., 2001) or 3 cm kyr-1 (Jacobel et al., 2020)."

Fig. 4: It could be made more obvious what boundary condition is changed between a and b.
Response: we will do this.

Line 709: … the occurrences of (singular)

Response: this will be corrected.

Figure 8: Please include the figure in higher resolution. In particular, the text looks pixelated.
Response: we will make sure this is submitted in higher resolution.

Citation: https://doi.org/10.5194/egusphere-2023-2981-RC2

---

## Author Comment (AC4)

Response to Thomas Algao

Page 5: seafloor is one word; superscript 15.

Response: thanks for spotting that we can make those changes.

Page 9: "lamination" is a process; "laminae" are the features themselves

Response: we can change this in the text to "laminae".

Page 15: awkward acronyms

Response: we will adapt this, and use TE etc.

Page 17: use subscripts for "HR" and "T"

Response: we will adapt this.

Page 24: "post-depositionally" or "in post-depositional environments"

Response:

Response: Thanks, we will use post-depositionally.

Page 26: there is no "Algeo and Lyons 2009" paper, this may be "Algeo and Tribovillard, 2009"

Response: we will make sure that we reference the right paper.

Page 27 I will call your attention to this paper:

Wang, X., Algeo, T.J., Li, C. and Zhu, M., 2023. Spatial pattern of marine oxygenation set by tectonic and ecological drivers over the Phanerozoic. Nature Geoscience, 16(11), pp.1020-1026.

Response: we can add this reference.

Page 29: "kyr" is plural no "s" needed.

Response: this can be changed in the text.

Page 48: "FB" should a subscript following "d15N"

Response: we disagree, as "FB-d15N" is the conventional way to abbreviate "foraminifera-bound nitrogen isotopes" in the community. We acknowledge that this is different to the format used for bulk sedimentary nitrogen isotopes (d15Nbulk), but to keep the proxy name similar to what is published in the fossil-bound nitrogen isotope papers we would like to keep the naming as it is.

Page 49: "CS" should a subscript following "d15N"

Response: same comment as before, this is the conventional format used in the fossil-bound d15N community.

Page 51: I am puzzled by the shown relationship of higher d15N to lower O2. In my experience, lower d15N signals lower O2, especially where ammonium plays a role.

Response: We will explain in the following three aspects. (1) As we have explained in section 6.4.1, denitrification is the predominant process determining d15N changes in an ODZ. Denitrification strongly discriminates against the heavier isotope 15N, progressively enriching the remaining nitrate pool in 15N as nitrate consumption proceeds. However, we also state that there is not a quantitative relationship between d15N and O2. It is rather that exceptionally high d15N (nitrate d15N is typically above 15 permil in the ODZ) is a result of denitrification which occurs when O2 concentration is below a threshold. (2) In the ocean since the oxygenation of the Earth's surface, nitrate should be the predominant inorganic species available in the ocean, which also serves as the main form of nitrogen nutrient for organisms. The nitrate is assimilated by organisms, thus its isotopic signal is recorded in the biomass, and if the fossils are preserved, their d15N provides information mostly on the d15N changes of nitrate pool in the ocean. As a result, the exceptionally high fossil d15N (in foram, diatom, coral and others) would be best explained by high nitrate d15N resulting from denitrification in low oxygen environment. This may be different from an ammonium ocean prior to the oxygenation of the ocean. Since our paper focuses on the ODZ which is a phenomenon only relevant in the oxygenated ocean, we will not consider the events/periods when the whole ocean is anoxic. (3) Ammonium concentration is almost always lower than 1uM in the open ocean today, including the ODZ. In the ODZ, the ammonium is produced by degradation of organic N, and consumed by anammox and nitrification. Measuring ammonium d15N is challenging because of its low concentration, but the both anammox and nitrification discriminate against the heavier isotopes, so should also cause ammonium d15N to increase in the ODZ. However, as most of ammonium is completely removed within the ODZ, these processes have no influence to the surrounding ocean. In some special environment such as observed in the Bering Sea shelf, high organic flux is remineralized in the sediment to produce ammonium. The ammonium is not completely nitrified so some of the ammonium enriched with 15N is released to the water column and subsequently is taken up by phytoplankton. It is seen in such environment that ammonium produced in low oxygen environment can also generate high d15N signal (see Granger et al., 2011 JGR ocean. https://doi.org/10.1029/2010JC006751). So in summary, the processes involving ammonium do not have a strong effect on d15N changes in the ODZ as most of the ammonium produced in the ODZ is completely removed, and even if there is remaining ammonium, its d15N should also be high.

Page 101: I noted citations to 3 or 4 Algeo papers in the text that are not included in the reference list, so the citations and references have not been cross-checked for completeness.

Response: in our proposed revised manuscript we will check all this.

---

## Author Response (AR1)

Response to Thomas Algeo

We are very grateful for your review and suggestions for improving our manuscript. Below is a detailed response of changes implemented (in green). In some cases, we kindly disagree with the suggested changes and instead opt to adhere to community standards. In those cases, we d/provided an explanation.

Page 5: seafloor is one word; superscript 15.

Thanks for spotting this, we have implemented this change.

Page 9: "lamination" is a process; "laminae" are the features themselves

Throughout the text this word represents the feature itself, and we have changed lamination with laminae throughout.

Page 15: awkward acronyms

Below we show the adapted acronyms:

"Quantification of redox-sensitive metal enrichment ($Metal_{EF}$) may be determined following Tribovillard et al. (2006) (Eq. 6.2.1) and Böning et al. (2009) (Eq. 6.2.2):

Equation 6.2.1  $Metal_{EF} = (Metal/NE)_{sample} / (Metal/NE)_{background}$

where NE corresponds to the element for normalization.

Equation 6.2.2   Metal excess (normalized by Al) = $Metal_{sample} - (Metal/Al)_{background} * Al_{sample}$ "

Page 17: use subscripts for "HR" and "T"

We have made those changes, and for consistency also subscripted "py".

Page 24: "post-depositionally" or "in post-depositional environments"

Response:

Thanks, we have changed this to "post-depositionally" in the text.

Page 26: there is no "Algeo and Lyons 2009" paper, this may be "Algeo and Tribovillard, 2009"

We have changed this to Algeo and Tribovillard, 2009.

Page 27 I will call your attention to this paper:

Wang, X., Algeo, T.J., Li, C. and Zhu, M., 2023. Spatial pattern of marine oxygenation set by tectonic and ecological drivers over the Phanerozoic. Nature Geoscience, 16(11), pp.1020-1026.

We added this reference.

Page 29: "kyr" is plural no "s" needed.

We have changed this in the text.

Page 48: "FB" should a subscript following "d15N"

Response: we disagree, as "FB-d15N" is the conventional way to abbreviate "foraminifera-bound nitrogen isotopes" in the community. We acknowledge that this is different to the format used for bulk sedimentary nitrogen isotopes (d15Nbulk), but to keep the proxy nomenclature similar to what is published in the fossil-bound nitrogen isotope papers we would like to keep the naming as it is.

Page 49: "CS" should a subscript following "d15N"

Response: same comment as before, this is the conventional format used in the fossil-bound d15N community.

Page 51: I am puzzled by the shown relationship of higher d15N to lower O2. In my experience, lower d15N signals lower O2, especially where ammonium plays a role.

Response: The following three aspects need to be considered when interpreting d15N data. (1) As we have discussed in section 6.4.1, denitrification is the predominant process determining d15N changes in an ODZ. Denitrification strongly discriminates against the heavier isotope 15N, progressively enriching the remaining nitrate pool in 15N as nitrate consumption proceeds. However, we also state that there is not a quantitative relationship between d15N and O2. It is rather that exceptionally high d15N (nitrate d15N is typically above 15 permil in an ODZ) is a result of denitrification which occurs when O2 concentration is below a threshold. (2) Since the oxygenation of the Earth's surface, nitrate should be the predominant inorganic species available in the ocean, which also serves as the main form of nitrogen nutrient for organisms. The nitrate is assimilated by organisms, thus its isotopic signal is recorded in the biomass, and if the fossils are preserved, their d15N provides information mostly on the d15N changes of nitrate pool in the ocean. As a result, the exceptionally high fossil d15N (in foram, diatom, coral and others) would be best explained by high nitrate d15N resulting from denitrification in low oxygen environments. This may be different from an ammonium ocean prior to the oxygenation of the ocean. Since our paper focuses on the ODZ which is a phenomenon only relevant in the oxygenated ocean, we do not consider the events/periods when the whole ocean was anoxic.

(3) Ammonium concentration is almost always lower than 1uM in the open ocean today, including the ODZ. In the ODZ, the ammonium is produced by degradation of organic N, and consumed by anammox and nitrification. Measuring ammonium d15N is challenging because of its low concentration, but both anammox and nitrification discriminate against the heavier isotopes, so both processes should also cause ammonium d15N to increase in the ODZ. However, as most of ammonium is completely removed within the ODZ, these processes have no influence on the surrounding ocean. In some special environment such as observed in the Bering Sea shelf, high organic carbon fluxes cause high remineralization rates in the sediment to produce ammonium. The ammonium is not completely nitrified and the remaining ammonium pool enriched in 15N is released to the water column and subsequently is taken up by phytoplankton. In such low oxygen environments ammonium can also generate high d15N signals (see Granger et al., 2011 JGR ocean. https://doi.org/10.1029/2010JC006751). However, the processes involving ammonium do not have a strong effect on d15N changes in the ODZ as most of the ammonium produced in the ODZ is completely removed, and even if there is remaining ammonium, its d15N should also be high.

Page 101: I noted citations to 3 or 4 Algeo papers in the text that are not included in the reference list, so the citations and references have not been cross-checked for completeness.

Many thanks for spotting this. We have added those references, and will carry out a thorough cross-check before resubmission.

Response reviewer 2

We are very grateful for your review and suggestions for improving our manuscript. Below is a detailed response of changes implemented (in green). In some cases, we kindly disagree with the suggested changes and instead opt to adhere to community standards. In those cases, we provide an explanation.

Summary:
Hoogakker and colleagues provide a very extensive review of various proxy approaches (ranging from trace elements & their isotopes, over biomarkers, nitrogen isotopes, and foraminifera-based proxies) that can be used to reconstruct marine oxygen changes over the Cenozoic. The amount of information included in the paper is immense and has almost textbook dimensions (100 pages of text, including 16 figures, and >53 pages of references). In my opinion, this can be seen as a strength and a weakness of the paper – and I suppose it is an editorial decision if *Biogeosciences* wants to publish such an extended review study or if it would be better to split the review paper into multiple review studies to make it more manageable for readers and also reviewers (a lot of different subsections have a separate introduction already).
Given the paper's extensive nature, I will concentrate my (more detailed) comments on the initial sections, up to and including Section 6.2.6.3. I will provide more general comments, particularly on what I perceive as the manuscript's primary limitation - its structure.
Overall, I am convinced that the information provided by the manuscript will be of great value to the community, but, in my opinion, the text should be shortened significantly. I initially thought the manuscript would be a fantastic way to learn about different proxy approaches used to quantify paleo-oxygenation changes, but I got discouraged by the very long text and vast amount of subsections of the manuscript. This might be fine for general readers who can pick and choose the sections they are interested in (in contrast to a job of a reviewer). However, I still think that a more focused text and a better organization of subsections would improve the usefulness and approachability of the manuscript.

Response: we thank the reviewer for their helpful comments and suggestions. The further suggestions below of which sections could be shortened is an especially helpful perspective. And we definitely appreciate the effort taken to review such a long and broad manuscript.

General comments:
Comment #1.1: Shortening the manuscript and combining duplicate information
Considering the long list of authors and diverse topics covered, I suppose that different groups of authors were responsible for different sections – which is absolutely fine and necessary – however, the manuscript would benefit from a few core authors reviewing

the entire manuscript and combining/deleting overlapping information (as also suggested in the review by Ellen Thomas; with nitrogen-dynamics being discussed in multiple sections only being one example).

Response: we agree with the reviewer and have made changes throughout the manuscript to eliminate repetition and improve flow. These changes can be found in the fully annotated manuscript.

Comment #1.2:

At times, the manuscript is quite wordy and/or provides a lot of detail on topics that are not directly related to the understanding of the specific redox proxies. A few example parts (mainly of the first half of the manuscript that I looked at in more detail) that could (in my opinion) be shortened are: Sections 3; 6.1.1. (especially in 2nd and 3rd paragraph); 6.1.2.

Response:  sections 6.1.1 and 6.1.2 have been shortened. Following other reviewers' comments we made some changes to section 3.

Is the information of the "Materials/Methods" type sections really important for the review paper (e.g., 6.2.2, 6.3.7, 6.5.3, 6.6.4, 6.7.3)? I found these sections rather technical and not very informative/crucial for understanding the specific proxy (but that might, of course, be personal preference).

Response: we feel that including some perspective on the resources needed for each of these approaches is important, however we have shortened and streamlined these sections.

Some of the future directions sections are rather long (especially 6.2.6 & 6.6.8 & 6.7.8).

Response: these sections have been shortened.

The introduction to "6.5 Foraminifera trace elements" (6.5.1+6.5.2) consists of more than 7 pages (just text) plus 4 Figures. It should be possible to shorten this text (or combine figures) without losing too much relevant content.

Response: we have condensed this section and merged figures 9-11 into one figure.

Comment #2: Structure

The manuscript includes too many subsections (sometimes up to 5 levels—see, e.g., Subsection in 6.2.3), which is confusing and makes it challenging to pinpoint where the current information 'lives' in relation to the overall structure of the manuscript. I think a depth of 3 or 4 subsections should be enough; otherwise, the reader loses orientation.

Response: we have restructured section 6, and eliminating the 5th level subsections as suggested also by other reviewers.

Section 6 consists of many, many subsections and forms the majority of the text. In contrast, the previous Sections 1 – 5 are very short and do not have any subsections. This should be better balanced. For instance, why are Sections 4 and 5 separate Sections at all – this information could be part of the general introduction. Section 6 could maybe be organized into multiple Sections of similar size (potentially just the current subsections of Sec. 6).

Response: Sections 4 and 5 have been combined and moved upwards as suggested by Reviewer 3. Part of section 5 now features at the end of the proxy section.

Comment #3: Introduction
The introduction does not introduce the topic of the review article. The second paragraph explains the causes for ocean deoxygenation and the second half of the introduction exclusively deals with problems in Earth system models to simulate ocean oxygen correctly. This is very surprising as Earth system models are not part of the review paper at all.

More relevant would be a general introduction to oxygenation changes over the Cenozoic and redox-proxies, and how they can help quantify the oxygenation changes. Information given in the different intros throughout the document could here be combined(e.g. such as the information given in 6.1, 6.2.1 and similar sections throughout the manuscript).

Response: we have moved the paragraph on modeling to the supplementary information, as we agree it was incongruent with the review format. We also consolidated repetitive introductory information throughout and move this earlier the document. We also moved our aim to the introduction. This reorganization should make clearer the topic and aims of the review.

Comment #4:
6.2. "Sedimentary redox trace elements and isotopes"
An overview table would be very useful that summarizes/compares the key characteristics, residence times and applications of the different proxies.
Response: this is a great idea - we have included such a table in this section.

Also, for Subsections 6.2.3: Why not combine the elements & their isotopes in one section?
Response: element and isotope sections have been consolidated as suggested.

Comment #5:
The large amount of references for some sentences (sometimes 6-12) makes it difficult to read the text (just a few examples: lines 261, 521, 561, 575, Sections 6.2.3.1.4, 6.2.5.2 in general, 628, … ). It would be helpful to shorten the references given, e.g., only

provide the most important references are given or a few examples. Also, it is not necessary to cite the same paper multiple times in consecutive sentences (for instance, see 6.2.3.1.2; 6.2.3.1.4; 6.2.3.3.1)

Response: we have attended to the referencing to avoid repetition.

More Technical Comments:

First two sentences of Section 2: Please rephrase. It sounds like seawater temperature, pH, and dissolved oxygen are environmental properties that can generally not be measured directly.

Response: we have changed this to "Proxies provide indirect representations of environmental variables in circumstances where they cannot be measured directly, such as the geological past."

Fig 1: the caption says: "Proxy types shown in olive can be used to reconstruct oxygen from benthic settings, those in green can be used for pelagic settings." But I do not see olive and green proxies in the figure.

Response: thank you for catching this. We have corrected the reference the colors in the caption to "grey" and "blue" respectively.

Ln 389: "Fully digested" what does this mean – not clear for a non-data person.

Response: this has been rewritten as "fully or partially dissolved" in line 398.

ln. 547: decomposition of organic matter is probably meant here. Organic carbon describes only the C itself contained in organic material. Please check the use of organic carbon throughout the document.

Response: you are correct. We now use "organic matter" here and have checked throughout the manuscript.

Some of the subsection titles are rather long and should be shortened, see e.g. Subsections 6.3.3; 6.3.6

Response: these have been shortened to "Biomarkers of microbial processes associated with oxygen deficiency" and "Non-specific/orphan biomarkers from oxygen-deficient depositional settings" respectively

Title 6.6.3 = 6.6.2 -- I suppose, 6.6.3 is Planktic foraminifera

Response: this has been corrected. Thank you.

Line 398: "This is especially true ..." please rephrase, it is unclear what 2 to 3 cm kyr-1 referes to.

Response: this has been rephrased as "This is especially true in environments with low accumulation rates of less than 2 (Jung et al., 1997; Mangini et al., 2001) or 3 cm kyr-1 (Jacobel et al., 2020)"

Fig. 4: It could be made more obvious what boundary condition is changed between a and b.

Response: we feel this is already explained in the caption (difference is in oxygen penetration depth.

Line 709: … the occurrences of (singular)

Response: this has been corrected.

Figure 8: Please include the figure in higher resolution. In particular, the text looks pixelated.

Response: we are including the higher resolution figure.

Citation: https://doi.org/10.5194/egusphere-2023-2981-RC2

Response reviewer 3

We are very grateful for your review and suggestions for improving our manuscript. Below is a detailed response of changes implemented (in green). In some cases, we kindly disagree with the suggested changes. In those cases, we provide an explanation.

Comments on Hoogakker et al. Reviews and syntheses: Review of proxies for low-oxygen paleoceanographic reconstructions.

This is a comprehensive review that covers all aspects of proxies for oxygen deficiency as well as a broader suite of related biogeochemical processes. Given current interest in marine deoxygenation, it is certainly timely and will be widely read by the extensive research community working on oxygen proxies. The structure of the review is quite complex, with a lot of sections and subsections, and would benefit from some reorganisation. I have made specific suggestions below, as well as some editorial suggestions and comments directly on the pdf.

Response: thank you for the thorough and constructive comments, including on the pdf. We especially appreciate the suggestions for improving structure and clarity throughout and implemented these in our revision.

Suggestions for improving structure

1) The Introduction mainly deals with the state of oxygen in modern oceans, drivers of deoxygenation and the difficulties of modelling current and future oxygen concentrations. The final paragraph of the Introduction (lines 119-143) stands out because it discusses in some detail, and at some length, the difficulties of representing oxygen in models of ocean physics. This may be important background information, but it gives the impression that it is what the review is about. The actual topic of the review is referred to only in the final sentence, where it appears almost as an afterthought ('We also need proxy-based oxygen reconstructions….').

Response: based on this and the comments of Reviewer 2, we have moved lines 119-142 to the supplementary information.

The rationale for the review, and its structure, are only outlined later in sections 4 and 5. I would suggest some editing of lines 119-143 to make this paragraph a bit shorter and less detailed and then following it with the justification for the review and its format, currently given in sections 4 and 5 (on p. 8). The section on proxies (currently section 2) would then follow on logically from this introduction.

Response: we have reorganized sections 2, 4 & 5 as suggested.

2) Section 3, Figure 1. You introduce the term 'Oxygen Deficient Zone' (with capitals) for the first time in Fig. 1, whereas oxygen minimum zones (without capitals) are first mentioned in the Introduction; subsequently, the terms are referred in the text as ODZs (54 times) and OMZs (25 times), so together they figure quite prominently in the text.

Fig. 1 shows the ranges of oxygen concentrations that each of these two kinds of zone is associated with, which would imply they are different. However, it's never explained what the terms mean and what, if any, are the differences between them. You often seem to use ODZ as a synonym of OMZ (e.g., p.46-54). Section 3 would be the obvious place to clarify whether these terms refer to different things, or whether OMZs are a particular kind of ODZ.

Response: we have made sure both terms are capitalized at first use. In answer to the question, the two terms really come from different literatures, with ODZs being used primarily in the nitrogen literature with the implication that an ODZ is an OMZ with low enough oxygen levels that denitrification may occur. Hence the frequent usage in the nitrogen isotope section. We have included a short explanation of this in Section 3.

Perhaps you could also mention in Section 3 the different units used to measure oxygen levels and maybe also the conversion factors for switching between them. In this case, the section heading could be simplified to 'Terminology and units'. You seem to use µmol kg$_{-1}$ consistently, except in a few places where other units creep in. These include nM in line 1527; µmol/L on Fig. 10 axis (p. 62); µmol on axis of Fig. 13 (p. 70).

Response: we have simplified the section heading. We added a short discussion of units, and correct the axis on Fig 13 to display µmol/L and altered the units of nM to nmol/L for consistency. Unfortunately, the conversion factors between O2 units are not straightforward as the temperature, salinity, and/or density of seawater usually needs to be accounted for.  Also see comment by Ellen Thomas.

3)  p. 21 and 22 - Section 6.2.4 (Organic carbon and trace element burial) and Section 6.2.5 (Other factors controlling trace element metal preservation/metal isotope fractionation). Perhaps I'm misunderstanding, but given the title of section 6.2.5, I'm unclear why these two sections are given equal status. If you rename 6.2.5 'Factors controlling trace metal preservation and metal isotope fractionation' (i.e., delete 'Other'), then you could renumber this section as 6.2.4 and place the existing Section 6.2.4 as a subsection of it (6.2.4.1), along with those current numbered 6.2.5.1-6.2.5.5.

Response: we have made this change.

4)  Section 6.6. This whole section seems overly complicated and quite confusing. The short introduction (6.6.1) mentions only benthic taxa, whereas section 6.6 covers planktic taxa as well. The following subsections 6.6.2 to 6.6.6.2 switch twice from benthic to planktic with two sections on methods in between. I would suggest putting everything relating to benthic forams together under one heading (split into two subsections), followed by a section on planktic forams. So the arrangement would be something like this -

6.6. Foraminiferal assemblages

Under this main heading you could add one or two sentences (not a separate section) to introduce the topic.

6.6.1. Benthic foraminifera

6.6.1.1. Relationship to bottom-water oxygen and proxies (your 'Historical perspectives….' Section).

6.6.1.2. Factors influencing proxies and their interpretation. This would include your current section 6.6.6.1, which includes two biological topics: the interplay between oxygen and the organic matter supply, and nitrate respiration.

6.6.2. Planktic foraminifera

6.6.2.1. Relationship to water-column oxygen. Perhaps this could combine your sections 6.6.3 and 6.6.6.2 since there only seems to be a tenuous distinction between them.

I'm not sure what to suggest for your sections 6.6.4 and 6.6.5 (which I think could be combined) and your section 6.6.7 (Marine archives and limitations). Are these just about benthic species or about both benthic and planktic species? If the former, then I would put them after the benthic part. If the latter I would put them after the planktic part.

Response: we have made this section less complicated in our revision, following the various reviewers' suggestions where possible. We have included a paragraph on planktics at the end of each benthic-focused section.

Section 6.6.8. is also problematic. First, the remarks about the importance of images (lines 2105-2112), which come under this main heading, could be merged with lines 1995-2001 (section 6.6.5), with which they largely overlap. Second, I don't believe that 6.6.8.1 and 6.6.8.2 are sufficiently different to merit separate subsections. If lines 2105-2112 disappear, then you could combine 6.6.8.1 and 6.6.8.2 as section 6.6.8 with the title 'Future directions and open questions.

Sections 6.6.8.1 and 6.6.8.2 have been merged, and we have merged all non-repeated comments under 6.6.8 into section 6.6.5 (now section 9).

Section 6.6.9 Contribution to Morphological Proxies. I'm not convinced that this belongs in the section on foraminiferal assemblages as proxies. It reads more like an introduction to the next main section on foraminiferal morphometrics (6.7). I suggest you combine it with Section 6.7.1. It also overlaps to some extent with section 6.7.2.2.

We have combined this with section 6.7.1 (now section 10.1).

Other points

Are the first three words of the title a statement of the manuscript type? If not, then I suggest shortening it, e,g. – 'Proxies for low-oxygen paleoceanographic reconstructions: reviews and syntheses'.

Response: yes, they are a statement of the manuscript type.

9. Section 6.1.1. I'm not sure that the statement - 'The presence of laminations is a key indicator of conditions that are inconsistent with the survival of benthic fauna beyond seasonal timescales' (line 255-256) is necessarily correct. Microbioturbation that is not visible to the naked eye has been described from laminated sediments in the Santa Barbara Basin (e.g., Pike et al., 2001 Geology, 29, 923– 926). This presumably reflects the presence of meiofaunal organisms such as nematodes and forams, which can survive on very little oxygen (or in the case of forams no oxygen).

This is a good point and this statement has been reworded.

10. Please specify here whether you are referring to low resolution CT scanning or high resolution micro-CT (µCT) scanning.

Response: It is not low-resolution because some CT machines can go down to 30um resolution. It's not a micro-CT per se because it cannot accommodate the whole section. We are now referring to it as standard-resolution.

Line 623 etc. The Carter et al. (2020) paper doesn't seem to be included in the reference list.

Thank you for catching this. This has been added.

34, Line 1006. You define BHP in line 1002, but you don't explain anywhere what BHT means, as far as I can see.

Thank you for catching this. We changed this line to ("a bacteriohopanetetrol (BHT) isomer with unknown stereochemistry, BHT-x…").

1. 70, Fig. 13 caption says that the figure compares 4 oxygen indices, in each case based on the main species and the complete assemblages. Data for different indices are shown in different colours and the legend indicates two shades of each colour, one for main species, the other for complete assemblages. However, the figure includes only four lines of data (green, yellow, blue, pink), one for each index, with no differentiation between main species and complete assemblages.

There are actually 8 lines here but substantial overlap makes them hard to see. Thank you for bringing this to our attention. We have changed the main species symbols to squares and are now explicit about the overlap in the figure caption.

[Figure]

2. 71. Section 6.6.4. Analyses and required resources. You refer to 'wet or dry sieving to separate different size fractions'. It would be useful to say a bit more about the use of different size fractions (usually 63, 125, or 150 μm, sometimes 32 or 250 μm) because these have a strong influence on the composition of foraminiferal assemblages and so are an important issue when analysing them.

   Response: we have added a statement, but note that size fraction can vary considerably and can influence the results.

3. 72, 6.6.6. Section heading. Here and elsewhere, I'm not sure that 'Proxy drivers' is the best expression. It doesn't sound quite right. Perhaps 'Environmental influences (or 'controls') on proxies' would be better.

   Response: we have renamed the heading "Environmental influences".

4. 72, Lines 2004-2004. Some metazoans can survive at very low oxygen concentrations. For example, high density, although low diversity, assemblages of nematodes flourish at 0.05 ml.L-1 off Costa Rica (Neira et al., 2018, Frontiers in Marine Science). A polychaete species is dominant at oxygen levels of 5-6 uM on in the Pakistan margin OMZ (e.g., Jeffreys et al. 2012, Marine Ecology Progress Series).

   Response: good point. We have removed the clause "compared to other benthic microorganisms such as nematodes or ciliates"

Lines 2044-2045. I'm not an expert in this, but from what I understand, the storage of nitrate allows them to live in the absence of nitrate as well as oxygen. So having stored

the nitrate, they can migrate to even deeper sediment levels where there is even less competition and danger of predation. When the nitrate stored in vacuoles is exhausted, they move back up into the nitrate zone and refuel.

Response: we have added the sentence: 'Some species are even able to survive, calcify and reproduce under anoxia (Langlet et al., 2013 & Nardelli et al., 2014) possessing diverse adaptations and survival strategies to oxygen depleted conditions (reviewed in Glock 2013).'

Line 2090. I'm not sure what you mean by complete foraminiferal assemblages. For modern faunas, this term would refer to the live plus dead assemblage. Obviously, that can't apply to fossil assemblages. Perhaps you could call them 'mixed assemblages'

Response: we mean to refer to assemblages that take into account all species rather than only one of a few indicator taxa. This has been clarified in the text.

Lines 2116-2118. You could mention that foraminiferal populations can fluctuate over inter-annual, as well as intra-annular time scales, even in the deep sea. Also, it would be worth adding a few words to touch on the wider issue of temporal and spatial heterogeneity and the need to analyse replicate samples in order to provide a realistic assessment of the species-level composition of modern assemblages.

Response: this has been added.

6.6.8.2. Lines 2140-2145. There's also ancient DNA, which can reveal ecosystem changes over historical and longer time scales across a wide range of taxa. I think this will become an increasingly important tool. For example -

- Barrenechea Angeles et al. (2023). Encapsulated in sediments: eDNA deciphers the

 ecosystem history of one of the most polluted European marine sites. Environment

 International,172, 107738. https://doi.org/10.1016/j.envint.2023.107738.

- Pawlowska et al. (2022). Ancient foraminiferal DNA: A new paleoceanographic proxy. https://doi.org/10.5194/egusphere-egu22-9392 EGU General Assembly 2022

Response: thank you for this suggestion. We have added a line mentioning this emerging direction as well.

Table 2, pp. 79-81 occupies a lot of space. You could reduce the size by 1) deleting the left-hand column ('Foraminifera'), and 2) inserting an extra row at the beginning of the benthic entries, merging the cells into one cell stretching across the width of the table, and putting BENTHIC in bold centred in the middle of this cell. The other column headings would remain above this merged cell. The same could be done above the

planktic entries. 3) You could then make the three right-hand columns wider, so that the entries in the cells take up less vertical space.

Response: we have reformatted the table as suggested.

Line 2373-2376. 'Nevertheless…..'. Is the intention here to contrast Bolivina pacifica with B. spissa?  In this case, it would be better to start the sentence with 'On the other hand….'

Response: It is not the intention to contrast B. spissa with B. pacifica but to contrast the pore density (which does not vary with sediment depth) with the size (which varies with sediment depth). This part has been adapted for clarification.

 Minor grammatical issues

I've made editorial suggestions directly on the pdf, which I hope will improve the clarity of the text.

In places, paragraph breaks are only indicated by a carriage return. Please indicate them by leaving a blank line, as you do elsewhere in the text.

Please hyphenate 'bottom water' when used as an adjective (bottom-water oxygenation')

'Foraminifera' is a noun. Please write 'foraminiferal' when using the taxon name as an adjective - e.g. 'foraminiferal species', 'foraminiferal assemblages' (not 'foraminifera species' etc). You could also use the phrase 'assemblages of foraminifera'.

Response: thank you for additional editorial comments. We have corrected this category of grammatical error and taken into account the suggestions made in the PDF as well.

Response Ellen Thomas

We are very grateful for your review and suggestions for improving our manuscript. Below is a detailed response of changes implemented (in green). In some cases, we kindly disagree with the suggested changes. In those cases, we provide an explanation.

Comments on:

Reviews and syntheses: Review of proxies for low-oxygen paleoceanographic reconstructions Hoogakker et al. (50+ authors)
by Ellen Thomas

Overall, this manuscript is a great resource for the community working on (paleo)oxygenation of the oceans, with the large numbers of authors clearly contributing to the highly various (and complex- aspects of oxygenation proxies, ranging from sedimentology to chemistry to biology. I think this review is a great service to the broader community, but despite my overall appreciation I have suggestions/recommendations. In view of the length and breadth of the paper (with only some topics within my expertise) I will not comment on the text in detail, but highlight some sections which from my personal point of view could be improved. In addition, I think that the paper would become easier to follow (important for such a complex and long paper) with reorganization of some of the sections. In addition, and maybe hard to prevent in a very long and muti;-authored paper, in my opinion there are - specifically where foraminifera are discussed (or maybe also in the biomarker section) - basically too many sections in which overlapping information is discussed (morphology, pores, wall thickness, shape, carbon isotopes). In my opinion there should be an introduction section for foraminifera in which shared information can be presented just once (e.g., denitrification is now discussed in various sections) before the proxies are debated, and the later sections should be simplified through reintegration.

Response: we are grateful for the in depth review of our manuscript and the useful comments proposed by Professor Thomas. We have adopted the proposed changes in terms of reorganizing some of the sections where possible, and added an introduction section for foraminifera where information is presented once before the proxies are discussed, and simplified the denitrification sections.

List of proxies: one fairly new proxy for OMZs are biogenic magnetic particles (e.g., Chang, Hoogakker et al., 2023, Indian Ocean glacial deoxygenation and respired carbon accumulation during mid-late Quaternary ice ages, NatureComm 14, 4841. Might be nice to mention at least shortly as another potential/beginning proxy?

Response: agreed, we have added this to our Concluding summary statement and future direction section.

**Line 94**: OMZs and/or ODZs? ODZs used in e.g., lines 866, 987, 1083, 1127, 1249, 1301, 1374, 1515 and more; please define both acronyms. Are they used for the same phenomenon or are they used specifically to dis;nguish between oxygen **minimum** and oxygen **deficient** zones? In 1521 ODZ seems to be defined as Oxygen Minimum Zone (OMZ)?

Response: this is discussed in Figure 1. We now refer back to this figure when mentioning the acronyms, and checked the use of nomenclature throughout the

manuscript. Additional explanation has been added to Figure 3.

**Lines 185- 209: Figure 1 (and 3)**: This is a good figure to include to the introduction to the paper. However, I think it misses some important information on 2 topics, relevant especially for 'older' literature (including papers cited later on in the ms), despite the text above this figure, explaining that there has been inconsistent and confusing nomenclature.

Topic 1: In marine science, oxygen levels used to be expressed in ml/L (however unfortunate), and terms for different levels of oxygenation such as 'dysoxic' have been defined in ml/L units. Even in recent publications we see data in ml/L, e.g., oxygenation as derived from ichnofossil assemblages (e.g., Rodriguez-Tovar, 2022; 2021-Earth Sci Rev 216, 103579: oxic - 8-2 ml/L; dysoxic 2-0.2 ml/L; suboxic 0.2->0 ml/L; anoxic 0 ml/L, i.e. as in Tyson & Pearson, 1991, see figure below).

Topic 2: Authors (e.g., chemists and biologists) have used (very) different definitions of terminology (see figure below, from Jorissen et al. 2007). Quite a few papers cited in this manuscript (e.g. Kaiho, 1994; Bernhard & Gupta, 1999) used different definitions of, for instance, 'dysoxic', and a commonly used term (suboxic) is not mentioned.

Response: We agree that the use of units and terminology for geochemical zonation is inconsistent in the literature and address this inconsistency in section 3. Rather than adding another level of complexity to the figure itself, we have opted to include this information in the figure caption. Please see our reply to the following comment for further details.

True, the caption to Figure 1 says it provides the values for 'Anoxic', 'Dysoxic/hypoxic' and 'Oxic' as **'most often** associated with specific terms of oxygen concentration', but how often we see which term depends upon the date of publication and the field of expertise, and it is easy to find examples of a different use of terms. As is, the text suggests to the unwary reader by its use of the terms placed along the vertical black bar of varying thickness (what does the thickness indicate?) that the sequence 'oxic-dysoxic/hypoxic-anoxic' provides the terms generally used, and that these terms mean the same in most cited papers. But the terms do not mean the same in various papers cited in section 6.6.2 (see figure below). and the excellent review by Glock 2023 (one of the co-authors of this ms) uses 'anoxia: $O_2$ - $0\mu M$; suboxic condi;ons: $O_2 \sim 1\text{-}10\ \mu M$; hypoxia $O_2 < 62.5\ \mu M$, which is not as shown in fig. 1.

Therefore I think that this manuscript aiming at a broad audience should make its audience familiar with the fact that there is confusion in terminology, and should provide a definition of terms (with reference to authors who used these terms), explaining that different definitions are used in the literature. An approximate scale for ml/L (at standard conditions, not calibrated for temperature/pressure, as e.g. shown in the values in line 2004) should be added to figure 1 (maybe also 3).

Response: Oxygen boundaries differ between different authors and disciplines, and trying to have a consistent scheme across all the different proxies is too challenging. This is one of the reasons that we used fading colors to represent uncertain thresholds. We have included mention of suboxic in the revised manuscripts but point out that geochemists are especially wary about using it. The varying thickness for the black line (next to ODZ) is a way of indicating uncertainty in usage, which has been added to the caption. We feel that with the changes added the figure and caption are now easy to understand. Ultimately, there are nearly as many definitions of these terms as there are authors using them. The result is confusing, and therefore we attempt only to summarize common usage, rather than to offer new definitions. Also see comment by reviewer 3.

**After reading much of the text**, I also wonder about the organization: would it not be better to have the discussion on oxygenation in the water column somewhere here, early on in the discussion? In my opinion that would make more sense, and then the concept of wanting to have planktic as well as benthic proxies can be placed upfront rather than very late in the paper. It could then also be mentioned here that benthic proxies work where the seafloor is within an ODZ/OMZ, but that we need planktic proxies to get an idea of the spatial extent of such zones. In addti;on, this discussion of the water column structure is highly relevant to the sections on Nitrogen-based proxies (6.4) and on Biomarkers (6.3), which now are placed before the section on water column structure (early part of 6.5). In my opinion this is rather important, since it could make the paper much easier to follow for people from outside the direct oxygenation-community.

Response: excellent idea, we added a subsection introducing foraminifera and important aspects (section 2.3.1).

**Lines 215-220**: the manuscript is said to be '*limited to proxies that can be applied through the Cenozoic*'...' *although we briefly touch upon some well-studied earlier examples, such as Cretaceous oceanic anoxic events (OAEs)*.' I agree with these statements, but think that there should be more explanation, because the reasons for time limitation are not just age of sea floor/recovery by drilling projects, or average state of preservation. This manuscript deals extensively with proxies based on foraminifera, i.e., their test morphology or chemistry. By far the most diverse and abundant living group of Foraminifera are the Rotaliida, which differ in morphology (test growth) and mode of calcifica;on - thus also trace metal incorporation - from other groups (e.g., Miliolida and Nodosariida; de Nooijer et al., 2023, 500 million years of foraminiferal calcifica;on, Earth Sci Rev 243, 104484; and references therein). The Rotaliida mainly diversified (arguably during the Mid Mesozoic Revolu;on) somewhere between the start of the Albian (~113) through the Santonian (~84), though the rate of diversification is not well constrained (e.g., Tappan & Loeblich, 1988, Foraminiferal Evolution, Diversification, and Extinction, Journal of Paleontology, 62 (5), 695-714; Kaiho, 1994, Phylogeny of deep-sea calcareous trochospiral benthic foraminifera: evolution and diversification. Micropaleontology 44 (3), 291-311). In my opinion, the authors should mention that the

proxies linked to foraminifera are limited through the evolutionary processes of benthic foraminifera and can be used from the latest part of the Cretaceous (~Campanian-Maastrichtian) on to Recent. For earlier times we may have problems in using an actualistic approach: benthic foraminifera across the Cretaceous OAEs were not necessarily analog to modern forms (e.g., bolivinids as we know them did not yet exist), thus potentially limiting proxy use.

Response: we agree wholeheartedly with this suggestion and have added text reflecting this in the revised manuscript.

**Lines 239-on: SecEon 6.1: Sediments as Proxy Carriers**.

6.1.1. Historical based sedimentary redox/bonom water oxygen reconstructions.

This section discusses laminations. Of course, absence of laminations is commonly due to a lack of bioturbation, and I would have liked to see a clear discussion of ichnofossils/bioturbation as tracers (quantitatively) of oxygen levels. There is a large literature on this topic, as e.g. reviewed by Rodriguez-Tovar in 2021 (Ichnology of the Toarcian Oceanic Anoxic Event: An under- estimated tool to assess palaeoenvironmental interpretations, Earth Sci Rev 216, 103579) and the cited Rodriguez-Tovar 2022. This topic is important not just for reconstruction of oxygen levels with a proxy that can be used back into deep time, but also for understanding oxygenti;on of sediment and its spatial heterogeneity, thus for the discussion in section 6.2.
Bioturbation is mentioned in the following section 6.1.2, but there the emphasis is on methodology (non-destructive), and ichnological reconstruction of oxygen levels has been done for many years before the availability of CT scanning (e.g. Francus, 2001, J Sed Res 71 (3), 501- 507; Nicolo et al., 2010, Paleoceanography 25, PA4210; Rodriguez-Tovar 2021). I thus think that a section on ichnofossils should be inserted in 6.1.1, or - alternatively- the authors could consider trace fossils as fossils and insert text in section 6.1.3.
As to CT scanning- see also Salas et al., 2022, J Petr Sci Engin 208, 109251.

Response: thanks for this suggestion, we have added this to the text.

**Line 546**: 6.2.3.2 typo - race rather than Trace

Response: we have changed this in the revised manuscript.

**Line 605-secEon 6.2.4**: in my opinion this section is not optimally placed - it breaks up the chemical discussion by getting into Ba as a tracer for produc;vity (not directly relevant to oxygenation proxies), and the organic carbon supply is a confusing proxy for both chemical and biological oxygenation proxies. Might it not be better to have a separate section on organic matter supply / effects on oxygenation of bottom/pore waters maybe before starting to talk about geochemical proxies, i.e. after the introduction in section 6.1?

Response: different proxies are affected in different ways (directly and indirectly) by organic carbon supply. For each proxy these caveats are discussed, and including a separate section does would potentially cause more confusion. We have thus removed this paragraph from the paper.

**Line 622: 6.2.5.4** - particulate shuttles; I wonder whether placement of this section is optimal. It refers back to the earlier section on trace element proxies, i.e. before the section on diagenesis i.e., directly after section 6.2.3.2?.

Response: We think it is fine where it is; disagree that directly after section 6.2.3.2 would be a better place for it.

**Line 782**: 'Towards more quantitative oxygen proxies..' text is not unequivocal, i.e. does 'more' refer to quantitative (i.e. get truely quantitative rather than semiquantitative proxies (which is what I think is meant) or does 'more' refer to proxies? we need more proxies?

Response: this refers to transitioning a qualitative proxy to semi-quantitative or fully quantitative proxy. We made sure the text is unequivocal and deleted more.

**Lines 896-889**: factor b)- mention the words 'biological pump'? that is what is described here, right

Response: the processes we summarize under factor b) are more diverse and broader than the biological pump. For example, they also include the availability and composition of ballast material, lateral transport/advection, and heterotrophic remineralization during export through the water column. We added this context to the sentence.

**Lines 1377-1380**: *'When foraminifera build their chambers, they form an organic sheet between calcite layers to facilitate the calcificaEon process'* - First, this would be valid for rotaliid foraminifera, not miliolids (see e.g. de Nooijer et al., 2009, Foraminifera promote calcification by elevating their intracellular pH, PNAS 106, 15374-15378a0, or lagenids (de Nooijer et al., 2023). Then, rotaliid foraminifera first produce an organic layer - the primary organic sheet (in the shape of the chamber to be formed), then precipitate the calcite (or aragonite) on that organic layer (e.g., de Nooijer et al., 2009, 2014, Biomineralization in perforate foraminifera, Earth Sci Rev 135, 48-58). This text reads as if the forams put the organic layer between the calcite layers. Line 1379 *'are encased within the shells after calcification'* also reads a bit 'off' - they are 'encased DURING calcification, I would say. Oscar et al. 2016 missing from reference section.

Response: in our revised manuscript we have changed this, plus we changed the reference to Branson et al. (2016, first name Oscar!)

**Line 1401**: ..'protective as in foraminifera tests' - in my opinion it is not necessarily generally accepted that foraminiferal tests are for 'protection'- against what?. Most organisms that eat forams take them up, test and all, indiscriminately (benthics by deposit feeders such as holothurians and Dentalium, plankton by suspension feeders).

Response: this is meant to be a reflection of the organic matter from the primary organic sheet being protected, not the foraminifera cytoplasm, and we have changed this in the revised text.

**Lines 1490-on**:

The whole first part (through lines 1664) should, in my opinion, be given a different title, and placed elsewhere in the paper. It is a solid description of elemental behavior

in the water column, without foraminifera being considered. Should this section not go to the beginning of the whole paper, i.e. BEFORE section 6.2.3? Maybe even directly after, or integrated within section 3? After all, this section is also relevant to the discussion of N proxies and the discussion of organic proxies earlier on.

Response: We prefer to keep this in this section (now section 5), and included some further references in the other sections.

**Figure 8**: once again, a very good explanatory figure. However, there is no vertical scale. Could we see at least a suggestion of what the depth range is of 'deep', 'intermediate' and 'shallow'? Is the latter used for shelf sites, the middle one for continental margin sites, or what? Are these indeed 'ODZ' = oxygen deficient zones? or could they be OMZ 'Oxygen minimum zones.'?

Response: All ODZs are OMZs but not visa versa. The ODZ term comes from the nitrogen literature and tends to imply (sometimes explicitly defined as such, but not always) an OMZ where denitrification occurs. We have better defined our usage of the terms throughout, and agree that OMZ might be the better term here. We added to the caption: Shelf= shallow, slope = intermediate, and abyssal plain = deep.

**Line 1497**: maybe mention that benthic proxies can be used over a large part of the Phanerozoic, planktic proxies only after the Jurassic/Cretaceous?

Response: we added this constraint to the introductory paragraph 2.3.1 about foraminifera.

**Line 1555 and on**: I think that sec;on **6.5 Foraminifera trace elements** might be better placed after section **6.6 Foraminifera assemblages**, since in that section foraminifera are generally described. **What I am missing**, before a discussion of trace element incorporation and vital effects, is a section broadly on **foraminiferal calcification.** We are now starting with planktic foram elemental incorporation without the reader being made aware of any knowledge of foraminiferal calcification processes, and of the fact that planktic forams have been evolutionarily derived from benthos many times, but only from the group Rotaliida, and that their calcification thus is limited to what we see in one group of benthos (e.g., Morard et al., 2022, Renewal of planktonic foraminifera diversity after the Cretaceous Paleogene mass extinction by benthic colonizers, Nature Comms 13, 7135). Reference to the broad overview in de Nooijer et al., 2023, 500 million years of foraminiferal calcifica;on Earth Sci Reviews 243, 104484 would be a good start to such as section (with many references in that paper). In my opinion in introduction on foraminiferal calcification is absolutely needed, e.g., before the text in line 1642-1675. The text in lines 1846-1858 could be incorporated in the section on calcification.

Response: we have added an introductory section about foraminifera with a short section about calcification.

In my opinion it would be better to re-organize the text on elemental/Ca values, so that all text is organized by element, rather than hopping from I/Ca to Mn/Ca to U/Ca several times.

Response: It is our preference to maintain an integrative discussion of the proxies. Each of the proxies have already been previously reviewed on their own in other places. The significance here is that we reviewed them together in a way that allowed us to compare and contrast. To some extent, these proxies have similarities that warrant mentioning them together. The paragraphs have a clear structure each time looking at I/Ca, Mn/Ca and then U/Ca, so that the "hopping" is systematic and the reader should not have difficulties to what the information useful for him/her. That said, we also interpret this feedback as indicating that some aspects of the current organization make the sections difficult to follow. In response, we have given the section a careful edit to ensure to improve readability and integration of the proxies into the same section.

**Line 1567-1572**: Winkelbauer et al., 2023 - but see comment on this paper, Lu et al., 2023, *Frontiers of Marine Sciences*, 10, 1095570.

Response (**check with Dalton, Sha, Babette**): we are not sure what is being referred to here. The comment by Lu et al. (2023) mainly reiterates what is being discussed in the Winkelbauer et al. (2023) manuscript. We added this sentence "Lu et al., (2023) confirm that proxy data from plankton tows and sediment core top samples may not necessarily agree with each other because of the complexity of foraminiferal calcification and postdepositional overprints in marine surface sediments."

**Line 1589**: please see my notes at the end of the next section - the genus *Ammonia* is a very shallow water taxon (inner neritic - intertidal), has been recorded as living infaunally (see below), and it remains a question how relevant its biology is for deeper water taxa.

Response: we added a sentence reflecting this.

**Line 1865-on: SecEon 6.6: Foraminifera Assemblages**.

I greatly appreciate the work by many authors starting with Kaiho (1994, 1999) through Kranner et al 2022 to develop a foraminifera assemblage -based, quantitative proxy for oxygenation, but in my opinion this proxy (broadly, BFOI-based) needs more discussion. Authors including Buzas et al 1993 (A statistical evaluation of the microhabitats of living (stained) infaunal benthic foraminifera, Mar Micropal 20 (3-4), 311-320), Gooday (2003) and (Jorissen et al. 2007) - and more - discussed various problems/uncertainties with the BFOI (some of which are in my opinion not resolved in the EBFOI), but their arguments are not represented in this paper.

Response: we added a comment that the EBFOI resolves some, but potentially not all uncertainties associated with the BFOI.

What problems can there be with this widely used (and modified/improved, e.g., Kranner et al., 2022) proxy? After all, numerous authors (since e.g., Smith 1964 and earlier) observed a correlation between 'morphotype' (size, thickness of wall, chamber arrangement - shape, porosity) and oxygenation levels, with species indicating lower oxygen levels more commonly living infaunally. Kaiho (1994, 1999, Effect of organic carbon flux and dissolved oxygen on the benthic foraminiferal oxygen index (BFOI), Mar Micropal 37, 67-76) defined the BFOI by assigning taxa to be indicators of oxic, suboxic or dysoxic conditions: '*Dysoxic (0.1-0.3 mL/L), Suboxic (0.3-1.5 mL/L), and Oxic (>1.5 ml/L) indicators based on the basis of the relaEon between specific morphologic characterisEcs (or species composiEon) and oxygen levels*' (cited from Kaiho 1994), with dysoxic and suboxic species living deeper in the sediment than oxic indicators. In order to calculate the BFOI, we thus assign an indicator status to all (in Kaiho's case, all calcareous) species present in samples.

But how do we assign species to the D, S or O group? Lists of species indicating whether each is D, S or O usually do not always contain a reference with species names provided with a reference showing on which data the assignment is based. For living species, one can use direct observations (though these are limited for the deep sea), for extinct species we can use stable carbon isotope or trace element data. However, such direct observations are not available for all the ~ 2000+ living species of benthic foraminifera (Murray, 2007, Biodiversity of living benthic foraminifera: How many species are there?. Mar Micropal 64 [3-4], 163-176), let along for extinct species. In practice, therefore, assignment for many taxa is based on **morphological similarity to species for which data are available**. However, the link between morphology and microhabitat - thus oxygenation - (epifaunal-shallow infaunal-deep infaunal) showed an accuracy of only 75% (Buzas et al., 1993). These authors (and others, e.g., Jorissen & Sen Gupta, 2003, Benthic foraminiferal microhabitats below the sediment-water interface, in 'Modern Foraminifera', 161-179) show evidence that there are few to none actual 'epifaunal taxa' in soft sediments- foraminifera cannot live on top of soft, sloppy sediment, except for these that live on objects sticking out above the sediment as reported for *C. wuellerstorfi*. However, even the widely used epifaunal 'oxic indicator' *C wuellerstorfi* survives under low oxygen condi;ons (e.g., Venturelli et al., 2018,

Frontiers Mar Sci 5; Rathburn et al., 2018). Then, 'small specimens' (<350 μm) of 'oxic indicators' are defined 'suboxic A indicators'(Kaiho, 1994, 1999), but how certain are we about placing the boundary between 'large' and 'small' at 350 μm? I thus think that the assignment of species to 'indicator groups is not that simple, and we do not know the errors involved. Therefore, what error bars should we consider in calculating a BFOI, especially at somewhat higher oxygenation levels (e.g., Kaiho's 'suboxic indicators A, B and C are not that clearly defined, see Kranner et al., 2022)? Kaiho 1999's figure 2C (see below) indicates error bars which are quite large at somewhat higher oxygenation levels (and that is where we see the differences with EBFOI, Kranner et al., 2022, making BFOI more semiquantitative than quantitative. This is even more so since we now have ample evidence (not available when Kaiho wrote the BFOI papers) that many species of benthic foraminifera can survive and flourish at very low oxygenation/no oxygen and many prati;ce denitrification (broad literature cited in this ms and in Glock 2023).

What I want to argue is that one should consider that BFOI and similar proxies are empirical, i.e., we do not really know WHY (for instance) a large trochospiral foraminifer should indicate higher oxygenation levels than a small, flat biserial foraminifer: correla;on is not causation. We do not really know what (if any) limitations a test form imposes: after all, both trochospiral and biserial tests are represented in planktic and benthic species. There are speculations as to volume/surface, but the fact remains that some large trochospiral foraminifera in some habitats have no problems with anoxia: ***Ammonia* species are large trochospiral species** (much used in studies of calcification and ecology) living in shallow coastal waters (neritic into intertidal), and they survive and even calcify under anoxic condi;ons (Nardelli et al., 2014, Biogeosciences 11, 4029-4038), and have been observed living infaunally down to 35 cm in the sediment (Moodley & Hess, 1992; Tolerance of infaunal benthic foraminifera for low and high oxygen condi;ons, Biol. Bull. 183, 94-98). Note that the remarks in line 2061 on circular foraminifera, and in lines 2170-2171 on large spirally arranged tests, and the discussion small/large foraminifera (sec;on 6.7.2.2) thus are not universally valid, even in the present world. We should probably question more clearly (as for chemical proxies) under which specific conditions our biotic proxies work - in agreement to what the authors say in lines 1937-1940. I do **not** think, that the statement that **'All of**

[Figure]

**these indices are considered valid (line 1936)**' is evidence based - how has that been tested? For me personally, Figure 13 is quite discouraging, if we want to consider the 'assemblage proxies' as quantitative: the compared proxies agree as to where oxygenation values are higher or lower (i.e. qualitative agreement), but the actual values - plotted on a logarithmic scale! - show very large differences, with some plotting at around 20 for the present value - which should be around correct, but others at >100 micromolar for the same sample (hypoxic and oxic in figure 1); lower values for the same samples are around 2 micromolar for the BFA method (dysoxic to anoxic), around 70 for the BFOI method dysoxic to oxic). I would therefore say that the biotic proxies, like the chemical incorporation proxies, need a considerable amount of work. Possibly - but this is looking far into the future, probably, we might do better work in both biotic and chemical proxies if we were to gain actual understanding of the 'vital effect' and its

genetic/metabolic base, nowadays pretty much a black box. Such papers as Ujie et al 2023 (Unique evolution of foraminiferal calcification to survive global changes, Sci Adv 9, eadd3584) point towards potential developments, and collaboration with biologists working at understanding calcification processes might also offer new insights (e.g., Davila-Hernandez et al., 2023, Directing polymorph specific carbonate formation with de novo protein templates, Nature Comm, 14, 8191).

In additon, I think that we should consider to what extent models defined for foraminifera living at below-shelf depth condi;ons are useable for intertidal-neritic dwellers. Possibly, trochospiral *Ammonia* differs from trochospiral *Cibicidoides* through its possession of an internal canal system, so we should look carefully at how we characterize taxa. These food-replete environments might not be well understood through the TROX model. This could be important, since shallow-water taxa are much easier kept in the lab, and we thus might wonder whether observa;ons on such taxa are valid for their deep-sea rela;ves. I thus agree with the authors (line 1950) that *'different approaches are appropriate for different environments and quesEons'*.

Response: we thank professor Thomas for these comments and will add additional information and discussions to this section, as discussed in the comment. We understand the criticism that *Ammonia* spp. do not fit into the general morphology approach, due to it´s trochospiral shape. Nevertheless, it is usually endemic in very shallow coastal environments, especially intertidal mudflats, which are usually not the aimed locations for the morphology based reconstructions. We have included this issue into the discussion and emphasize that the choice of the environment for the paleoreconstruction matters. Also, we have been more critical regarding the contrasting results in fig.13, since despite the different approaches are showing the same trends, they significantly differ quantitatively.

**Line 1954**: At the end of this sec;on, I want to mention that I also missed any reference to Jorissen et al., 2022. The 4GFOR model - coupling 4G early diagenesis and benthic foraminiferal ecology. Mar Micropal 170, 102078. In this paper we see a first effort to link benthic assemblages not just to oxygenation, but to the several ;mes mentioned 'redox ladders ' in the sediment (e.g., Figure 3). In my opinion this is a highly significant paper that should be men;oned. This could go in the section 6.6.6, as well.

Response: Good point, we have added information about the 4GFOR model of Jorissen et al. 2022 in the revised manuscript.

**Line 2002** (sec;on 6.6.6 Proxy drivers): I think that this section could be placed in a separate, much earlier section, e.g. placed just before a section on foram calcification, and BEFORe the proxies are discussed. This is information we should have had before discussing the proxies. Line 2053-2054: In my opinion, the reference to Glock's 2023 review paper should have come much earlier in the text, which text then could have been guided by this recent review.

Response: we revised this section and call it Environmental influences.

**Line 2022**: an early an important paper on environmental controls on benthic foraminifera including *Uvigerina* (organic carbon - oxygen) that should be cited is was Lutze & Coulbourn, 1984, Recent benthic foraminifera from the continental margin of northwest Africa: Community structure and distribution, Mar Micropal 8, 361-401.

Response: we have added this to the revised version of the manuscript.

**Line 2058-on, secEon 6.6.6.2**. This short section is not very useful, and gives no description of planktonic foraminifera in low oxygen environments. In fact, relevant text is presented in the section on porosity, lines **2208-2016**). I suggest to either remove this section, or just refer to ] section 6.7.2, where important references are provided.

Response: We disagree with this suggestion. The text following 2208 is about porosity of planktic foraminifera, while the paragraph from 2058 discusses planktic species that might be able to survive under low O2 conditions. Of course, there is not so much information in this paragraph compared to the benthics. But this is the whole point of this section: To mention our gaps of knowledge about planktic forams from low O2 habitats. We have removed subheaders referring to benthic and planktic foraminifera to reduce complexity.

**Line 2076**: an important early review paper on morphology (including thin-walled tests): Boltovskoy et al., 1991, Morphological variations of benthic foraminiferal tests in response to changes in ecological parameters: a review, J of Paleontol 65, 175-185.

Response: We added this reference to the revised version of the manuscript.

**Line 2083-2084**: excellent examples of evidence for such sediment mixing are papers by Hupp et al., 2022, PNAS 119, e2115561119; Hupp & Kelly, 2020, Paleoc PaleoCl 35, 1-19; Hupp et al., 2019, Geology).

Response: We added these references at the appropriate sections to the revised version of the manuscript.

**Lines 2095-2100**: fairly superficial; paragraph. Presence of non-analogue fauna of course in the first place is due to evolution/extinction. One factor that is not mentioned specifically: coexistence of taxa in samples that lived in different seasons (e.g. seasonal anoxia is. common phenomenon); see e.g. Stassen et al., 2015 Mar Micropal 115, 1-23; or Wagner et al., 2023 Paleoc PaleoCl 37, e2022PA004502.

Response: Fair point, we added information about coexistence of taxa that lived in different seasons.

**Line 2155: secEon 6.6.9.**
This text could have been helped with an earlier general introduction to foraminifera, their calcification and evolution, as mentioned above (line 1555). Calcareous hyaline foraminifera (Rotaliida) outnumber other groups since the Late Cretaceous. Specific taxa nowadays more common in oxygen rich environments (Nodosariida) are not necessarily so in deep-time (including the time of OAEs).

Response: yes, we followed up on Prof. Thomas' earlier suggestion of having an earlier general introduction to foraminifera, which addresses this (+ include the comment here).

**Line 2160**: note that these (*Dentalina, Lagena, Nodosaria* - and *LenEculina*) are all

Nodosariidae, which use a different mode of calcification (de Nooijer et al., 2023, and Pacho et al., 2023, Element/Ca ratios in Nodosariida (Foraminifera) and their potential application for paleoenvironmental reconstructions, Biogeosciences 20, 4043-4056).

Response: Good point, we added this detail. It will be mentioned in the about foraminiferal calcification section.

**Line 2161**: as mentioned above, the statement on more circular foraminifera dominating on oxygenated environments is not valid generally even in the present world: in inner neritic- intertidal settings: large trochospiral *Ammonia* species are among the most anoxia-surviving taxa in the world. I greatly like Tetard et al.'s 2017 work and think that it is extremely useful, but it should be kept in mind (as mentioned in this paper) that one can use such shape analysis only in specific settings. Also relevant to circularity in Table 2 (lines around 2220). As long as we do not know **why** large trochospiral taxa in deeper water setting are more common in oxygen-rich settings, we cannot use this observation for all environments.

Response: We added these caveats to the text.

**Line 2164-2166, 2070-2071**: here again - test shape and size cannot that simply be understood in a universal way - see notes on *Ammonia*.

Response: We added this caveat.

**Line 2175**: see notes above- morphology is not that simply/universally a reac;on to environment.

Response: we have addressed this in the revised manuscript. Maybe not universally interspecifically but intraspecifically it usually reflects phenotypic plasticity, which should reflect the environment. Otherwise: Why should forams put energy into morphologic adaptation, if not due to changes in the environment?

**Line 2204**: the important word here, in discussing porosity, is 'conspecifics'. Porosity has been used successfully to look at oxygenation in shallow waters (e.g. various cited papers by Richirt et al), but pore size in *Ammonia* species is an interplay between its genetic identity and environmental factors (e.g. Hayward et al., 2021. Molecular and morphological taxonomy of living Ammonia and related taxa (Foraminifera) and their biogeography, Micropaleontology 67, 109-313). this thus might become a problem if the 'conspecifity' is not that easily worked out, as e.g. in *Ammonia*. Note that similar problems can be predicted for other neritic/intertidal taxa such as *Elphidium clavatum* (Table 2- note typo in *Elphidium*).Table 2: circularity in plankton: by far the most plankton (presently) are circular, and occurrence of biserials/triserials is generally linked to high food (rather than low oxygen) conditions.

Response: We do not think this comment helps to improve the review. Most plankton lives in better oxygenated environments and round forms typically have lower surface to volume ratios, which could represent this. If this is refers to planktics then it could only be the meroplanktic bolinivinids that are being referenced here. We certainly find them in both the high-productivity, low-oxygen California borderlands and the low-productivity, relatively low-oxygen North Atlantic gyre. Plus the Mediterranean. So, I'm not sure they're either canonical or functionally indicative of either...
In benthic environment f there is high food, there is most likely also lower O2. In addition, non-circular forams often are found in low-oxygen environments. Take bolivinids for example. They are conical and have a very high surface to volume ratio, which is considered to be an adaptation to low O2 conditions. They are also often found under low O2 conditions. The Ammonia problem is something completely different. Of course we get problems, if we can not distinguish species properly because they look so similar.

**Line 2239-on: Size and Morphotype**. Would be nice to cite Schmidt et al. 2004 paper in Science: Abiotic forcing of Plankton Evolution in the Cenozoic (303, 207-210), for a more long-term view of size of planktonic forams.

Response: We added this citation to the revised version of the manuscript.

**Line 2300**: as to acidification/dissolution see also Foster et al., 2013, Surviving rapid climate change in the deep-sea during the Paleogene hyperthermals. *PNAS*, 110: 9273-9276, and Schmidt et al., 2018.

Response: We added this reference to the revised version of the manuscript.

**Line 2434**: for ostracods, oxygen morphology linkages have been very long discussed - whether one thinks these linkages are correct or not. See e.g. McKenzie et al., 1989, The *KRITHE* problem — first test of Peypouquet's hypothesis, with a redescrip;on of *KRITHE PRAETEXTA PRAETEXTA* (Crustacea, Ostracoda), Palaeo[3] 74, 343-354.

Response: We mention this paper at the appropriate section in the revised version of the manuscript.

**Line 2477: SecEon 6.8.1**: in my opinion this section could also benefit from a general benthic foram introduction.

Response: We added a general foram introduction to section 2.

**Lines 2559-2562**: There is discussion (various papers co-authored by Jorissen et al.) whether 'epifaunal' is a correct term for many species, with the exception of species living on hard subject sticking out above the sediment/water interface; see discussion. of use of 'Average Living Depth). Many species of *Cibicidoides* are biconvex, thus do not live attached to surfaces, and thus may not be truely 'epifaunal'. This is further discussed in sec;on 6.8.5 (lines 2606 on); in my opinion this discussion - what is infaunal and what not - should have been provided before the �8$^{13}$C proxy was discussed.

Response: We have made these changes to the revised version. The terms epifaunal and infaunal are introduced and discussed according to reviewers suggestions in the foram section.

**Line 2560**: *C. mundulus* is the correct name, *C. kullenbergi* is the junior synonym (see book by Holbourne et al. on benthic foraminifera), not the other way around.

Response: We have changed this around.

**Line 2621**-on: not mentioned, but in my opinion the most important limiting factor for use of the d$^{13}$C proxy is the fact that we see common deep infaunal taxa ONLY if the food supply is sufficiently high (see TROX model, Jorissen et al 1995), whatever

species it is. The use of this proxy is thus limited to regions and depths where there is such a sufficient food supply.

Response: Yes, this is absolutely true. We mention this in the revised manuscript.